# Steerable Adversarial Scenario Generation through Test-Time Preference Alignment

**Tong Nie**[1,2*], **Yuewen Mei**[2*], **Yihong Tang**[3], **Junlin He**[1], **Jie Sun**[2], **Haotian Shi**[2],
**Wei Ma**[1†], **Jian Sun**[2†]
[1]The Hong Kong Polytechnic University, Hong Kong SAR, China
[2]Tongji University, Shanghai, China
[3]McGill University, Montreal, QC, Canada

## ABSTRACT

Adversarial scenario generation is a cost-effective approach for safety assessment of autonomous driving systems. However, existing methods are often constrained to a single, fixed trade-off between competing objectives such as adversariality and realism. This yields behavior-specific models that cannot be steered at inference time, lacking the efficiency and flexibility to generate tailored scenarios for diverse training and testing requirements. In view of this, we reframe the task of adversarial scenario generation as a multi-objective preference alignment problem and introduce a new framework named **S**teerable **A**dversarial scenario **GE**nerator (SAGE). SAGE enables fine-grained test-time control over the trade-off between adversariality and realism without any retraining. We first propose hierarchical group-based preference optimization, a data-efficient offline alignment method that learns to balance competing objectives by decoupling hard feasibility constraints from soft preferences. Instead of training a fixed model, SAGE fine-tunes two experts on opposing preferences and constructs a continuous spectrum of policies at inference time by linearly interpolating their weights. We provide theoretical justification for this framework through the lens of linear mode connectivity. Extensive experiments demonstrate that SAGE not only generates scenarios with a superior balance of adversariality and realism but also enables more effective closed-loop training of driving policies. **Project page**: https://tongnie.github.io/SAGE/.

# 1 INTRODUCTION

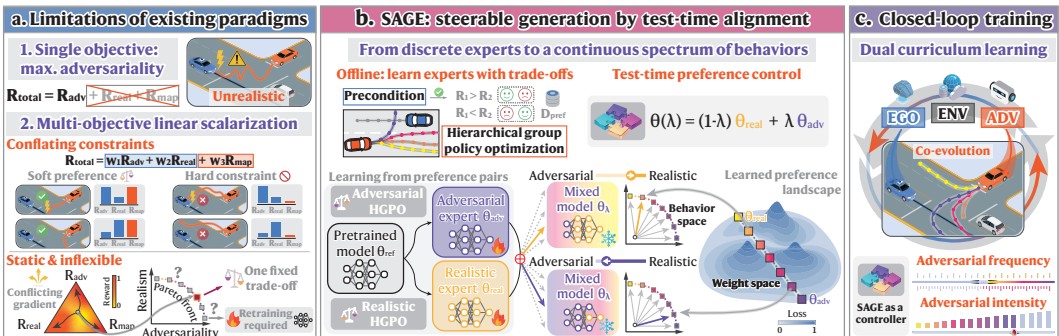

Figure 1: Limitation of existing adversarial generation methods, our solution, and its application.

Safety assessment of autonomous driving (AD) systems is a prerequisite for their large-scale deployment. A cornerstone of verifying and enhancing their capabilities is simulated testing, particularly through scenario-based methods (Xu et al., 2022; Ding et al., 2023). To improve testing efficiency and alleviate the issue of "curse of rarity" (Liu & Feng, 2024), adversarial scenario generation has emerged as a cost-effective technique. Built on this method, practitioners can move beyond simple

---

*Equal contributions.
†Corresponding authors.

testing and enable closed-loop adversarial training, where an AD policy is continuously improved by the very challenges it fails (Ma et al., 2018; Wachi, 2019). This is achieved by systematically creating challenging, long-tailed corner cases that efficiently probe the system's operational boundaries, identify its blind spots, and thereby accelerate the development of more robust driving policies.

Instead of creating scenarios from scratch, many modern generation paradigms operate by perturbing a learned naturalistic driving prior and converting it into an adversarial posterior. Viewed through this lens, whether the adversarial model is built using Reinforcement Learning (RL) (Feng et al., 2023; Ransiek et al., 2024a), guided diffusion (Xie et al., 2024; Chang et al., 2024; Xu et al., 2025), or direct optimization (Wang et al., 2021; Hanselmann et al., 2022; Zhang et al., 2023), the task can be framed as an *objective-oriented adversarial optimization problem*. The core challenge lies in controllably navigating the inherent trade-off between adversariality (capacity to induce a failure) and realism (physical and behavioral plausibility). This controllability is crucial for serving two distinct purposes: targeted stress testing may require highly extreme scenarios, while data augmentation for closed-loop training demands challenging yet realistic behaviors to ensure effective improvement. Fulfilling both needs requires a generation framework that is exceptionally flexible and steerable.

However, existing paradigms that directly solve the adversarial optimization problem fall short in providing this necessary level of controllability, especially at test time (Fig. 1(a)). On the one hand, naively optimizing for a single objective, such as maximizing adversariality, often leads to a collapse in realism. For example, methods based on aggressive search for collisions can generate physically implausible trajectories where a vehicle spins in place or executes a kinematically impossible turn to intercept the ego agent (Zhang et al., 2022; 2023). While adversarial, such scenarios offer limited value as they do not reflect behaviors that could occur in reality. On the other hand, while framing it as a multi-objective problem is a natural solution, the prevailing approach linearly scalarizes multiple loss terms (Chang et al., 2024; Xu et al., 2025), which reduces the problem to managing a set of delicate weighting hyperparameters. This is highly heuristic and prone to instability, particularly when objectives induce conflicting gradients and involve a mixture of soft and hard constraints (Deb et al., 2016). More critically, even a well-tuned model remains tied to a single fixed preference in the parameter space, mapping to a single point in the behavior space of possible trade-offs. This limits their test-time steerability, as adapting to a new preference usually requires expensive retraining.

In this paper, we propose the first paradigm that reframes the solution of the adversarial optimization problem through the lens of preference alignment (Fig. 1(b)). Drawing inspiration from multi-objective alignment (Wortsman et al., 2022; Rame et al., 2023), we tackle intricate relationships between competing objectives on the Pareto front directly from preference data, balancing *adversariality*, *realism*, and *feasibility*, analogous to the "3H" (helpful, honest, and harmless) principles in Large Language Models (LLMs) (Bai et al., 2022). Crucially, our method enables *test-time control* in both weight and behavior spaces. Instead of producing a single model for a fixed trade-off, we derive a mixture-of-preferences model that generates a continuous spectrum of policies. With this posteriori selection, a user can adjust preferences at inference to generate better-informed trade-offs on the Pareto front tailored to specific needs without retraining. This framework is positioned as generalizable, applicable to various methods that employ adversarial optimization, as well as to different driving policies under test. In essence, we shift the paradigm from manually designing weighted objectives to learning a controllable preference landscape. Our contributions are fourfold:

- We introduce SAGE (Steerable Adversarial scenario GEnerator), the first policy- and backbone-agnostic paradigm that reframes the task as a multi-objective preference alignment problem.
- We propose hierarchical group-based preference optimization, a sample-efficient offline alignment fine-tuning technique that decouples hard feasibility constraints from soft preference trade-offs.
- SAGE uniquely provides a principled way for test-time preference control, enabling users to navigate the entire Pareto front to generate tailored scenarios without retraining. Furthermore, we theoretically and empirically ground this paradigm in the hypothesis of linear mode connectivity.
- Comprehensive experiments demonstrate that SAGE establishes a superior trade-off, delivering competitive adversariality alongside the highest realism, and thereby enabling more effective closed-loop training of driving policies (Fig. 1(c)) than state-of-the-art methods.

## 2 RELATED WORK

**Adversarial Scenario Generation.** This problem has been explored through several distinct paradigms. RL methods are well-suited for closed-loop training but suffer from limited transferabil-

ity and a tendency toward unrealistic reward hacking without meticulous engineering design (Kuutti et al., 2020; Feng et al., 2023; Ransiek et al., 2024a; Qiu et al., 2025). Conversely, diffusion-based models excel in generation with controllability (Zhong et al., 2023; Xie et al., 2024; Xu et al., 2025), but their high computational expense currently renders them impractical for the on-the-fly generation required in closed-loop settings. Optimization and sampling-based methods (Wang et al., 2021; Hanselmann et al., 2022; Zhang et al., 2022; 2023) offer a computationally efficient and transferable alternative suitable for closed-loop training. However, they often struggle to balance adversariality with realism, frequently producing physically implausible scenarios without principled control.

**Preference Alignment.** Our SAGE is inspired by recent advances in steering LLMs. This field aims to align model behavior with complex and often conflicting human values, exemplified by the "3H" principles (Bai et al., 2022). Representative methods such as Reinforcement Learning from Human Feedback (RLHF) (Ouyang et al., 2022) and Direct Preference Optimization (DPO) (Rafailov et al., 2023) offer powerful tools for aligning models into an aggregated preference signal. However, a key limitation is that they only produce a single model for a fixed trade-off. To overcome this, recent multi-objective alignment research has shifted towards enabling *test-time control* (Rame et al., 2023; Shi et al., 2024; Xie et al., 2025). They typically train a set of experts, each specializing in a different objective, and then interpolate parameters or predictions at test time to dynamically navigate the Pareto front of trade-offs without retraining. While preference-based methods are emerging in scenario generation (Cao et al., 2024; Qiu et al., 2025; Yu et al., 2025; Chen et al., 2025), they primarily focus on aligning with a single objective, such as realism or safety, by using standard DPO or RLHF during training. Our work is the first to ground the multi-objective test-time alignment paradigm in adversarial scenario generation. By learning the trade-offs directly from preference data, we provide a principled framework for steerable generation without retraining.

## 3 SAGE: STEERABLE ADVERSARIAL SCENARIO GENERATOR

### 3.1 PROBLEM FORMULATION

The goal of adversarial scenario generation is to discover or synthesize situations that reveal the vulnerabilities of a given driving policy $\pi_{\text{ego}}$ (Ding et al., 2023). Instead of creating scenarios from scratch, a common and effective paradigm is to perturb existing, typically naturalistic scenarios $\mathcal{S}_{\text{log}}$ logged from the real world. This approach, which we refer to as *adversarial optimization* (Zhang et al., 2022; Xu et al., 2025), can be abstracted into a general multi-objective optimization problem.

**Definition 1** (Perturbation-based Adversarial Optimization)**.** *Given a naturalistic scenario $\mathcal{S}_{log}$ containing an initial trajectory $\tau_{log}$ for an opponent agent, the goal is to find an optimal perturbation $\delta$ that, when applied to $\tau_{log}$, maximizes the objective $J_{obj}$ while adhering to a set of constraints $\mathcal{T}_{valid}$:*

$$\delta^* = \arg\max_{\delta} J_{obj}(\mathcal{S}_{log}, \delta; \pi_{ego}) \quad s.t. \quad \tau_{log} + \delta \in \mathcal{T}_{valid}, \tag{1}$$

*where $J_{obj}$ typically involves multi-objective trade-offs, and $\mathcal{T}_{valid}$ enforces validity constraints.*

Eq. 1 provides a unifying lens through which various existing methods can be viewed. Trajectory optimization (Hanselmann et al., 2022; Zhang et al., 2022) solves Eq. 1 directly using gradient-based techniques. Sampling-based methods (Zhang et al., 2023) approximate the solution by sampling from re-weighting candidates based on $J_{\text{obj}}$. Diffusion-based generators (Xie et al., 2024; Xu et al., 2025) use $J_{\text{obj}}$ as a guidance signal to steer the reverse denoising process toward adversariality. The trade-off between competing objectives gives rise to a *Pareto front*: a set of optimal solutions where no single objective can be improved without degrading another. However, directly solving Eq. 1 for a fixed $J_{\text{obj}}$ yields only a single strategy corresponding to one point on the Pareto front. Our goal is to shift from optimizing for a single $\delta$ to generating a continuous spectrum of adversarial policies $\{\pi_{\theta,i}\}_i$ that can trace the entire Pareto front at test time. This paradigm is generalizable to various methods grounded in the adversarial optimization problem, enabling steerable scenario generation.

### 3.2 PREFERENCE-ALIGNED ADVERSARIAL TRAJECTORY OPTIMIZATION

Existing adversarial generation methods are often optimized for aggressiveness without sufficient constraints, leading to physically unrealistic or infeasible trajectories. Therefore, we fine-tune a pretrained motion generation model to generate trajectories that are simultaneously *adversarial*, *realistic*, and *map-compliant*. We frame this as a preference alignment problem, developing a new hierarchical preference optimization method that strategically handles the multi-objective nature.

**Multi-Objective Formulation.** Let $\pi_{\text{ref}}$ be a pretrained, probabilistic motion generation model that maps a scenario context $c$ to a distribution over future trajectories for a specific agent. The context $c$ includes the road map geometry $\mathcal{M}$ and the historical states of all agents. Our objective is to learn a new policy $\pi_\theta$, fine-tuned from $\pi_{\text{ref}}$, that generates trajectories $\tau \sim \pi_\theta(\cdot|c)$ biased towards our desired attributes. The quality of a generated trajectory $\tau$ is multifaceted. A naive approach would be to define a scalar preference score function $R_{\text{total}}(\cdot)$ that holistically evaluates a trajectory:

$$R_{\text{total}}(\tau; c) = w_{\text{adv}} R_{\text{adv}}(\tau, \tau_{\text{ego}}; c) - w_{\text{real}} P_{\text{real}}(\tau) - w_{\text{map}} P_{\text{map}}(\tau, \mathcal{M}), \tag{2}$$

where $R_{\text{adv}}$ measures the adversarial potency against the future trajectory of the ego vehicle $\tau_{\text{ego}}$; It should maximize the probability of causing a failure or a near-miss event to expose vulnerabilities of $\pi_{\text{ego}}$. $P_{\text{real}}$ ensures $\tau$ to adhere to the patterns of naturalistic driving; It should be statistically plausible and avoid behaviors that are physically possible but extremely unlikely, which could lead to an overly conservative ego policy. $P_{\text{map}}$ penalizes violations of map constraints (e.g., crossing solid lines or driving off-road). The weights $w_{(\cdot)}$ balance these competing objectives. A trajectory $\tau^w$ is preferred over $\tau^l$, denoted as $\tau^w \succ \tau^l$, if and only if $R_{\text{total}}(\tau^w) > R_{\text{total}}(\tau^l)$.

However, this linear scalarization formulation presents a structural issue: it conflates *soft preference objectives* (trade-off between adversariality and realism) with *hard feasibility constraints* (map compliance). A trajectory that passes through a building is totally invalid, not just less preferable. The binary nature is not well captured by a continuous penalty. This complicates the optimization landscape, which can result in unreliable outcomes where the model only partially satisfies constraints.

**Hierarchical Group-based Preference Optimization.** To address this issue, we propose Hierarchical Group-based Preference Optimization (HGPO), a new framework that decouples hard constraints from soft preferences and improves the data efficiency of the alignment process. First, we treat map compliance not as a component of the reward, but as a precondition. We define a binary feasibility function $F(\tau, \mathcal{M}) \in \{0, 1\}$, which returns 1 if the trajectory $\tau$ is fully compliant with $\mathcal{M}$ and 0 otherwise. The soft preference is then judged by a separate preference reward function $R_{\text{pref}}$:

$$R_{\text{pref}}(\tau; c) = w_{\text{adv}} R_{\text{adv}}(\tau, \tau_{\text{ego}}; c) - w_{\text{real}} P_{\text{real}}(\tau). \tag{3}$$

Second, a direct way would be to sample many feasible trajectories, identify the single best ($\tau^w$) and worst ($\tau^l$) according to $R_{\text{pref}}(\cdot)$, and apply the standard DPO loss. However, this approach is data-inefficient as it discards fine-grained preference information contained within the rest of the samples. Inspired by the efficiency of group-wise learning (Shao et al., 2024), but adapted for an offline setting, we move beyond the single winner-loser pair. For each context $c$, we sample a group of $N$ candidates $\mathcal{G}_c = \{\tau_i\}_{i=1}^N \sim \pi_\theta(\cdot|c)$. We then partition $\mathcal{G}_c$ into a feasible set $\mathcal{G}_c^{\text{feas}} = \{\tau \in \mathcal{G}_c \mid F = 1\}$ and an infeasible set $\mathcal{G}_c^{\text{infeas}} = \{\tau \in \mathcal{G}_c \mid F = 0\}$. From these sets, we construct a dataset of preference pairs $\mathcal{D}_c^{\text{pref}} = \{(\tau^w, \tau^l)\}$ based on the following two principles, which *hierarchically* inject the desired inductive bias: (1) **Feasibility First**: Any feasible trajectory is strictly preferred over any infeasible one. We generate pairs $(\tau^w, \tau^l)$ where $\tau^w \in \mathcal{G}_c^{\text{feas}}$ and $\tau^l \in \mathcal{G}_c^{\text{infeas}}$. (2) **Preference within Feasibility**: Among feasible trajectories, we prefer those with higher preference reward. We generate pairs $(\tau^w, \tau^l)$ where both are in $\mathcal{G}_c^{\text{feas}}$, but satisfy $R_{\text{pref}}(\tau^w; c) > R_{\text{pref}}(\tau^l; c) + \delta_m$. The margin $\delta_m$ prevents learning from noisy preferences where the reward difference is negligible.

**Fine-tuning Objective.** Hierarchical principles enable the model to learn from a wider distribution of suboptimal and near-optimal examples. The preference fine-tuning process optimizes the policy $\pi_\theta$ to explain the preference data better while being regularized by the reference policy $\pi_{\text{ref}}$. Our proposed HGPO extends this to maximize the likelihood of the entire set of group-sampled pairs:

$$\mathcal{L}_{\text{HGPO}}(\theta) = \mathbb{E}_{\substack{c \sim \mathcal{D} \\ \mathcal{G}_c \sim \pi_\theta(\cdot|c) \\ (\tau^w, \tau^l) \sim \mathcal{D}_c^{\text{pref}}}} \left[ -\log \sigma \left( \beta \left( \log \frac{\pi_\theta(\tau^w|c)}{\pi_{\text{ref}}(\tau^w|c)} - \log \frac{\pi_\theta(\tau^l|c)}{\pi_{\text{ref}}(\tau^l|c)} \right) \right) \right], \tag{4}$$

where $\beta$ controls the strength of the alignment. This formulation offers several advantages. First, by decoupling hard constraints, it simplifies the optimization landscape and directly enforces absolute feasibility. Second, by using group-wise sampling, it significantly improves data efficiency and learns a more robust model from a richer set of comparisons. Last, the entire process remains offline, ensuring stable and efficient fine-tuning without the complexities of online reward modeling.

### 3.3 Test-time Steerable Generation by Mixture of Preferences

While HGPO provides an efficient way to manage hard constraints, it still requires fine-tuning a policy for a fixed preference trade-off. This static approach has a notable limitation: a model fine-

tuned on a fixed set of linearly scalarized rewards produces solutions corresponding to only *a single point on the Pareto front* of competing objectives. Adapting to different testing requirements needs costly retraining. Drawing inspiration from LLM alignment (Rame et al., 2023), we introduce a test-time alignment method that enables continuous control over the properties of generated trajectories by creating a mixed preference model, allowing users to navigate the entire Pareto front at inference.

**Training Expert Preference Models.** Instead of training a single policy, we first train a set of expert models, each specialized towards a different trade-off between core goals. In our primary case with two objectives (i.e., adversariality and realism), we train two expert policies, $\pi_{\theta_{\text{adv}}}$ and $\pi_{\theta_{\text{real}}}$. Crucially, these are not trained on single objectives in isolation. Instead, they are fine-tuned from the same pretrained model $\pi_{\text{ref}}$ using HGPO described in Section 3.2, but with opposing rewards:

$$
\begin{aligned}
R_{\text{adv-pref}}(\tau) &= w^* R_{\text{adv}}(\tau) - (1 - w^*) P_{\text{real}}(\tau) \quad \text{for training } \pi_{\theta_{\text{adv}}}, \\
R_{\text{real-pref}}(\tau) &= (1 - w^*) R_{\text{adv}}(\tau) - w^* P_{\text{real}}(\tau) \quad \text{for training } \pi_{\theta_{\text{real}}},
\end{aligned}
\tag{5}
$$

where $w^* \in (0.5, 1]$ is a fixed hyperparameter that pushes the experts towards the extremes of the preference space. This preconditions the models to understand the trade-off space, a key difference from methods that train on purely orthogonal rewards. These two expert models effectively anchor the endpoints of the achievable Pareto front for further preference interpolation in the entire space.

**Steerable Generation via Test-Time Weight Interpolation.** At test time, given a user-specified preference weight $\mu \in [0, 1]$ defining the desired trade-off between adversariality and realism, we do not need to retrain. Instead, inspired by Wortsman et al. (2022); Rame et al. (2023), we directly construct a novel, preference-mixed model $\pi_{\theta(\lambda)}$ by linearly interpolating the weights of two experts:

$$
\theta(\lambda) = (1 - \lambda)\theta_{\text{real}} + \lambda\theta_{\text{adv}},
\tag{6}
$$

where $\lambda$ is a coefficient linked to the user preference $\mu$. The mixed model $\pi_{\theta(\lambda)}$ then generates a set of $K$ candidates $\{\tau_k\}_{k=1}^K$. The final output $\tau^*$ is selected by ranking these candidates according to the user's real-time reward $\tau^* = \arg\max_{\tau_k \in \{\tau_k\}} R_\mu(\tau_k)$ with $R_\mu(\tau) = \mu R_{\text{adv}}(\tau) - (1 - \mu) P_{\text{real}}(\tau)$. By varying $\lambda \in [0, 1]$, we can trace a continuous path in weight spaces, which in turn generates a continuous Pareto front of behaviors, from naturalistic to aggressive, without any retraining. This defines a set of (near) Pareto-optimal solutions $\{\pi_{\theta(\lambda)}\}_\lambda$, replacing costly multi-policy strategies. Furthermore, we can extrapolate beyond the interpolation range using preference vectors to generate more extreme and out-of-distribution cases for rigorous stress testing, as detailed in Appendix B.3.

**SAGE in Closed-loop Adversarial Training.** The steerability of SAGE is particularly useful when integrated into a closed-loop adversarial training pipeline to improve an ego policy $\pi_{\text{ego}}$. This process follows a min-max structure where, at each iteration, SAGE generates a new adversary tailored to the latest ego policy's weaknesses using the adversarial policy $\pi_{\theta(\lambda)}$. The ego agent is then updated to handle these new challenges. To stabilize training and prevent the agent from overfitting to adversarial cases while failing in normal driving, we employ a dual-axis curriculum. This curriculum progressively increases the challenge by gradually annealing two parameters: (1) the *intensity* of the scenarios, controlled by the interpolation weight $\lambda$, which shifts from realistic to more aggressive behaviors, and (2) the *frequency* of adversarial encounters. This structured approach ensures the ego agent develops robust, generalizable skills. Further details are provided in Appendix B.1.

### 3.4 Linear Mode Connectivity in Fine-tuned Motion Generation Models

We now provide a theoretical justification for our test-time weight interpolation scheme (Eq. 6), grounding its effectiveness in the phenomenon of *Linear Mode Connectivity* (LMC) (Frankle et al., 2020; Ainsworth et al., 2022). LMC posits that when models are fine-tuned from the same pretrained initialization on related tasks, the resulting solutions often lie within a single, wide, low-loss basin and can be connected by a linear path in the parameter space along which performance remains high. SAGE leverages this property, hypothesizing that this path effectively traces the Pareto front of the underlying multi-objective problem. This geometric property of the reward landscape has two key consequences that we formalize below, with full derivations provided in Appendix C.

**Bounded Suboptimality of Weight Interpolation.** First, LMC implies that the path between $\theta_{\text{adv}}$ and $\theta_{\text{real}}$ likely traverses a region of high reward. This suggests that an interpolation along it, $\theta(\lambda) = (1 - \lambda)\theta_{\text{real}} + \lambda\theta_{\text{adv}}$, can serve as a high-quality approximation for the true optima $\hat{\theta}_\mu$ corresponding to any user preference $\mu \in [0, 1]$. We formalize this by bounding the suboptimality gap.

**Theorem 1** (Suboptimality of Weight Interpolation). *Let the base reward functions $R_{adv}(\theta)$ and $R_{real}(\theta)$ be L-smooth and m-strongly concave in the local region of the fine-tuned optima. Let $\theta_1$ and $\theta_2$ be the optimal parameters for the two expert models (e.g., Eq. 5) trained with mixing weight $\beta \in (0.5, 1]$, and let $\hat{\theta}_\mu$ be the true optimum for a user preference $R_\mu(\theta) = \mu R_{adv}(\theta) + (1 - \mu)R_{real}(\theta), \mu \in [0, 1]$. Then the suboptimality gap of our interpolated model is bounded by:*

$$R_\mu(\hat{\theta}_\mu) - \max_{\lambda \in [0,1]} R_\mu((1 - \lambda)\theta_2 + \lambda\theta_1) \leq C(\mu, \beta, L, m) \cdot \|\theta_2 - \theta_1\|^2, \tag{7}$$

*where $C(\mu, \beta, L, m)$ is a constant dependent on the user preference $\mu$, the expert weight $\beta$, and the geometry of the reward landscape (smoothness $L$ and concavity $m$). See Appendix C.2 for details.*

Theorem 1 provides a quantitative guarantee on the performance of our method. The LMC hypothesis suggests that because both experts are fine-tuned from the same powerful pretrained model $\pi_{\text{ref}}$ on closely related objectives, they likely reside in the same basin of the reward landscape, making their distance small. This controlled suboptimality ensures that the weight interpolation can closely approximate the true Pareto front. This result generalizes the intuition from a simplified quadratic reward setting (Appendix C.1), where we show that there exists a range of user preferences $\mu \in [1 - \beta, \beta]$ defined by the experts' training weights for which it is exactly optimal.

**Superiority of Weight-Space over Output-Space Mixing.** Second, LMC provides a principled reason to prefer mixing model *weights* over mixing their *outputs* (i.e., ensembling trajectories). While ensembling is a common alternative, the concavity of the reward landscape between solutions makes weight-space interpolation particularly effective. We analyze the difference in expected loss (e.g., mean square error (MSE) against an ideal trajectory) between the two strategies.

**Proposition 1** (Advantage of Weight Mixing over Output Ensembling). *Consider the expected loss $L^{weight}(\lambda)$ of a weight-mixed model $\pi_{\theta(\lambda)}$, and an output-mixed model, $L^{ens}(\lambda)$. The difference in their performance is approximated by:*

$$L^{weight}(\lambda) - L^{ens}(\lambda) \approx \underbrace{-\frac{\lambda(1 - \lambda)}{2}\frac{d^2 L^{weight}}{d\lambda^2}}_{\text{Term 1: Benefit from Reward Concavity}} + \underbrace{\frac{\lambda(1 - \lambda)}{2}\mathbb{E}\left[\|\Delta\tau(c)\|_2^2\right]}_{\text{Term 2: Benefit from Output Diversity}}, \tag{8}$$

*where $\Delta\tau(c) = \pi_{\theta_2}(\cdot|c) - \pi_{\theta_1}(\cdot|c)$ is the difference in the mean predicted trajectories for a given context c. The proof is detailed in Appendix C.3.*

Proposition 1 reveals a crucial trade-off. Term 2 is always non-negative and represents the benefit of ensembling diverse outputs. However, Term 1 depends on the geometry of the loss landscape along the linear path connecting $\theta_1$ and $\theta_2$. For weight mixing to be superior, Term 1 must be negative and dominate Term 2. This requires the loss function to be convex along the interpolation path, i.e., $\frac{d^2 L^{weight}}{d\lambda^2} > 0$. This convexity in the loss landscape corresponds to a concave geometry in the reward landscape. Intuitively, a linear path in the weight space traces a curved and concave path in the reward space. This means an interpolated model $\theta(\lambda)$ can achieve a reward greater than the linear average of the experts' rewards. While LMC posits that both experts reside in a common low-loss basin, this does not imply zero curvature. Instead, the path connecting them acts as a high-reward ridge (Fig. 6(c)). The concavity of this ridge provides the necessary positive curvature in the loss, making Term 1 negative and dominant. This phenomenon is empirically validated by the concave reward curves shown in Fig. 6(d), which lie strictly above the linear interpolation line. By operating in the weight space of a well-structured model, our method exploits this geometric property to generate high-quality behaviors learned by a coherent model.

Together, these results suggest that the geometric properties of the fine-tuning landscape make linear weight interpolation not merely a heuristic but a principled method for controllable generation.

## 4 EXPERIMENTS

We conduct comprehensive experiments to validate SAGE. Our evaluation is designed to answer three research questions: (1) In an open-loop setting, how does it compare against SOTA baselines in generating challenging yet realistic scenarios? (2) In a closed-loop setting, does training with these scenarios translate into superior driving policies? (3) How can the steerable generation capability be comprehended in a principled way? (4) Do the key designs, such as HGPO, have positive impacts

on the performance? All experiments are conducted in the MetaDrive (Li et al., 2022) simulator using real-world scenarios from the Waymo Open Motion Dataset (Ettinger et al., 2021). Detailed setups and supplementary results are provided in Appendix B and D.

## 4.1 BENCHMARKING SAFETY-CRITICAL SCENARIO GENERATION METHODS

Table 1: Evaluation of adversarial generation methods against the **Replay** policy. Higher is better for Attack Success Rate and Adversarial Reward (↑), while lower is better for penalty metrics (↓). WD denotes Wasserstein distance. SAGE is trained using separate adversarial weights $w_{adv}$.

| Methods | Attack Succ. Rate ↑ | Adv. Reward ↑ | Real. Pen. ↓ Behav. | Kine. | Map Comp. ↓ Crash Obj. | Cross Line | Dist. Diff. (WD) Accel. | Vel. | Yaw |
|---|---|---|---|---|---|---|---|---|---|
| Rule | 100.00% | 5.048 | 2.798 | 5.614 | 1.734 | 7.724 | 2.080 | 8.546 | 0.204 |
| CAT (Zhang et al., 2023) | 94.85% | 3.961 | 8.941 | 3.143 | 2.466 | 9.078 | 1.556 | 7.233 | 0.225 |
| KING (Hanselmann et al., 2022) | 40.85% | 2.243 | 5.883 | 3.434 | 3.126 | 6.056 | 0.972 | 255.5 | 0.098 |
| AdvTrajOpt (Zhang et al., 2022) | 70.46% | 2.652 | 4.500 | 2.775 | 2.547 | 10.16 | 1.754 | 6.177 | 0.268 |
| SEAL (Stoler et al., 2025) | 63.93% | 1.269 | 3.017 | 2.423 | 2.732 | 11.612 | 1.544 | 6.959 | 0.202 |
| GOOSE (Ransiek et al., 2024a) | 36.07% | 2.378 | 4.718 | 21.32 | 4.426 | 14.48 | 6.368 | 8.286 | 0.154 |
| SAGE ($w_{adv} = 0.0$) | 16.26% | 1.065 | 0.332 | 1.998 | 0.677 | 0.948 | 1.459 | 9.313 | 0.054 |
| SAGE ($w_{adv} = 0.5$) | 50.41% | 2.523 | 0.483 | 2.064 | 0.755 | 0.949 | 1.521 | 8.471 | 0.079 |
| SAGE ($w_{adv} = 1.0$) | 76.15% | 4.121 | 1.429 | 2.479 | 0.731 | 1.084 | 2.098 | 8.088 | 0.184 |

Table 2: Evaluation of adversarial generation methods against the **RL** policy.

| Methods | Attack Succ. Rate ↑ | Adv. Reward ↑ | Real. Pen. ↓ Behav. | Kine. | Map Comp. ↓ Crash Obj. | Cross Line | Dist. Diff. (WD) Accel. | Vel. | Yaw |
|---|---|---|---|---|---|---|---|---|---|
| Rule | 65.57% | 2.761 | 2.180 | 113.7 | 1.803 | 6.148 | 10.85 | 13.47 | 0.336 |
| CAT (Zhang et al., 2023) | 30.33% | 1.319 | 8.191 | 3.039 | 2.623 | 6.967 | 1.539 | 8.877 | 0.187 |
| KING (Hanselmann et al., 2022) | 19.14% | 1.148 | 2.041 | 2.596 | 3.114 | 5.857 | 0.983 | 259.1 | 0.097 |
| AdvTrajOpt (Zhang et al., 2022) | 19.40% | 0.992 | 4.542 | 2.779 | 2.459 | 9.973 | 1.749 | 6.187 | 0.269 |
| SEAL (Stoler et al., 2025) | 31.40% | 0.752 | 5.871 | 2.684 | 3.030 | 11.98 | 1.563 | 8.267 | 0.267 |
| GOOSE (Ransiek et al., 2024a) | 12.46% | 0.667 | 4.369 | 15.47 | 3.507 | 11.45 | 4.662 | 8.070 | 0.152 |
| SAGE ($w_{adv} = 0.0$) | 11.20% | 0.722 | 0.332 | 2.000 | 0.738 | 0.956 | 1.456 | 9.344 | 0.055 |
| SAGE ($w_{adv} = 0.5$) | 13.66% | 0.819 | 0.496 | 2.066 | 0.820 | 0.820 | 1.515 | 8.475 | 0.080 |
| SAGE ($w_{adv} = 1.0$) | 28.42% | 1.400 | 1.468 | 2.496 | 0.792 | 1.366 | 2.098 | 8.114 | 0.188 |

We first benchmark SAGE in an open-loop setting against SOTA adversarial baselines. Tabs. 1 and 2 suggest that SAGE shows a superior balance between adversariality and realism. When configured for maximum adversariality, SAGE achieves a high attack success rate and adversarial reward, competitive with the strongest baselines. More importantly, it maintains significantly lower realism and map compliance penalties across the board. For instance, against the Replay policy, SAGE reduces map violations (e.g., Cross Line penalty of 1.084) by over 85% compared to Rule-based (7.724) and CAT (9.078), while also achieving the highest behavioral realism. This highlights the effectiveness of treating map compliance as a hard constraint rather than a soft penalty. A noteworthy observation arises from the distributional distance metrics. While SAGE excels at maintaining low realism penalties for individual trajectories, it can produce a higher distributional distance than other methods, though within a reasonable range. This suggests that it achieves adversariality not by generating physically implausible actions, but by discovering coherent and statistically rare long-tail scenarios that challenge the ego agent precisely because they are underrepresented in the training distribution.

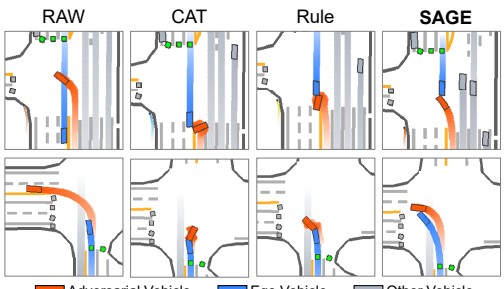

Figure 2: Behavioral realism comparison. Adversarial generation against the Replay policy.

Qualitative results in Fig. 2 (more examples in section D.5.2) confirm these findings. While baselines often produce physically awkward or rule-violating trajectories, SAGE generates challenging yet plausible maneuvers. This ability is crucial for meaningful safety validation and robust agent training. Similar trends are observed when testing against a stronger RL policy (Tab. 2), confirming the robustness of SAGE. Quantitative results against other reactive policies are provided in Appendix D.1 and visualized in Fig. 12. Fig. 3 shows that SAGE generates diverse types of plausible adversarial behaviors for effective stress tests and training.

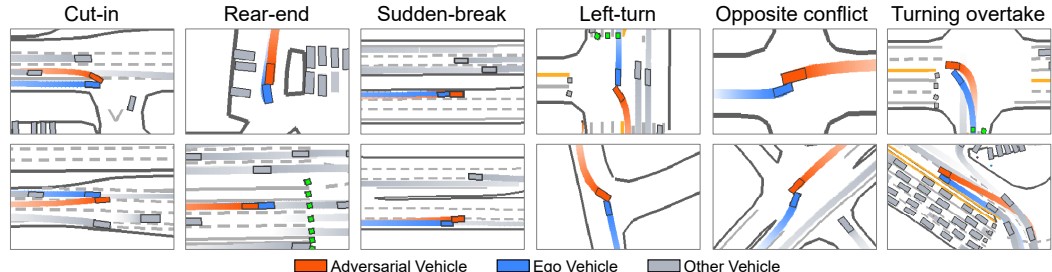

Figure 3: SAGE generates diverse types of meaningful adversarial behaviors (Replay policy).

## 4.2 TEST-TIME STEERABLE GENERATION VIA PREFERENCE ALIGNMENT

A key contribution of SAGE is the ability to steer scenarios at test time without retraining. Fig. 4(a) illustrates this capability by comparing the Pareto fronts of different mixing strategies. Our proposed weight mixing traces a superior Pareto front, achieving a better realism score for any given level of adversariality compared to logit-space or trajectory-space mixing (see section B.3), which empirically validates Proposition 1. The continuous and monotonic curves in Fig. 4(b) further indicate the fine-grained controllability: as $w_{adv}$ increases, the collision rate smoothly rises, with a corresponding increase in the distributional distance from naturalistic data, indicating a controlled trade-off. The performance of SAGE at different weight configurations, shown in Tabs. 1 and 2, further quantifies this smooth transition. This controlability is visualized in Fig. 5: the generated trajectory transitions smoothly from a compliant lane-following maneuver to a highly aggressive cut-in or sudden brake.

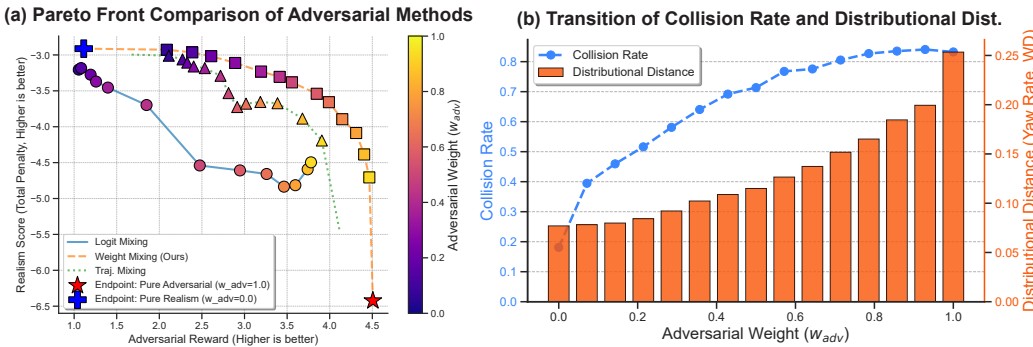

Figure 4: Pareto front and continuous performance transition at test time. (a) We compare the trade-off curves for different model merging strategies in terms of their adversariality and realism. (b) SAGE achieves smooth and continuous outcome control by varying the adversarial weight.

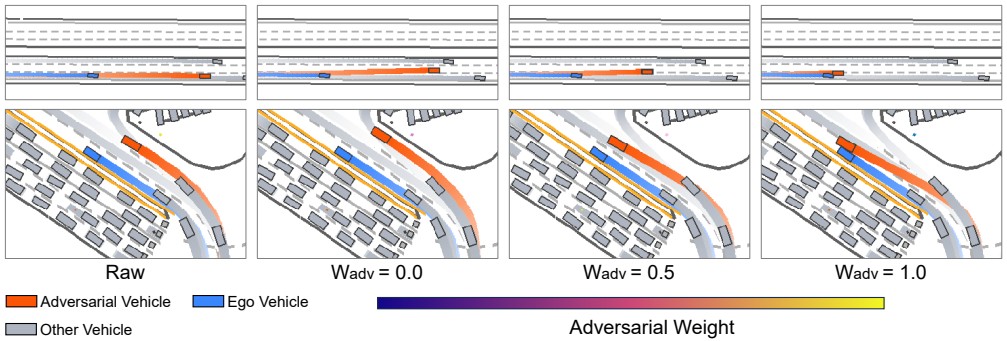

Figure 5: More aggressive behaviors are generated from SAGE by increasing $w_{adv}$ from 0 to 1 (see Appendix D.5.3 for more examples). Adversarial generation against the Replay policy.

## 4.3 DECIPHERING THE PREFERENCE-ALIGNED SPACES OF FINE-TUNED MOTION MODELS

We empirically validate the theoretical understanding of weight interpolation and LMC. Since we fine-tune a well-pretrained model, it is plausible that the learned experts remain in a local region of

the weight space with favorable geometry. Fig. 6 provides strong evidence. Fig. 6(a) visualizes the preference vectors in a PCA-projected space, showing they are distinct and non-collinear, capturing unique aspects of the desired behaviors. The interpolated points lie between the line segment connecting two vectors, while the extrapolation points are scattered in space, which aligns with our understanding. In addition, the angle between the directions of two vectors lies within $(0°, 90°)$, which is reasonable to cover a wide Pareto front. As we interpolate between $\theta_{adv}$ and $\theta_{real}$, Fig. 6(b) shows a smooth, monotonic transform in both the L2 norm of the preference vector and its cosine similarity to the target adversarial vector, indicating a well-behaved and controllable path.

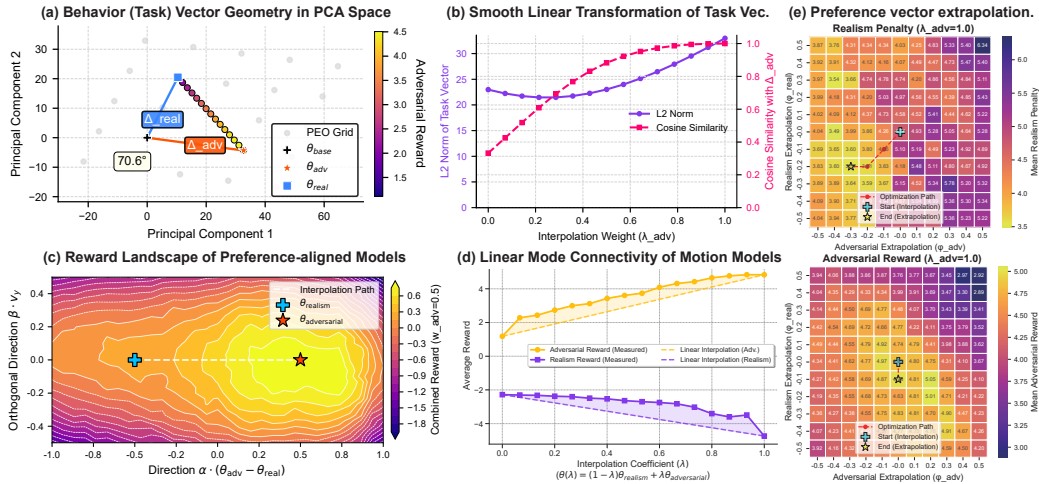

Figure 6: Empirical evidence for LMC and preference vector manipulation in the weight space. (a) PCA plot shows the preference vectors ($\Delta_{adv}$, $\Delta_{real}$, see Eq. 14) forming a well-defined space for interpolation. (b) The interpolated vector's properties transform smoothly, indicating a controllable path. (c) The reward landscape reveals a high-reward ridge connecting two experts, which is a key property of LMC. (d) The rewards exhibit concavity along the path, justifying the superiority of weight mixing. (e) Preference vector extrapolation improves performance beyond the convex hull.

Crucially, Fig. 6(c) shows the reward landscape along a 2D plane defined by the two experts. The linear path connecting them traverses a high-reward plateau, consistent with the LMC prediction of a flat loss basin. This flatness and curvature are key to the success of weight interpolation, making the suboptimality gap well-bounded. Fig. 6(d) provides direct evidence: the measured rewards of interpolated models closely track or even exceed the rewards predicted by a linear interpolation of the rewards of endpoints. This confirms that interpolating in weight space is more effective than simply expecting a linear combination of outcomes, supporting Proposition 1. Furthermore, Fig. 6(e) demonstrates that preference vector extrapolation (see Appendix B.3) successfully extends generation beyond the convex hull of the experts, yielding a path that achieves even higher rewards and lower penalties than the solution on the original Pareto front, thereby enabling the creation of out-of-distribution scenarios for rigorous testing. More results are provided in Appendix D.4.

## 4.4 CLOSED-LOOP ADVERSARIAL TRAINING FOR IMPROVED DRIVING POLICY

Table 3: Evaluation of trained RL policies in the generated (adversarial, $w_{adv} = 1.0$) environments.

| Methods | Reward ↑ | Cost ↓ | Compl. ↑ | Coll. ↓ | Ave. Speed ↑ | Ave. Jerk ↓ |
|---|---|---|---|---|---|---|
| SAGE | **45.14 ± 3.27** | **0.61 ± 0.04** | **0.69 ± 0.03** | **0.31 ± 0.02** | **8.98 ± 0.02** | **28.85 ± 0.68** |
| CAT | 37.70 ± 1.53 | 0.70 ± 0.04 | 0.58 ± 0.02 | 0.37 ± 0.04 | 6.85 ± 0.03 | 31.83 ± 1.10 |
| Replay (No Adv) | 41.32 ± 3.21 | 0.68 ± 0.04 | 0.62 ± 0.04 | 0.44 ± 0.06 | 8.77 ± 0.01 | 30.42 ± 1.12 |
| Rule-based Adv | 32.99 ± 4.89 | 0.72 ± 0.03 | 0.50 ± 0.04 | 0.33 ± 0.02 | 5.99 ± 0.04 | 30.51 ± 0.99 |

To assess the downstream utility of SAGE, we integrate it into a closed-loop RL training pipeline. As shown in Fig. 9 (in Appendix D.2) and Tab. 3, an ego agent trained with SAGE consistently outperforms baselines across all metrics: it achieves higher rewards, lower costs, greater route completion, and a lower crash rate. Beyond the safety metrics, it also achieves better efficiency and comfort performances. Moreover, SAGE alleviates catastrophic forgetting as evidenced by Tab. 10 (in Ap-

pendix D.2). The policy trained with SAGE also achieves the best overall performance in log-replay scenarios. This justifies the effectiveness of our dual-axis curriculum (see Appendix B.1).

To further ensure a fair evaluation and assess the risk of overfitting, we conducted a cross-evaluation of the trained RL policies. In this setup, we benchmarked every trained agent on held-out test sets generated by the CAT and Rule-based methods. The results in Tab. 4 provide several key insights. First, we observe that an agent tends to achieve the best performance on the adversarial metric when evaluated on scenarios created by its own training generator. For example, the CAT-trained agent has the lowest collision rate in CAT-generated scenarios. This validates the importance of this cross-evaluation. Second, and more importantly, the SAGE-trained agent demonstrates superior generalizability across all environments. Its safety performance remains highly competitive, ranking a close second. Notably, the SAGE-trained agent achieves the highest route completion rate across all adversarial test sets, demonstrating that it learns to handle challenges more effectively without compromising its primary driving objective. This robustness across diverse adversarial distributions suggests that the curriculum-based training with SAGE, which exposes the agent to a wide spectrum of adversarial intensities, prevents overfitting to a specific attack pattern and enables a more generalizable and practical driving policy. In contrast, agents trained on more static adversarial distributions (CAT, Rule-based) show a more significant performance drop when tested out-of-distribution.

Table 4: Evaluation of Trained RL Policies in Different Adversarial Environments.

| Training Methods | CAT-generated scenarios | | Rule-generated scenarios | |
|---|---|---|---|---|
| | Compl. ↑ | Coll. ↓ | Compl. ↑ | Coll. ↓ |
| SAGE | $\mathbf{0.678 \pm 0.034}$ | $\underline{0.307 \pm 0.037}$ | $\mathbf{0.655 \pm 0.037}$ | $\underline{0.334 \pm 0.038}$ |
| CAT | $0.660 \pm 0.039$ | $\mathbf{0.298 \pm 0.050}$ | $0.633 \pm 0.043$ | $0.389 \pm 0.044$ |
| Replay (No Adv) | $\underline{0.665 \pm 0.042}$ | $0.454 \pm 0.051$ | $\underline{0.642 \pm 0.027}$ | $0.461 \pm 0.046$ |
| Rule-based Adv | $0.569 \pm 0.050$ | $0.348 \pm 0.028$ | $0.556 \pm 0.059$ | $\mathbf{0.293 \pm 0.043}$ |

## 4.5 ABLATION STUDY

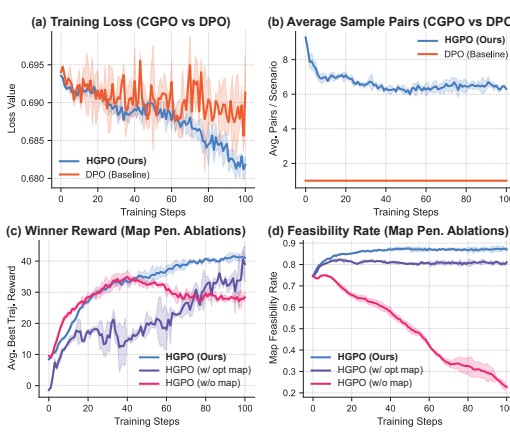

Figure 7: Ablation studies.

We conduct ablation studies to validate key design choices. Fig. 7(b) highlights the superior sample efficiency of HGPO over DPO. By constructing multiple preference pairs from a group of samples, HGPO utilizes much more information per scenario, leading to a more stable training loss and faster convergence (Fig. 7(a)). We also validate our approach of treating map compliance as a precondition. Figs. 7(c,d) show that this maintains a high map feasibility rate of nearly 90% and a higher reward throughout training. In contrast, removing the map constraint leads to a collapse in feasibility, as the model learns to exploit off-road shortcuts. Using a weighted penalty improves feasibility but remains suboptimal compared to our HGPO. This confirms that decoupling hard constraints from soft preferences is critical for generating valid trajectories. See more results in Appendix D.3.

## 5 CONCLUSION

This work introduced SAGE, a new paradigm that reframes adversarial scenario generation as a test-time preference alignment problem. By fine-tuning experts on opposing preferences and interpolating their weights at inference, SAGE enables fine-grained, steerable control over the trade-off between adversariality and realism. SAGE not only yields a superior Pareto front in open-loop evaluations but also results in improved driving policies in closed-loop training. Our work establishes a principled foundation for steerable generation and opens several future avenues. First, the current framework could be extended to incorporate a richer set of objectives, such as scenario novelty or complexity. Second, more advanced model merging techniques could potentially uncover better paths in the parameter space. Finally, an automated curriculum where the model is dynamically adapted based on the agent's learning progress could lead to more intelligent adversarial training.

ACKNOWLEDGMENT

This research was in part supported by the National Natural Science Foundation of China (Project No. 524B2164, 52232015, and 52125208); the Research Grants Council of the Hong Kong Special Administrative Region, China (Project No. PolyU/15206322 and PolyU/15227424); and Otto Poon Charitable Foundation Smart Cities Research Institute, The Hong Kong Polytechnic University (CD06).

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

## DECLARATION OF THE USE OF LLMS IN THE WRITING PROCESS

During the preparation of this work, the authors used LLMs to improve language clarity and readability. The authors reviewed the content as needed and take full responsibility for the content.

## APPENDIX

This technical appendix provides comprehensive supplementary materials to the main paper. It is structured as follows: Section A offers a detailed review of essential background concepts. Section B outlines the full implementation details of our experiments. Section C presents a rigorous theoretical analysis. Finally, Section D provides extensive supplementary experimental results.

## APPENDIX CONTENTS

# A    BACKGROUND AND PRELIMINARY

## A.1    PRELIMINARY ABOUT ADVERSARIAL SCENARIO GENERATION

This section first introduces concepts and notation for representing traffic scenarios and agent behaviors. We then formalize the closed-loop adversarial training framework discussed in section 3.3.

### A.1.1    TRAFFIC SCENARIO REPRESENTATION

We model a traffic scenario $\mathcal{S}$ as a collection of interacting agents operating within a static environment. The environment is defined by a high-definition map $\mathcal{M}$, which contains geometric and semantic information. The set of agents in the scenario is denoted by $\mathcal{A} = \{a_0, a_1, \ldots, a_{N-1}\}$, where $a_0$ is the AD agent being tested (ego), and $\{a_i\}_{i=1}^{N-1}$ are the other traffic participants, referred to as other agents. The state of an agent $a_i$ at timestep $t$ is represented by $s_t^i = (x_t^i, y_t^i, \theta_t^i, v_t^i)$, which includes its 2D position, heading, and speed. The trajectory of an agent $a_i$ over a time horizon $T$ is a sequence of its states, $\tau^i = \{s_0^i, s_1^i, \ldots, s_T^i\}$. The collective behavior of all agents in the scenario is described by the set of all trajectories $\mathcal{T} = \{\tau^i\}_{i=0}^{N-1}$. Thus, a complete scenario can be defined as a tuple $\mathcal{S} = (\mathcal{M}, \{\tau^{\text{ego}}, \tau^1, \ldots, \tau^{N-1}\})$. In this sense, the scenario context $c$ refers to the historical scenario or the initial scenario to be predicted. Here, scenarios are sourced from real-world driving logs. In this work, we consider the ego agent $a_0$ is controlled by an end-to-end driving policy $\pi_{\text{ego}}$, which directly maps a sequence of observations $o_t$ to a future plan or a low-level action.

### A.1.2    CLOSED-LOOP ADVERSARIAL TRAINING FOR SAFE DRIVING

To enhance the robustness of an ego agent's driving policy, we leverage the paradigm of adversarial training (Ma et al., 2018; Wachi, 2019). This approach goes beyond training on a fixed dataset of benign scenarios by dynamically generating challenging scenarios that are specifically tailored to the policy's current weaknesses. In the context of reinforcement learning (RL), the standard objective for the ego agent is to learn a policy $\pi_{\text{ego}}$ that maximizes the expected cumulative reward $\mathcal{J}$ within a given environment, characterized by a transition function $f$:

$$\max_{\pi_{\text{ego}}} \mathcal{J}(\pi_{\text{ego}}, f) = \max_{\pi_{\text{ego}}} \mathbb{E}_{\tau \sim \pi_{\text{ego}}, f} \left[ \sum_{t=0}^{T} R(s_t, a_t) \right], \tag{9}$$

where $R$ is the environment reward function and the trajectory $\tau$ is sampled by executing policy $\pi_{\text{ego}}$ in the environment $f$. Adversarial training extends this into a min-max optimization problem. The adversary actively modifies the environment's transition function to $f_{\text{adv}}$ to minimize the ego agent's reward, while the ego agent simultaneously learns to maximize its reward in this worst-case environment:

$$\max_{\pi_{\text{ego}}} \min_{f_{\text{adv}} \in \mathcal{F}} \mathcal{J}(\pi_{\text{ego}}, f_{\text{adv}}), \tag{10}$$

where $\mathcal{F}$ is the space of feasible adversarial environments. The inner loop, $\min_{f_{\text{adv}}}$, corresponds to the adversarial scenario generation process, and the outer loop $\max_{\pi_{\text{ego}}}$ is the policy optimization step, where $\pi_{\text{ego}}$ is updated using an RL algorithm to overcome the challenges posed by $f_{\text{adv}}$. This formulation can lead to a closed-loop and iterative training pipeline (Zhang et al., 2023). The policy is continuously improved by training on adversarial scenarios generated on-the-fly against its most recent version. This co-evolution of the agent and environment forces the policy to learn generalizable and robust driving skills, rather than overfitting to a static set of scenarios (Anzalone et al., 2022). The efficiency and quality of the generator are therefore critical to the success of the pipeline.

## A.2    DETAILED RELATED WORK ON ADVERSARIAL SCENARIO GENERATION

The generation of safety-critical scenarios is crucial for testing and enhancing the robustness of autonomous driving (AD) systems (Xu et al., 2022; Ding et al., 2023; Nie et al., 2025). In recent years, several distinct paradigms have emerged to automate this process, each with its own strengths and limitations. In this section, we provide a comprehensive review of these paradigms, focusing on *Reinforcement Learning-based, Diffusion-based, and Optimization/Sampling-based methods*. We position our work within this landscape, addressing key unresolved challenges of existing paradigms.

**RL-based Generation.** RL has been widely adopted to generate adversarial scenarios by training an adversary agent to maximize a reward function correlated with the failure of the ego policy (Wachi, 2019; Kuutti et al., 2020; Feng et al., 2021; 2023; Chen et al., 2024; Ransiek et al., 2024a; Qiu et al., 2025). In this paradigm, the ego agent is treated as part of the environment, and the adversary learns complex, non-intuitive behaviors to exploit its weaknesses. A significant advantage of RL-based methods is their inherent suitability for closed-loop training, enabling the ego agent to be hardened against an adapting adversary (Kuutti et al., 2020; Ransiek et al., 2024a). However, this approach suffers from two major drawbacks. First, *limited transferability*: the adversarial policy is tightly coupled with the specific ego agent it was trained against. A scenario generated for one AD stack may not be challenging for another, necessitating costly retraining for each new system under test (Ransiek et al., 2024b; Ding et al., 2020). This makes it ill-suited for large-scale, general-purpose testing. Second, *unrealistic behaviors*: RL agents are prone to reward hacking, where they discover unrealistic or physically implausible strategies to maximize the reward, such as erratic movements or violating basic traffic rules, unless the reward function is meticulously engineered (Kuutti et al., 2020; Qiu et al., 2025).

**Diffusion-based Generation.** Leveraging their success in high-fidelity image and video synthesis, diffusion models have recently been applied to traffic scenario generation (Zhong et al., 2023; Xie et al., 2024; Xu et al., 2025). These methods learn the distribution of real-world traffic data and can generate realistic, diverse, and controllable scenarios. By incorporating controllability during the reverse diffusion process, these models can be steered to produce safety-critical events, such as collisions or near-misses, based on textual descriptions (Zhong et al., 2023) or predefined objective functions (Xie et al., 2024; Chang et al., 2024; Xu et al., 2025). This controllability is primarily achieved with guidance and conditioning. Diffusion guidance is strong in control but slow, and the other is statistically true but weak in adjustability. This paradigm excels in generating realistic and highly controllable scenarios. Moreover, these methods are generally policy-agnostic, as the generative model is trained on real-world data rather than against a specific ego agent, leading to better transferability. The primary limitation of diffusion-based methods is their *significant computational overhead*. The iterative denoising process is computationally expensive and slow, which makes it currently unsuitable for closed-loop adversarial training, where scenarios need to be generated on-the-fly to continuously challenge and improve the ego agent's policy.

**Optimization and Sampling-based Generation.** This category of methods focuses on perturbing existing real-world scenarios or sampling from a learned prior to generate adversarial variants (see section A.1). The search for adversarial parameters is often performed in different spaces, such as the parameter space of vehicle kinematics (Hanselmann et al., 2022; Cao et al., 2022; Mei et al., 2024), the latent space of a generative model such as a VAE (Rempe et al., 2022), or directly in the trajectory space (Wang et al., 2021; Zhang et al., 2023; Mei et al., 2025a;b). Especially, CAT (Zhang et al., 2023) falls into this category by modeling adversarial traffic as a probabilistic factorization and sampling trajectories with the highest collision probability. Their main advantage lies in their computational efficiency, which makes them viable for closed-loop adversarial training frameworks. They are also typically policy-agnostic and thus highly transferable. Nevertheless, these methods often struggle with *controllability and realism*. The optimization or sampling process, if not properly constrained, can push trajectories into unrealistic regimes. As noted in our motivation, methods like CAT can produce overly aggressive yet physically implausible trajectories, such as vehicles spinning in place, which limits their utility for training robust and realistic AD policies.

**Summary.** Existing paradigms for adversarial scenario generation present a fundamental dilemma between *generalizability*, *efficiency* and *controllability*. Reinforcement Learning (RL) methods, for instance, can discover complex adversarial strategies but are often coupled to a specific AD policy and prone to reward hacking, leading to unrealistic behaviors (Feng et al., 2023; Qiu et al., 2025). Conversely, diffusion-based generative models offer superior generalizability and fine-grained control (Zhong et al., 2023; Xie et al., 2024; Xu et al., 2025), yet their significant computational overhead renders them impractical for on-the-fly generation required in closed-loop settings. A promising avenue lies in optimization or sampling-based methods, which operate on perturbing a learned naturalistic driving prior (Wang et al., 2021; Hanselmann et al., 2022; Zhang et al., 2023). They are efficient and generalizable for closed-loop training, but achieving meaningful control over the generation process remains a challenge. A new paradigm that can inherit the merits of both is desired.

## B DETAILED IMPLEMENTATION

### B.1 EXPERIMENTAL SETUPS

This section provides a detailed description of our experimental setups, including the dataset, simulation environment, backbone model, baseline methods, and our fine-tuning and RL training procedures, to ensure the reproducibility of our results.

**Datasets.** Our experiments are conducted using the Waymo Open Motion Dataset (WOMD) (Ettinger et al., 2021), which is consistent with prior works (Zhang et al., 2023; Stoler et al., 2025). WOMD provides a large collection of real-world, complex, and interactive traffic scenarios. From the standard set of 500 scenarios used in Zhang et al. (2023), we select a filtered subset of 370 scenarios for our experiments. This subset is curated by excluding scenarios where the adversarial agent's initial trajectory is either too short for meaningful interaction or already violates map boundaries, ensuring a high-quality and challenging cases for adversarial generation. This dataset is utilized for both generating adversarial scenarios and for fine-tuning our generative models.

**Environment.** All our experiments are performed in the `MetaDrive` (Li et al., 2022) simulator, a lightweight and efficient simulation platform for AD research. Following the setup in Zhang et al. (2023), we use `MetaDrive` to reconstruct scenarios from the WOMD dataset. This allows for closed-loop evaluation, where the ego agent can react dynamically to the behavior of the adversarially controlled traffic participants. The simulation runs at a frequency of 10Hz. To ensure a fair and direct comparison with prior work, we adopt the specific environmental and agent training parameters from Zhang et al. (2023). The key hyperparameters for the scenario generation process used for evaluation are detailed in Tab. 5.

Table 5: Hyperparameters for the open-loop scenario generation evaluation.

| Hyper-parameter | Value |
|---|---|
| Scenario Horizon $T$ | 9s |
| History Horizon $t$ | 1s |
| # of OV candidates $M$ | 32 |
| # of EV candidates $N$ | 1 |
| Penalty Factor $\alpha$ | 0.99 |
| Policy Training Steps | $1 \times 10^6$ |

Table 6: Hyperparameters for training the TD3 RL agent.

| Hyper-parameter | Value |
|---|---|
| Discounted Factor $\gamma$ | 0.99 |
| Train Batch Size | 256 |
| Critic Learning Rate | $3 \times 10^{-4}$ |
| Actor Learning Rate | $3 \times 10^{-4}$ |
| Policy Delay | 2 |
| Target Network $\tau$ | 0.005 |

**Pretrained Backbone Model.** The foundation of our scenario generation framework is a probabilistic motion generation model. To ensure a fair comparison with the baselines, we employ `DenseTNT` (Gu et al., 2021), the same backbone model used in CAT (Zhang et al., 2023). `DenseTNT` is one of the state-of-the-art goal-conditioned motion prediction model that generates a diverse set of future trajectory proposals along with their corresponding confidence scores. The pretraining task is to train the model to predict future trajectories using historical observations and scene context, supervised by human experts. This follows the standard imitation learning (Gu et al., 2021). This pretrained model serves as the learned traffic prior that SAGE fine-tunes and that all baselines (except for the rule-based one) leverage to generate adversarial behaviors. To enable end-to-end preference optimization, we modify the decoding procedure of `DenseTNT` by outputting the top-k candidates directly, bypassing the non-differentiable non-maximum suppression (NMS) sampling process. Furthermore, our backbone-agnostic framework is generalizable and applicable to any probabilistic motion forecasting (generation) backbone that can generate multimodal trajectories with their probabilities (predicted logits). We take `DenseTNT` as an example in the experiments. We adopt the publicly released pretrained model checkpoint from the CAT's repository for experiments to ensure a fair comparison.

**Baselines.** We compare SAGE against a comprehensive set of strong baselines from recent literature on safety-critical (adversarial) scenario generation.

- **CAT** (Zhang et al., 2023): A state-of-the-art method that generates adversarial scenarios by resampling trajectories from a learned traffic prior (`DenseTNT`) to maximize a collision-based objective in a closed-loop setting. Following the original implementation, CAT employs a heuristic objective. It generates a set of candidate trajectories $\mathcal{T}$ from the `DenseTNT` prior and selects the adversarial trajectory $\tau_{\text{adv}}^*$ that maximizes the collision probability with the ego vehicle's planned path $\tau_{\text{ego}}$: $\tau_{\text{adv}}^* = \arg\max_{\tau_i \in \mathcal{T}} P(\text{coll}(\tau_i, \tau_{\text{ego}}))$.

- **SEAL** (Stoler et al., 2025): A recent approach that employs a skill-enabled adversary. It combines a learned objective function with a reactive, skill-based policy to generate more realistic and human-like adversarial behaviors than CAT. We utilize the official open-sourced implementation of SEAL. Its objective is based on a learned scoring network, $\pi_{\text{score}}$, which is trained to predict two criticality metrics: collision closeness ($f_{\text{coll}}$) and ego behavior deviation ($f_{\text{diff}}$). The final adversarial objective is to select trajectories that maximize the sum of these predicted scores: $\max \mathbb{E}_{\tau_{\text{adv}}}[f_{\text{coll}}(\tau_{\text{adv}}, \tau_{\text{ego}}) + f_{\text{diff}}(\tau_{\text{adv}}, \tau_{\text{ego}})]$. This learned objective, combined with a skill-based policy, aims to produce more nuanced and realistic adversarial behaviors.

- **KING** (Hanselmann et al., 2022): A gradient-based approach that perturbs adversarial trajectories by backpropagating through a differentiable kinematic bicycle model to induce collisions. For a fair comparison, we re-implemented its core mechanism within our experimental environment, as detailed in our supplementary code. The optimization is performed for 100 steps with a learning rate of 0.005 using an Adam optimizer.

- **AdvTrajOpt** (Zhang et al., 2022): This method models adversarial scenario generation as a trajectory optimization problem and solves it using Projected Gradient Descent (PGD). We adapted its PGD-based trajectory perturbation logic into our environment, ensuring it operates on the same initial conditions and interacts with the same ego agent. Our implementation uses 50 PGD steps with a learning rate of 0.2. For both **KING** and **AdvTrajOpt**, we use the same multi-objective reward function as SAGE, defined in Eq. (2) to ensure a fair comparison. To align with their goal of generating highly challenging scenarios, we set the weights to a fixed point on the Pareto front that heavily prioritizes adversariality over realism, with $w_{\text{adv}} = 0.9$ and $w_{\text{real}} = 0.1$. This provides a direct comparison of the generation quality under a shared, aggressive objective.

- **GOOSE** (Ransiek et al., 2024a): A goal-conditioned RL framework for safety-critical scenario generation. GOOSE models the entire trajectory using Non-Uniform Rational B-Splines (NURBS), and the RL agent learns to manipulate the NURBS control points to achieve predefined adversarial goals. For our experiments, we use the pretrained model checkpoint released by SEAL's repository for the WOMD dataset to ensure a fair comparison.

- **Rule-based**: A heuristic baseline consistent with the one described in Appendix F of Zhang et al. (2023). This method generates an adversarial path by selecting waypoints from the ego vehicle's future path, mixing them with the adversary's original waypoints, and fitting a smooth Bezier curve through them to create an aggressive cut-in or blocking maneuver.

To justify the superiority of the test-time alignment scheme proposed in section 3.3, we further compare two different alignment methods as described below:

- **Trajectory mixing**: The output is generated by directly mixing trajectories from the two expert models: $\tau(\lambda) = (1-\lambda)\tau_{\text{real}} + \lambda\tau_{\text{adv}}$. This is an instantiation of the output ensembling method discussed in Proposition 1.

- **Multi-objective decoding** (Shi et al., 2024): Inspired by Shi et al. (2024), we mix the predicted logits of two experts. Following the architecture of DenseTNT (Gu et al., 2021), we use the logits of all candidate goal points as the approximation of trajectory probability. This gives a mixed goal score $g(c; \lambda) = (1 - \lambda)g(c; \theta_{\text{real}}) + \lambda g(c; \lambda_{\text{adv}})$. Then, the final trajectory is generated by decoding $g(c; \lambda)$ with the highest score value using the pretrained trajectory decoder. This scheme mimics the token prediction stage of LLMs.

Compared to our weight-space mixing, both of them are action-space mixing strategies, thus serving as direct counterparts to justify Proposition 1.

**HGPO Fine-tuning.** Our preference fine-tuning approach has several hyperparameters. For the number of trajectory candidates, we let the policy model generate a group of $M = 32$ trajectories used for reward computing. A reward difference exceeding a margin of $\delta_m = 0.2$ is regarded as a valid signal. We sample up to 8 pairs per scenario for the group-based learning process. The temperature parameter is set to $\beta = 0.05$ to regularize the alignment. We use the AdamW optimizer with a learning rate of $1 \times 10^{-5}$. We train two specialized models by adjusting the weights in the preference reward calculation:

- **Adversarial Model**: We set the adversarial weight $w_{\text{adv}} = 0.9$ and the realism weight $w_{\text{real}} = 0.1$. This configuration strongly encourages the model to prioritize generating trajectories that result in safety-critical interactions.
- **Realism Model**: We set $w_{\text{adv}} = 0.1$ and $w_{\text{real}} = 0.9$. This configuration prioritizes kinematic and behavioral plausibility, encouraging the model to generate more human-like and realistic, albeit potentially less aggressive trajectories.

The models are trained for 200 epochs on the 370 selected training scenarios. Hyperparameters for HGPO are provided in Tab. 7.

**Details of the System Under Test.** To thoroughly evaluate the generated adversarial scenarios, we test them against a diverse set of ego agent policies, representing different levels of driving capability and design philosophies of AD systems.

- **Replay Policy**: This is the most basic agent, which deterministically replays the ground-truth trajectory of the ego vehicle as recorded in the WOMD dataset. It serves as a non-reactive baseline to measure the raw difficulty of a scenario.
- **Intelligent Driver Model (IDM)**: A standard, rule-based car-following model widely used in traffic simulation. The IDM agent maintains a safe following distance and adjusts its speed based on the vehicle directly ahead. It is also equipped with rule-based lane change and overtake maneuvers to enable lateral control.
- **Trained RL Agent**: A standard TD3 agent (Fujimoto et al., 2018) trained via RL on the original, non-adversarial WOMD scenarios. The observation input of the RL agent contains ego states, navigation information and surrounding information provided by 2D LiDAR sensor. This agent represents a competent end-to-end learning-based policy but is naive to adversarial behaviors, allowing us to measure the impact of adversarial training.
- **Rule-based Expert**: A sophisticated and safety-oriented rule-based policy. The rule-based policy operates on perfect state observations, which directly provide the ground-truth positions of surrounding vehicles. This expert agent employs a hierarchical control structure. At the low level, it uses separate PID controllers for lateral and longitudinal control. Critically, it features a proactive safety layer that uses a kinematic bicycle model to forecast its own trajectory and the future states of surrounding vehicles. It continuously checks for potential future collisions by performing oriented bounding box intersection tests. If a high-risk situation is predicted, it activates the collision avoidance strategy, which is to reduce the vehicle speed. This expert serves as a challenging SUT, representing a robust, well-engineered AD system.

**Open-loop Evaluation.** The open-loop evaluation protocol is designed to benchmark the effectiveness and characteristics of different scenario generation methods against a target ego agent behavior, ensuring a fair comparison. The process follows a two-stage procedure (Zhang et al., 2023) for each scenario and each SUT:

1. **Ego Trajectory Collection**: First, the SUT (e.g., the Rule-based Expert or the Trained RL Agent) is executed in the original, unmodified WOMD scenario. Its complete trajectory over the 9-second horizon is recorded. This step establishes a consistent and deterministic behavioral target for all adversarial generation methods.
2. **Adversarial Attack and Re-simulation**: The environment is then reset to the initial state of the same scenario. The adversarial generation method under evaluation uses the ego trajectory recorded in the first stage as a target to generate an optimal adversarial trajectory

for a designated background vehicle. Finally, the scenario is re-simulated with the SUT and the newly generated adversarial trajectory.

During the re-simulation, we measure key metrics, including the success rate of the attack (i.e., collision rate), the adversarial reward (measuring the severity of the interaction), and various realism metrics that penalize kinematically implausible behaviors and map violations. In addition, we calculate the Wasserstein distance (WD) using the kinematic profiles (speed, yaw rate, and acceleration) of the generated and logged scenarios, serving as a distributional metric. This open-loop setup isolates the performance of the generator by holding the ego agent's reactive behavior constant across all compared methods.

Note that the measure of adversarial reward is different for the replay policy and others (Tables 1 and 2). For interactive policies, this reward is calculated using the resulting ego trajectory after interacting with the adversarial agent, i.e., after the re-simulation step.

**Closed-loop Adversarial Training with Dual Curriculum Learning.** The ultimate goal is to integrate SAGE into a closed-loop training pipeline to progressively improve the capabilities of the policy $\pi_{\text{ego}}$, as discussed in section 3.3 and A.1.2. In this setup, the scenario generator and the RL agent (a TD3 agent with hyperparameters detailed in Tab. 6) are trained concurrently. A dual curriculum is designed to alleviate catastrophic forgetting, where $\pi_{\text{ego}}$ is overly optimized for corner cases and fails in normal conditions.

- **SAGE in RL Loops.** The adversarial training follows the min-max structure in section A.1.2. At each iteration, we solve the inner-loop optimization by generating a new adversary tailored to the current policy $\pi_{\text{ego}}^{(i)}$. Instead of having the adversary find a single worst case, we define the adversarial environment $f_{\text{adv}}$ from the mixed policy $\pi_{\theta(\lambda^{(i)})}$. This process samples from a controllable adversarial priors: $\tau_{\text{adv}}^{(i+1)} \sim \text{Generate}(\pi_{\theta(\lambda^{(i)})}, \pi_{\text{ego}}^{(i)}, c^{(i)})$, where $c^{(i)}$ is the current context and $\lambda^{(i)}$ is the interpolation weight at iteration $i$. The ego is then trained for several steps, forming the outer loop: $\pi_{\text{ego}}^{(i+1)} \leftarrow \text{Update}_{\text{RL}}(\pi_{\text{ego}}^{(i)}, \tau_{\text{adv}}^{(i+1)})$. At the end of each training episode, the RL agent's trajectory is recorded and stored in a buffer. When an adversarial episode is initiated, the scenario generator uses the most recent trajectories from this buffer to generate a new adversarial scenario specifically tailored to the agent's current policy and its emergent weaknesses. This iterative co-evolution forces the ego agent to continuously adapt. Unlike training on a fixed set of scenarios, SAGE generates a dynamically shifting distribution of scenarios. This enhances the generalizability of $\pi_{\text{ego}}$, as it learns to handle a wide range of adversarial behaviors rather than overfitting to a narrow set of hard but unrealistic cases.

- **Dual Curriculum.** As discussed in section 3.3, SAGE enables a natural implementation of a scenario curriculum along two axes: intensity and frequency. Exposing a nascent policy to frequent aggressive scenarios from the start can destabilize RL training. We mitigate this by smoothly increasing the challenge as the agent becomes more competent. (1) *Intensity:* The adversarial intensity is controlled by annealing $\lambda$ over the course of training. Let $T_{\text{total}}$ be the total number of training steps. At timestep $t$, $\lambda^{(t)}$ is given by: $\lambda^{(t)} = \lambda_{\text{start}} + (\lambda_{\text{end}} - \lambda_{\text{start}}) \cdot \min\left(\frac{t}{T_{\text{total}}}, 1.0\right)$. This ensures the ego agent first learns to handle common deviations before being confronted with safety-critical maneuvers. (2) *Frequency:* Simultaneously, we control how often the ego agent encounters an adversarial scenario. At the end of each episode, we decide whether to generate an adversary with a probability $p_{\text{adv}}^{(t)}$, which is given by: $p_{\text{adv}}^{(t)} = p_{\text{start}} + (p_{\text{end}} - p_{\text{start}}) \cdot \min\left(\frac{t}{T_{\text{total}}}, 1.0\right)$, where $p_{\text{start}}$ and $p_{\text{end}}$ are initial and final values. Specifically, it starts from a low probability of 0.1 and achieves 0.9 by the halfway point. Second, the weight of the adversarial objective $w_{\text{adv}}$ is annealed from an initial value of 0.5 to a final value of 1.0.

After several training steps, the RL agent is evaluated on both normal and adversarial scenarios (with a fixed $w_{\text{adv}} = 1.0$ for the latter). The performance is measured by collision rate, route completion rate, total environment reward, and total cost. The environment reward is the cumulative sum of rewards throughout an episode, reflecting overall task performance. It includes a dense reward for progress along the planned route, a positive reward for successfully reaching the destination, and

significant penalties for failures like crashing or going off the road. The total cost is a pure safety metric, representing the cumulative sum of costs from safety violations during an episode. A cost of 1.0 is incurred for each instance of a crash or an out-of-road event. An episode with zero total cost is considered perfectly safe, making this a direct measure of the agent's safety failures. In this experiment, we use 300 randomly selected scenarios for RL training, and 70 scenarios are used for evaluation. To evaluate the superiority of SAGE in closed-loop training, we compare it with several different scenario generators in the same training environment, including Replay, CAT, and a rule-based generator.

## B.2 DETAILED FORMULATION OF REWARD AND PENALTY FUNCTIONS

This section provides the detailed mathematical formulations for the functions used to evaluate and filter generated trajectories. These include the adversarial reward ($R_{\text{adv}}$), the realism penalty ($P_{\text{real}}$), and the binary map compliance function ($F(\tau, \mathcal{M})$). For full reproducibility, we provide a list of all hyperparameters used in reward functions in Table 7.

Table 7: Hyperparameters for reward functions and HGPO.

| Category | Hyperparameter | Value |
|---|---|---|
| HGPO Fine-tuning | Optimizer | AdamW |
| | Learning Rate | $1.0 \times 10^{-5}$ |
| | Training Epochs | 200 |
| | Batch Size | 1 (scenario) |
| | Group Size (N) | 32 |
| | Max Pairs per Group (K) | 8 |
| | Preference Margin ($\delta_m$) | 0.2 |
| | Beta ($\beta$) | 0.05 |
| Preference Reward | Adversarial Expert ($\pi_{\theta_{\text{adv}}}$) | |
| | $w_{\text{adv}}$ / $w_{\text{real}}$ | 0.9 / 0.1 |
| | Realism Expert ($\pi_{\theta_{\text{real}}}$) | |
| | $w_{\text{adv}}$ / $w_{\text{real}}$ | 0.1 / 0.9 |
| $R_{\text{adv}}$ Components | Collision Reward Scale | 10.0 |
| | Proximity Reward Scale | 1.0 |
| | Proximity Decay Rate | 0.2 |
| $P_{\text{real}}$ Components | $w_{\text{turn}}$ | 5.0 |
| | $w_{\text{stop-turn}}$ | 3.0 |
| | Kinematic Penalty Factor (Accel) | 5.0 |
| | Kinematic Penalty Factor (Ang. Vel.) | 5.0 |
| | Accel. Comfort Zone | 7.0 m/s$^2$ |
| | Lat. Accel. Comfort Zone | 6.0 m/s$^2$ |
| | Ang. Vel. Comfort Zone | 0.8 rad/s |
| | Max Reasonable Turn | $\pi$ radians |
| Map Feasibility | Cross Solid Line Penalty | 50.0 |
| | Crash Object Penalty | 10.0 |

**Adversarial Reward ($R_{\textbf{adv}}$)** The adversarial reward $R_{\text{adv}}(\tau_{\text{adv}}, \tau_{\text{ego}})$ is designed to quantify the threat level of a generated adversarial trajectory $\tau_{\text{adv}}$ with respect to the ego vehicle's recorded (re-simulated) future trajectory $\tau_{\text{ego}}$. A high reward should correspond to a high-risk scenario. The function provides a dense signal by rewarding both direct collisions and near-miss events.

Let the adversarial and ego trajectories be represented as sequences of 2D positions over $T$ future timesteps:

$$\tau_{\text{adv}} = \{\mathbf{p}_t^{\text{adv}}\}_{t=1}^T, \quad \mathbf{p}_t^{\text{adv}} \in \mathbb{R}^2$$
$$\tau_{\text{ego}} = \{\mathbf{p}_t^{\text{ego}}\}_{t=1}^T, \quad \mathbf{p}_t^{\text{ego}} \in \mathbb{R}^2$$

We define $B(\mathbf{p}, \psi, l, w)$ as the oriented bounding box (polygon) of a vehicle at position $\mathbf{p}$ with yaw $\psi$, length $l$, and width $w$. A collision occurs at timestep $t$ if the bounding boxes of the two vehicles

intersect: $B(\tau_{\text{adv},t}) \cap B(\tau_{\text{ego},t}) \neq \emptyset$. Let $t_{\text{coll}}$ be the first timestep at which a collision occurs:

$$t_{\text{coll}} = \min\{t \in [1, T] \mid B(\tau_{\text{adv},t}) \cap B(\tau_{\text{ego},t}) \neq \emptyset\}$$

If no collision occurs, we define $t_{\text{coll}} = \infty$.

The adversarial reward is then defined as a piecewise function:

$$R_{\text{adv}}(\tau_{\text{adv}}, \tau_{\text{ego}}) = \begin{cases} C_{\text{coll}} \left(1 - \frac{t_{\text{coll}}}{T}\right) & \text{if } t_{\text{coll}} \leq T \\ C_{\text{prox}} \exp(-\lambda_{\text{prox}} \cdot d_{\text{min}}) & \text{if } t_{\text{coll}} > T \end{cases} \tag{11}$$

where $C_{\text{coll}}$ is a large constant reward for achieving a collision (e.g., 10.0). The term $(1 - t_{\text{coll}}/T)$ incentivizes earlier collisions, as they are typically more critical and harder to avoid. $d_{\text{min}} = \min_{t \in [1,T]} \|\mathbf{p}_t^{\text{adv}} - \mathbf{p}_t^{\text{ego}}\|_2$ is the minimum Euclidean distance between the vehicle centers throughout the horizon. $C_{\text{prox}}$ is a scaling factor for the proximity reward (e.g., 1.0), and $\lambda_{\text{prox}}$ is a decay rate (e.g., 0.2). This term provides a smooth, dense reward signal for near-misses, encouraging the agent to generate trajectories that are spatially close to the ego vehicle even if they do not result in a collision.

**Realism Penalty ($P_{\text{real}}$)** The realism penalty, $P_{\text{real}}(\tau)$, is designed to discourage trajectories that are physically implausible or exhibit unnatural driving behavior. It is composed of two sub-penalties: a kinematic penalty $P_{\text{kin}}$ and a behavioral penalty $P_{\text{beh}}$.

Let a trajectory $\tau$ be a sequence of states $\{\mathbf{p}_t, \psi_t\}_{t=1}^{T}$, where $\psi_t$ is the yaw angle at timestep $t$. We first compute the primary kinematic quantities with a time interval of $\Delta t = 0.1s$:

- Speed: $s_t = \|\mathbf{p}_t - \mathbf{p}_{t-1}\|_2 / \Delta t$

- Longitudinal Acceleration: $a_{\text{long},t} = (s_t - s_{t-1})/\Delta t$

- Angular Velocity: $\omega_t = (\psi_t - \psi_{t-1})/\Delta t$, where the heading $\psi$ is unwrapped to handle angle discontinuities.

- Lateral Acceleration: $a_{\text{lat},t} = s_t \cdot \omega_t$

We define a smooth penalty function $S(x, x_{\text{thresh}}) = \log(1 + \exp(|x| - x_{\text{thresh}}))$, which penalizes values of $|x|$ that exceed a threshold $x_{\text{thresh}}$.

Kinematic Penalty ($P_{\text{kin}}$). This penalty discourages violations of physical limits.

$$P_{\text{kin}}(\tau) = \frac{1}{T} \sum_{t=1}^{T} \Big( w_a \left[ S(a_{\text{long},t}, a_{\text{max}}) + S(a_{\text{lat},t}, a_{\text{lat,max}}) \right]$$
$$+ w_\omega S(\omega_t, \omega_{\text{max}}) \Big) \tag{12}$$

where $w_a$ and $w_\omega$ are weight factors, and $\{a_{\text{max}}, a_{\text{lat,max}}, \omega_{\text{max}}\}$ are comfort thresholds for longitudinal acceleration (e.g., 7.0 m/s$^2$), lateral acceleration (e.g., 6.0 m/s$^2$), and angular velocity (e.g., 0.8 rad/s), respectively.

Behavioral Penalty ($P_{\text{beh}}$). This penalty targets unnatural maneuvers, such as spinning in place or executing excessively sharp turns over the trajectory horizon.

$$P_{\text{beh}}(\tau) = w_{\text{turn}} S(\Delta\psi_{\text{total}}, \Delta\psi_{\text{max}}) + \frac{w_{\text{stop-turn}}}{T} \sum_{t=1}^{T} \frac{|\omega_t|}{s_t + \epsilon} \tag{13}$$

where: $\Delta\psi_{\text{total}} = |\psi_T - \psi_1|$ is the total change in heading over the trajectory. The penalty discourages total turns exceeding a reasonable maximum $\Delta\psi_{\text{max}}$ (e.g., $\pi$ radians). The second term penalizes high angular velocity at low speeds, a characteristic of unrealistic spinning maneuvers. $w_{\text{turn}}$ and $w_{\text{stop-turn}}$ are weight factors, and $\epsilon$ is a small constant to prevent division by zero.

The total realism penalty is the sum of these components: $P_{\text{real}}(\tau) = P_{\text{kin}}(\tau) + P_{\text{beh}}(\tau)$.

**Map Compliance Function ($F(\tau, \mathcal{M})$)** Unlike the continuous-valued rewards and penalties, map compliance is treated as a hard, binary constraint. The function $F(\tau, \mathcal{M})$ returns 1 if the trajectory $\tau$ is feasible with respect to the map $\mathcal{M}$, and 0 otherwise. A trajectory is deemed infeasible if it violates any of the following conditions:

1. **Road Boundary Violation**: The trajectory must remain within the drivable area. Let $\mathcal{L}_{\text{impassable}} \subset \mathcal{M}$ be the set of impassable map polylines (e.g., road edge boundaries). The trajectory is infeasible if the vehicle's bounding box $B(\tau_t)$ intersects with any of these lines at any time.

$$\exists t \in [1, T], \exists l \in \mathcal{L}_{\text{impassable}} \quad \text{s.t.} \quad B(\tau_t) \cap l \neq \emptyset \implies F(\tau, \mathcal{M}) = 0$$

2. **Static Object Collision**: The trajectory must not collide with other static or near-static background vehicles. Let $\mathcal{O}_{\text{static}}$ be the set of non-player vehicles in the scenario, each with their own future trajectory of bounding boxes $B(\tau_{\text{obj},t})$.

$$\exists t \in [1, T], \exists \tau_{\text{obj}} \in \mathcal{O}_{\text{static}} \quad \text{s.t.} \quad B(\tau_t) \cap B(\tau_{\text{obj},t}) \neq \emptyset \implies F(\tau, \mathcal{M}) = 0$$

If none of these violation conditions are met, the trajectory is considered feasible, and $F(\tau, \mathcal{M}) = 1$. This strict separation of feasibility from preference allows the optimization to focus on learning meaningful trade-offs within the space of valid behaviors. Computationally, these checks are implemented efficiently using spatial data structures like STR-trees and broad-phase/narrow-phase collision detection.

### B.3 WEIGHT EXTRAPOLATION VIA PREFERENCE VECTORS

While weight interpolation effectively traces the Pareto front *between* the two expert models, it cannot generate scenarios that are more extreme than what the experts themselves can produce. To overcome this, we introduce a post-hoc extrapolation technique inspired by the model editing (Ilharco et al., 2022; Liu, 2025). This section provides a more detailed discussion on this weight extrapolation technique introduced in section 3.3. The core idea is to reframe the model merging process in terms of preference (task) vectors, which enables principled extrapolation beyond the space spanned by the expert models.

**Preference Vectors as Learned Task Representations.** We begin by formally defining the preference vectors (Ilharco et al., 2022). These vectors capture the specific parameter updates that involve each preference. Given a pretrained reference model $\pi_{\text{ref}}$ with parameters $\theta_{\text{ref}}$, and two expert models, $\pi_{\theta_{\text{adv}}}$ and $\pi_{\theta_{\text{real}}}$, trained to specialize in adversariality and realism respectively, we define their corresponding preference vectors as:

$$\begin{aligned} \Delta_{\text{adv}} &= \theta_{\text{adv}} - \theta_{\text{ref}}, \\ \Delta_{\text{real}} &= \theta_{\text{real}} - \theta_{\text{ref}}. \end{aligned} \tag{14}$$

Each vector $\Delta$ represents the directional update in the high-dimensional parameter space that adapts the general-purpose reference model to satisfy a specific preference objective (e.g., maximizing $R_{\text{adv-pref}}$). It contains the knowledge required for that particular task.

**Equivalence of Model Interpolation and Vector Interpolation.** The linear interpolation of model weights, as described in Eq. 6, can be shown to be equivalent to applying an interpolated preference vector to the reference model. This is clear by rewriting interpolated parameters $\theta(\lambda)$:

$$\begin{aligned} \theta(\lambda) &= (1 - \lambda)\theta_{\text{real}} + \lambda\theta_{\text{adv}}, \\ &= (1 - \lambda)(\theta_{\text{ref}} + \Delta_{\text{real}}) + \lambda(\theta_{\text{ref}} + \Delta_{\text{adv}}), \\ &= (1 - \lambda)\theta_{\text{ref}} + (1 - \lambda)\Delta_{\text{real}} + \lambda\theta_{\text{ref}} + \lambda\Delta_{\text{adv}}, \\ &= \theta_{\text{ref}} + \lambda\Delta_{\text{adv}} + (1 - \lambda)\Delta_{\text{real}}. \end{aligned} \tag{15}$$

This shows that creating a mixing of expert models is mathematically identical to starting with the reference model and adding a weighted combination of the preference vectors. This perspective shifts the focus from merging final models to combining the underlying learned skills.

**Extrapolation for Pareto Front Improvement.** The interpolated model $\pi_{\theta(\lambda)}$ is confined to the line segment connecting $\theta_{\text{real}}$ and $\theta_{\text{adv}}$ in the parameter space. This traces out a corresponding path on the Pareto front of objectives, but this path is not guaranteed to be the globally optimal one (will be discussed below). The optimization landscape for the combined objectives may have better solutions that lie outside this segment. Extrapolation allows us to search for these superior solutions (Liu, 2025). Instead of being limited to convex combinations of the experts, we can form an extrapolated policy $\pi_{\theta_{\text{ext}}}$ by moving beyond the interpolation segment. We construct a new policy by adding a linear combination of preference vectors to an existing point:

$$\theta_{\text{ext}} = \theta_{\text{base}} + \sum_{i \in \{\text{adv, real}\}} \phi_i \Delta_i, \tag{16}$$

where $\theta_{\text{base}}$ is a starting point (e.g., $\theta_{\text{ref}}$, one of the experts, or an interpolated $\theta(\lambda)$) and $\phi_i$ are scalar coefficients. For instance, setting $\lambda = 1$ and $\phi_{\text{adv}} > 0$ pushes the model to become even more adversarial than the original $\pi_{\theta_{\text{adv}}}$. This allows for the generation of truly extreme, out-of-distribution scenarios, providing a more rigorous stress test for the ego agent without any additional training.

**Intuition from Gradient Optimization.** To understand why this works, we can draw an analogy to gradient-based optimization, following the insights from Liu (2025). Let $R_{\text{adv}}(\theta)$ and $R_{\text{real}}(\theta)$ be the expected preference rewards that the expert models $\pi_{\theta_{\text{adv}}}$ and $\pi_{\theta_{\text{real}}}$ were optimized for, respectively. The fine-tuning process that produced $\theta_{\text{adv}}$ from $\theta_{\text{ref}}$ can be seen as an approximation of moving along the gradient of $R_{\text{adv}}$. Therefore, the resulting preference vector $\Delta_{\text{adv}}$ is closely related to the integrated gradient of the objective function. A first-order approximation suggests:

$$\Delta_{\text{adv}} = \theta_{\text{adv}} - \theta_{\text{ref}} \approx \eta \cdot \mathbb{E}_{\tau \sim \pi} \left[ \nabla_\theta R_{\text{adv}}(\theta) \right], \tag{17}$$

where $\eta$ represents an effective learning rate over the entire training process.

Next, the extrapolation step $\sum \phi_i \Delta_i$ is approximately equivalent to taking a new optimization step in a direction defined by a weighted combination of the original objective gradients:

$$\sum_i \phi_i \Delta_i \approx \eta \cdot \mathbb{E}_{\tau \sim \pi} \left[ \nabla_\theta \left( \sum_i \phi_i R_i(\theta) \right) \right]. \tag{18}$$

Recall that the interpolated model $\pi_{\theta(\lambda)}$ might have converged to a point in the parameter space that is a local optimum or a saddle point with respect to the true, combined multi-objective landscape. The extrapolation, by adding $\sum \phi_i \Delta_i$, provides a new gradient-like momentum. This momentum, constructed from the gradients of diverse objectives, can help the policy skip out of the local optimum and move towards a region of the parameter space that yields a better trade-off, which is a point on a superior Pareto front. This allows us to generate out-of-distribution scenarios that can be crucial for the safety assessment of AD systems.

### B.4 PSEUDOCODE FOR OFFLINE PREFERENCE OPTIMIZATION AND ONLINE RL TRAINING

The complete procedures for offline preference optimization and closed-loop RL training are summarized in Algorithms 1 and 2.

## C THEORETICAL ANALYSIS

In this section, we provide a detailed theoretical analysis to bound the suboptimality gap between the true optimal solution for a user's preference and the best solution achievable by linearly interpolating two mixed-reward expert motion models. We aim to show that for a given user preference, the optimal parameters for the mixed reward are well approximated by interpolation of expert model weights, as a justification for Theorem 1. We first consider a simplified version and then extend the analysis to a more general case. Finally, we provide a detailed analysis of Proposition 1.

**General Assumptions.** Before presenting the detailed derivations, we first clarify general assumptions that bridge the theoretical analysis with the practical implementation of our method. Recall that the fine-tuning is performed using the HGPO loss (Eq. 4), which is a preference-based objective.

---

**Algorithm 1** Offline Fine-tuning of Expert Models with HGPO

---

1: **Input:** Pre-trained motion model $\pi_{\text{ref}}$, scenarios dataset $\mathcal{D}_{\text{scenarios}}$, expert rewards $R_{\text{adv-pref}}$, $R_{\text{real-pref}}$ (from Eq. 5), group size $N$, pairs per group $K$, margin $\delta_m$, DPO parameter $\beta$.
2: **function** FineTuneExpert($R_{\text{expert}}$)
3:     Initialize policy $\pi_\theta$ with weights from $\pi_{\text{ref}}$.
4:     Initialize an optimizer for $\theta$.
5:     **for** each training epoch **do**
6:         **for** each scenario context $c \in \mathcal{D}_{\text{scenarios}}$ **do**
7:             Sample a group of trajectories using the latest policy $\mathcal{G}_c = \{\tau_i\}_{i=1}^N \sim \pi_\theta(\cdot|c)$.
8:             Partition $\mathcal{G}_c$ into feasible $\mathcal{G}_c^{\text{feas}}$ and infeasible $\mathcal{G}_c^{\text{infeas}}$ sets using $F(\tau, \mathcal{M})$.
9:             Initialize preference pairs set $\mathcal{D}_c^{\text{pref}} \leftarrow \emptyset$.
10:                                                                            ▷ Rule 1: Feasibility First
11:             Sample pairs $(\tau^w, \tau^l)$ with $\tau^w \in \mathcal{G}_c^{\text{feas}}$, $\tau^l \in \mathcal{G}_c^{\text{infeas}}$ and add to $\mathcal{D}_c^{\text{pref}}$.
12:                                                 ▷ Rule 2: Preference within Feasibility
13:             Sample pairs $(\tau^w, \tau^l)$ with $\{\tau^w, \tau^l\} \subset \mathcal{G}_c^{\text{feas}}$ s.t. $R_{\text{expert}}(\tau^w) > R_{\text{expert}}(\tau^l) + \delta_m$, add to $\mathcal{D}_c^{\text{pref}}$.
14:             Limit $|\mathcal{D}_c^{\text{pref}}|$ to $K$.
15:             Compute HGPO loss: $\mathcal{L} = \mathbb{E}_{(\tau^w, \tau^l) \sim \mathcal{D}_c^{\text{pref}}}[-\log \sigma(\beta(\log \frac{\pi_\theta(\tau^w|c)}{\pi_{\text{ref}}(\tau^w|c)} - \log \frac{\pi_\theta(\tau^l|c)}{\pi_{\text{ref}}(\tau^l|c)}))]$.
16:             Update $\theta$ using the gradient of $\mathcal{L}$.
17:         **end for**
18:     **end for**
19:     **return** fine-tuned parameters $\theta$.
20: **end function**
21: $\theta_{\text{adv}} \leftarrow$ FineTuneExpert($R_{\text{adv-pref}}$)
22: $\theta_{\text{real}} \leftarrow$ FineTuneExpert($R_{\text{real-pref}}$)
23: **Output:** Expert model parameters $\theta_{\text{adv}}, \theta_{\text{real}}$.

---

**Algorithm 2** Closed-Loop Adversarial Training with Dual-Axis Curriculum

---

1: **Input:** Fine-tuned expert models $\pi_{\theta_{\text{adv}}}, \pi_{\theta_{\text{real}}}$; RL algorithm; Total timesteps $T_{\text{total}}$; Warm-up steps $T_{\text{start}}$; Schedules $\lambda^{(t)}, p_{\text{adv}}^{(t)}$; Environment 'env'.
2: Initialize ego policy $\pi_{\text{ego}}$ and replay buffer $\mathcal{B}$.
3: $s \leftarrow$ env.reset()
4: **for** $t = 0$ to $T_{\text{total}} - 1$ **do**
5:                                                                        ▷ Ego agent interaction
6:     **if** $t < T_{\text{start}}$ **then** $a \sim$ Uniform(env.action_space)                           ▷ Initial exploration
7:     **else** $a \sim \pi_{\text{ego}}(s)$
8:     **end if**
9:     $s', r, d, \text{info} \leftarrow$ env.step($a$)
10:     Store $(s, a, r, s', d)$ in $\mathcal{B}$; $s \leftarrow s'$
11:     Train $\pi_{\text{ego}}$ using samples from $\mathcal{B}$.
12:     **if** $d$ is `True` **then**
13:         $s \leftarrow$ env.reset(); Get context $c$.                           ▷ Episode ends, reset and generate
14:         Update curriculum $\lambda \leftarrow \lambda^{(t)}$, $p_{\text{adv}} \leftarrow p_{\text{adv}}^{(t)}$ (Sec. B.1).
15:         **if** random() $< p_{\text{adv}}$ **then**                           ▷ Generate an adversarial scenario
16:              $\theta(\lambda) \leftarrow (1 - \lambda)\theta_{\text{real}} + \lambda\theta_{\text{adv}}$
17:             $\tau_{\text{adv}} \sim$ Generate($\pi_{\theta(\lambda)}, \pi_{\text{ego}}, c$) (Sec. 3.3 and Sec. B.1).
18:             env.set_adversary($\tau_{\text{adv}}$)
19:         **else**                                                   ▷ Generate a benign scenario
20:             env.set_adversary(None)
21:         **end if**
22:     **end if**
23: **end for**
24: **Output:** Robust ego policy $\pi_{\text{ego}}$.

---

However, our theoretical analysis is based on the direct optimization of expected reward functions $R(\theta)$ and the geometry of their corresponding loss landscapes. To bridge this gap, we introduce the following assumption, which is well-grounded in the literature, to make the analysis tractable while still capturing the core principles that enable our approach.

**Assumption 1** (Implicit Reward Function for HGPO). *We assume that the process of optimizing the HGPO objective is equivalent to implicitly maximizing an underlying reward function, regularized by the KL-divergence from the reference policy $\pi_{ref}$. This connection is formally established for DPO under the Bradley-Terry framework (Rafailov et al., 2023), which shows that minimizing the DPO loss is equivalent to solving a reward-maximization problem. Therefore, we posit that the continuous and differentiable functions, the expected adversarial reward $R_{adv}(\theta) = \mathbb{E}_{\tau \sim \pi_\theta}[R_{adv}(\tau)]$ and the expected realism reward $R_{real}(\theta) = \mathbb{E}_{\tau \sim \pi_\theta}[-P_{real}(\tau)]$, in our analysis represent these implicitly learned reward proxies for the optimization landscape shaped by the HGPO objective.*

This assumption allows us to analyze the behavior and geometry of the HGPO-optimized solution using reward landscape analysis, even though the practical algorithm operates on preference pairs.

## C.1 Suboptimality Gap with Quadratic Reward Approximation

Before presenting the general suboptimality analysis for non-quadratic functions, we first provide an intuitive derivation under the assumption that the reward landscapes are quadratic. This simplified setting, inspired by the analysis in Rame et al. (2023), allows for a closed-form solution for the suboptimality gap and offers clear insights into how our mixed-expert interpolation method behaves. This analysis serves as a foundation for understanding the more general results in section C.2.

### C.1.1 Problem Formulation

We model the reward functions as quadratic forms, which can be seen as a second-order Taylor approximation of the true reward landscape in the vicinity of the optima.

**Assumption 2** (Simplified Quadratic Rewards). *The expected base rewards $R_{adv}(\theta)$ and $R_{real}(\theta)$ are quadratic functions of the model parameters $\theta \in \mathbb{R}^d$. Specifically, their Hessians are proportional to the identity matrix:*

$$R_{adv}(\theta) = C_{adv} - \frac{\eta_{adv}}{2}\|\theta - \theta_{adv}^*\|^2,$$
$$R_{real}(\theta) = C_{real} - \frac{\eta_{real}}{2}\|\theta - \theta_{real}^*\|^2, \tag{19}$$

*where $\theta_{adv}^*$ and $\theta_{real}^*$ are the unique global optima for the pure adversarial and realism rewards, respectively. The constants $\eta_{adv}, \eta_{real} > 0$ determine the curvature of the reward landscapes, and $C_{(\cdot)}$ are the maximum reward values.*

Under this assumption, we can analytically find the optima for the expert and user reward functions.

**Expert Optima.** Two expert models are trained on mixed rewards, defined by $\beta \in (0.5, 1]$:

$$R_1(\theta) = \beta R_{\text{adv}}(\theta) + (1 - \beta)R_{\text{real}}(\theta),$$
$$R_2(\theta) = (1 - \beta)R_{\text{adv}}(\theta) + \beta R_{\text{real}}(\theta). \tag{20}$$

The global optimal parameters $\theta_1$ and $\theta_2$ that maximize these rewards are found by setting their gradients to zero. The gradient of $R_1(\theta)$ is:

$$\nabla R_1(\theta) = -\beta\eta_{\text{adv}}(\theta - \theta_{\text{adv}}^*) - (1 - \beta)\eta_{\text{real}}(\theta - \theta_{\text{real}}^*). \tag{21}$$

Setting $\nabla R_1(\theta_1) = 0$ and solving for $\theta_1$ yields:

$$\theta_1 = \frac{\beta\eta_{\text{adv}}\theta_{\text{adv}}^* + (1 - \beta)\eta_{\text{real}}\theta_{\text{real}}^*}{\beta\eta_{\text{adv}} + (1 - \beta)\eta_{\text{real}}}. \tag{22}$$

By symmetry, the optimum for the second expert, $\theta_2$, is:

$$\theta_2 = \frac{(1 - \beta)\eta_{\text{adv}}\theta_{\text{adv}}^* + \beta\eta_{\text{real}}\theta_{\text{real}}^*}{(1 - \beta)\eta_{\text{adv}} + \beta\eta_{\text{real}}}. \tag{23}$$

This shows that the expert optima, $\theta_1$ and $\theta_2$, are themselves weighted averages of the pure optima $\theta_{\text{adv}}^*$ and $\theta_{\text{real}}^*$, and thus lie on the line segment connecting them.

**True User Optimum.** A user's preference is defined by $R_\mu(\theta) = \mu R_{\mathrm{adv}}(\theta) + (1 - \mu)R_{\mathrm{real}}(\theta)$, with $\mu \in [0, 1]$. Following the same procedure, the true optimal parameters $\hat{\theta}_\mu$ for the user are:

$$\hat{\theta}_\mu = \frac{\mu\eta_{\mathrm{adv}}\theta^*_{\mathrm{adv}} + (1 - \mu)\eta_{\mathrm{real}}\theta^*_{\mathrm{real}}}{\mu\eta_{\mathrm{adv}} + (1 - \mu)\eta_{\mathrm{real}}}. \tag{24}$$

### C.1.2 DERIVATION OF THE EXACT SUBOPTIMALITY GAP

Our method approximates $\hat{\theta}_\mu$ by interpolating the expert weights: $\theta(\lambda) = (1 - \lambda)\theta_2 + \lambda\theta_1$ for $\lambda \in [0, 1]$. Since $\theta_1$, $\theta_2$, and $\hat{\theta}_\mu$ are all collinear (as they are all different weighted averages of $\theta^*_{\mathrm{adv}}$ and $\theta^*_{\mathrm{real}}$), we can perfectly represent $\hat{\theta}_\mu$ if it falls within the line segment $[\theta_2, \theta_1]$.

To simplify the analysis and gain clearer insight, we consider the case where the reward curvatures are equal, i.e., $\eta_{\mathrm{adv}} = \eta_{\mathrm{real}} = \eta$. The optima then become simple linear interpolations:

$$\begin{aligned}
\theta_1 &= \beta\theta^*_{\mathrm{adv}} + (1 - \beta)\theta^*_{\mathrm{real}}, \\
\theta_2 &= (1 - \beta)\theta^*_{\mathrm{adv}} + \beta\theta^*_{\mathrm{real}}, \\
\hat{\theta}_\mu &= \mu\theta^*_{\mathrm{adv}} + (1 - \mu)\theta^*_{\mathrm{real}}.
\end{aligned} \tag{25}$$

**Optimal approximation.** Our interpolated solution is $\theta(\lambda) = (1 - \lambda)\theta_2 + \lambda\theta_1$. We seek a $\lambda \in [0, 1]$ such that $\theta(\lambda) = \hat{\theta}_\mu$. Substituting the expressions for $\theta_1$ and $\theta_2$:

$$\begin{aligned}
\theta(\lambda) &= (1 - \lambda)((1 - \beta)\theta^*_{\mathrm{adv}} + \beta\theta^*_{\mathrm{real}}) + \lambda(\beta\theta^*_{\mathrm{adv}} + (1 - \beta)\theta^*_{\mathrm{real}}), \\
&= [(1 - \lambda)(1 - \beta) + \lambda\beta]\theta^*_{\mathrm{adv}} + [(1 - \lambda)\beta + \lambda(1 - \beta)]\theta^*_{\mathrm{real}}.
\end{aligned} \tag{26}$$

Then we have the following equality of the coefficients:

$$(1 - \lambda)(1 - \beta) + \lambda\beta = \mu \implies \lambda = \frac{\mu + \beta - 1}{2\beta - 1}. \tag{27}$$

Since $\beta \in (0.5, 1]$, the denominator $2\beta - 1$ is positive. The solution is achievable via interpolation if $\lambda \in [0, 1]$. This condition holds if and only if $\mu \in [1 - \beta, \beta]$.

**Suboptimality Gap.** If the user's preference $\mu$ falls outside the range $[1 - \beta, \beta]$, we cannot perfectly match the true optimum $\hat{\theta}_\mu$. The best achievable solution is found at the boundary of the interpolation range, i.e., at $\lambda = 0$ (giving $\theta_2$) or $\lambda = 1$ (giving $\theta_1$). The suboptimality gap, $\Delta R_\mu = R_\mu(\hat{\theta}_\mu) - R_\mu(\theta_{\mathrm{best}})$, can then be calculated. Consider the linear combination of quadratic functions with the same quadratic coefficient, the user reward function is given by $R_\mu(\theta) = C_\mu - \frac{\eta}{2}\|\theta - \hat{\theta}_\mu\|^2$. Therefore, the gap is:

$$\Delta R_\mu = \frac{\eta}{2}\|\theta_{\mathrm{best}} - \hat{\theta}_\mu\|^2. \tag{28}$$

Let $\mu_{\mathrm{clipped}} = \mathrm{clip}(\mu, 1 - \beta, \beta)$. The optimal point achievable through interpolation corresponds to the user preference $\mu_{\mathrm{clipped}}$. The distance between the true optimum and the best achievable one is:

$$\begin{aligned}
\hat{\theta}_\mu - \theta_{\mathrm{best}} &= (\mu\theta^*_{\mathrm{adv}} + (1 - \mu)\theta^*_{\mathrm{real}}) - (\mu_{\mathrm{clipped}}\theta^*_{\mathrm{adv}} + (1 - \mu_{\mathrm{clipped}})\theta^*_{\mathrm{real}}) \\
&= (\mu - \mu_{\mathrm{clipped}})(\theta^*_{\mathrm{adv}} - \theta^*_{\mathrm{real}})
\end{aligned} \tag{29}$$

This leads to the final expression for the suboptimality gap.

**Theorem 2** (Suboptimality Gap for Quadratic Rewards). *Under Assumption 2 with equal curvatures $(\eta_{adv} = \eta_{real} = \eta)$, the suboptimality gap of the mixed-expert model for a user preference $\mu \in [0, 1]$ is given by:*

$$R_\mu(\hat{\theta}_\mu) - \max_{\lambda \in [0,1]} R_\mu(\theta(\lambda)) = \frac{\eta}{2}(\mu - clip(\mu, 1 - \beta, \beta))^2\|\theta^*_{adv} - \theta^*_{real}\|^2 \tag{30}$$

*where $\beta \in (0.5, 1]$ is the mixing coefficient used to train the expert models.*

**Interpretation.** This result provides several key insights. (1) *Region of Optimality*: Our method is completely optimal when the user's preference $\mu$ lies within the range $[1 - \beta, \beta]$. The width of this range is controlled by the expert mixing coefficient $\beta$. A value closer to $0.5$ creates less specialized experts but covers a wider range of user preferences optimally. A value closer to $1$ creates more specialized experts but narrows the region of perfect optimality. (2) *Error Growth*: Outside the optimal region, the suboptimality gap grows quadratically with the distance of $\mu$ from the boundary of the region. (3) *Task Dissimilarity*: The gap is scaled by $\|\theta_{adv}^* - \theta_{real}^*\|^2$, the squared distance between the pure optima. This confirms the intuition that the approximation is worse when the underlying objectives are fundamentally more conflicting, requiring very different model parameters. Since we fine-tune a pretrained model $\pi_{ref}$, it is plausible that the learned parameters remain in a local region of the weight space with similar geometry.

This simplified analysis demonstrates the effectiveness and predictable behavior of our method. We now proceed to the more general analysis for non-quadratic reward functions, which builds upon these core ideas.

## C.2 Suboptimality Gap with Non-quadratic Reward

In the following, we generalize the above discussion to non-quadratic rewards by leveraging standard assumptions from convex optimization theory. This provides a complete proof of Theorem 1.

### C.2.1 Problem Formulation

Following the above ideas, we have similar components but with the consideration of a more class of reward functions: the expected adversarial reward $R_{adv}(\theta) = \mathbb{E}_{\tau \sim \pi_\theta}[R_{adv}(\tau)]$ and the expected realism reward $R_{real}(\theta) = \mathbb{E}_{\tau \sim \pi_\theta}[-P_{real}(\tau)]$. The definition of other ingredients such as expert rewards, user reward, expert optima, and true optimum keep the same. Our goal is to derive an upper bound for the suboptimality gap: $\Delta R_\mu = R_\mu(\hat{\theta}_\mu) - \max_{\lambda \in [0,1]} R_\mu(\theta_\lambda)$.

To analyze the behavior of our non-quadratic reward functions, we introduce two standard assumptions from optimization theory.

**Assumption 3** (L-Smoothness). *The base reward functions $R_{adv}(\theta)$ and $R_{real}(\theta)$ are differentiable and their gradients are Lipschitz continuous with constants $L_{adv} > 0$ and $L_{real} > 0$, respectively. That is, for any $\theta_a, \theta_b \in \mathbb{R}^d$:*

$$\begin{aligned}
\|\nabla R_{adv}(\theta_a) - \nabla R_{adv}(\theta_b)\| &\leq L_{adv}\|\theta_a - \theta_b\|, \\
\|\nabla R_{real}(\theta_a) - \nabla R_{real}(\theta_b)\| &\leq L_{real}\|\theta_a - \theta_b\|.
\end{aligned} \tag{31}$$

**Assumption 4** (m-Strong Concavity). *The base reward functions $R_{adv}(\theta)$ and $R_{real}(\theta)$ are strongly concave with constants $m_{adv} > 0$ and $m_{real} > 0$, respectively. That is, for any $\theta_a, \theta_b \in \mathbb{R}^d$:*

$$R_{adv}(\theta_a) \leq R_{adv}(\theta_b) + \nabla R_{adv}(\theta_b)^T(\theta_a - \theta_b) - \frac{m_{adv}}{2}\|\theta_a - \theta_b\|^2. \tag{32}$$

*A similar inequality holds for $R_{real}(\theta)$ with constant $m_{real}$.*

**Notes.** While the global reward landscape in deep learning is highly non-concave, these assumptions are reasonable in our case for analyzing the behavior of models in the vicinity of a local optimum found during fine-tuning. (1) *L-Smoothness*: The reward functions $R_{adv}$ and $P_{real}$ described in section B.2 are constructed from compositions of smooth operations (e.g., exponential, logarithm, norms) and the output of a neural network, which is itself a smooth function of its weights $\theta$. The expectation over trajectories further smooths the landscape, making the L-smoothness assumption plausible. (2) *m-Strong Concavity*: This is a stronger assumption. However, it is common in theoretical analysis to assume that the reward landscape is locally strongly concave around a good solution (Kawaguchi, 2016). Furthermore, the use of regularization techniques like L2 weight decay during training explicitly adds a quadratic term to the objective, which helps enforce strong concavity. Our analysis thus characterizes the suboptimality within such a well-behaved local region.

### C.2.2 Derivation of the Suboptimality Gap Bound

The derivation proceeds in four steps.

**Step 1.** We begin by relating the approximation gap to the gradient norm. A standard result for an $m$-strongly concave function $f(x)$ with the maximum at $x^*$ is that $f(x^*) - f(x) \leq \frac{1}{2m}\|\nabla f(x)\|^2$. The user reward $R_\mu(\theta)$ is a linear combination of strongly concave functions and is thus itself strongly concave with constant $m_\mu = (1-\mu)m_{\text{adv}} + \mu m_{\text{real}}$. Applying this result, we can bound the suboptimality gap by:

$$\Delta R_\mu = R_\mu(\hat{\theta}_\mu) - R_\mu(\theta_{\bar{\lambda}}) \leq \frac{1}{2m_\mu}\|\nabla R_\mu(\theta_{\bar{\lambda}})\|^2. \tag{33}$$

Our task now is to bound the squared norm of the gradient at the best interpolated point $\theta_{\bar{\lambda}}$.

**Step 2.** We decompose the user's preference gradient in the expert basis. The user gradient $\nabla R_\mu$ is expressed as a linear combination of the expert gradients $\nabla R_1$ and $\nabla R_2$. The expert rewards are defined by the linear system:

$$\begin{aligned}
\nabla R_1 &= \beta \nabla R_{\text{adv}} + (1-\beta)\nabla R_{\text{real}}, \\
\nabla R_2 &= (1-\beta)\nabla R_{\text{adv}} + \beta \nabla R_{\text{real}}.
\end{aligned} \tag{34}$$

Solving this system for the base gradients $\nabla R_{\text{adv}}$ and $\nabla R_{\text{real}}$ yields:

$$\begin{aligned}
\nabla R_{\text{adv}} &= \frac{\beta \nabla R_1 - (1-\beta)\nabla R_2}{\beta^2 - (1-\beta)^2} = \frac{\beta \nabla R_1 - (1-\beta)\nabla R_2}{2\beta - 1}, \\
\nabla R_{\text{real}} &= \frac{-(1-\beta)\nabla R_1 + \beta \nabla R_2}{\beta^2 - (1-\beta)^2} = \frac{-(1-\beta)\nabla R_1 + \beta \nabla R_2}{2\beta - 1}.
\end{aligned} \tag{35}$$

Substituting these into the definition of the user gradient $\nabla R_\mu = (1-\mu)\nabla R_{\text{adv}} + \mu \nabla R_{\text{real}}$:

$$\begin{aligned}
\nabla R_\mu(\theta) &= \frac{1-\mu}{2\beta - 1}(\beta \nabla R_1 - (1-\beta)\nabla R_2) + \frac{\mu}{2\beta - 1}(-(1-\beta)\nabla R_1 + \beta \nabla R_2), \\
&= \left(\frac{(1-\mu)\beta - \mu(1-\beta)}{2\beta - 1}\right)\nabla R_1(\theta) + \left(\frac{\mu\beta - (1-\mu)(1-\beta)}{2\beta - 1}\right)\nabla R_2(\theta), \\
&= c_1(\mu, \beta)\nabla R_1(\theta) + c_2(\mu, \beta)\nabla R_2(\theta),
\end{aligned} \tag{36}$$

where we define the coefficients $c_1$ and $c_2$ for notational simplicity.

**Step 3.** Using the decomposition from Step 2 and the triangle inequality, we can bound the gradient norm at the interpolated point:

$$\|\nabla R_\mu(\theta_{\bar{\lambda}})\| \leq |c_1|\|\nabla R_1(\theta_{\bar{\lambda}})\| + |c_2|\|\nabla R_2(\theta_{\bar{\lambda}})\|. \tag{37}$$

We now bound each expert gradient term using Assumption 3. The expert reward $R_1$ is $L_1$-smooth with $L_1 = \beta L_{\text{adv}} + (1-\beta)L_{\text{real}}$. Since $\theta_1$ is the maximizer of $R_1$, we have $\nabla R_1(\theta_1) = 0$. Thus:

$$\begin{aligned}
\|\nabla R_1(\theta_{\bar{\lambda}})\| &= \|\nabla R_1(\theta_{\bar{\lambda}}) - \nabla R_1(\theta_1)\| \leq L_1\|\theta_{\bar{\lambda}} - \theta_1\|, \\
&= L_1\|(1-\bar{\lambda})\theta_1 + \bar{\lambda}\theta_2 - \theta_1\| = \bar{\lambda}L_1\|\theta_2 - \theta_1\|.
\end{aligned} \tag{38}$$

Similarly, $R_2$ is $L_2$-smooth with $L_2 = (1-\beta)L_{\text{adv}} + \beta L_{\text{real}}$, and $\nabla R_2(\theta_2) = 0$. Thus:

$$\begin{aligned}
\|\nabla R_2(\theta_{\bar{\lambda}})\| &= \|\nabla R_2(\theta_{\bar{\lambda}}) - \nabla R_2(\theta_2)\| \leq L_2\|\theta_{\bar{\lambda}} - \theta_2\|, \\
&= L_2\|(1-\bar{\lambda})\theta_1 + \bar{\lambda}\theta_2 - \theta_2\| = L_2\|(1-\bar{\lambda})(\theta_1 - \theta_2)\| = (1-\bar{\lambda})L_2\|\theta_2 - \theta_1\|.
\end{aligned} \tag{39}$$

Substituting these bounds back into Eq. 37:

$$\|\nabla R_\mu(\theta_{\bar{\lambda}})\| \leq \left(|c_1|L_1\bar{\lambda} + |c_2|L_2(1-\bar{\lambda})\right)\|\theta_2 - \theta_1\|. \tag{40}$$

**Step 4.** The bound in Eq. 40 depends on the unknown optimal interpolation coefficient $\bar{\lambda}$. To obtain a general bound, we find the maximum value of the term in the parenthesis over all possible $\lambda \in [0, 1]$. Let $f(\lambda) = |c_1|L_1\lambda + |c_2|L_2(1-\lambda)$. Since $f(\lambda)$ is a linear function of $\lambda$, its maximum over the interval $[0, 1]$ must occur at one of the endpoints:

$$\max_{\lambda \in [0,1]} f(\lambda) = \max(f(0), f(1)) = \max(|c_2|L_2, |c_1|L_1). \tag{41}$$

This gives us a general upper bound on the gradient norm:

$$\|\nabla R_\mu(\theta_{\bar{\lambda}})\| \leq \max(|c_1|L_1, |c_2|L_2)\|\theta_2 - \theta_1\|. \tag{42}$$

Finally, we substitute this into our initial inequality from Eq. 33 to arrive at the final bound on the suboptimality gap:

$$\Delta R_\mu \leq \frac{1}{2m_\mu} \left(\max(|c_1|L_1, |c_2|L_2)\|\theta_2 - \theta_1\|\right)^2. \tag{43}$$

The final result is given by:

$$\Delta R_\mu \leq \frac{\max\left(\left(\frac{(1-\mu)\beta - \mu(1-\beta)}{2\beta - 1}\right)^2 L_1^2, \left(\frac{\mu\beta - (1-\mu)(1-\beta)}{2\beta - 1}\right)^2 L_2^2\right)}{2((1-\mu)m_{\text{adv}} + \mu m_{\text{real}})}\|\theta_2 - \theta_1\|^2, \tag{44}$$

where $L_1 = \beta L_{\text{adv}} + (1-\beta)L_{\text{real}}$, $L_2 = (1-\beta)L_{\text{adv}} + \beta L_{\text{real}}$. This concludes our derivation.

**Interpretation.** The final expression shows that the suboptimality of our method is quadratically dependent on the distance between the expert models in parameter space, and is modulated by the conditioning of the reward functions (ratio of smoothness to strong concavity) and the alignment between the user's preference $\mu$ and the expert design $\beta$.

## C.3 ANALYTICAL COMPARISON OF WEIGHT MIXING AND OUTPUT MIXING SCHEMES

In this section, we provide a theoretical analysis comparing the performance of weight mixing (ours) and trajectory mixing (output ensemble) for trajectory generation tasks. Our goal is to understand the conditions under which one method is favored over the other and provide theoretical justification on the Linear Mode Connectivity (LMC) discussed in sections 3.3 and 4.3. The analysis is conducted in a generic loss landscape. In line with Assumption 1, we interpret this loss function $L(\theta)$ as the negative of the implicitly learned expected reward, i.e., $L(\theta) = -R(\theta)$. Therefore, the properties of the loss landscape, such as its curvature (convexity), are directly related to the geometry of the underlying reward landscape (concavity), making this analysis applicable to our problem setting.

**Assumption 5** (Loss as an Analytical Proxy for Preference Objective). *The HGPO loss implicitly encourages the model's output distribution to shift towards trajectories that better align with the desired preferences. In this context, the ideal trajectory $y$ can be conceptualized not as a single ground truth, but as a representative point in the high-preference region of the output space that the model is being steered towards.*

Under this assumption, the conclusions drawn from the MSE-based analysis are argued to hold in principle for our preference-driven optimization setting.

**Lemma 1.** *Let $f(x; \theta) \in \mathbb{R}^C$ be a model that predicts a future trajectory for input $x$, parameterized by weights $\theta \in \mathbb{R}^d$. Consider two fine-tuned models with parameters $\theta_1$ and $\theta_2$. For an interpolation parameter $\alpha \in [0, 1]$, we define:*

1. *The weight-mixed model with parameters $\theta_\alpha = (1-\alpha)\theta_1 + \alpha\theta_2$, producing output $f^{weight}(x; \alpha) = f(x; \theta_\alpha)$.*

2. *The output-mixed (ensemble) model producing output $f^{ens}(x; \alpha) = (1-\alpha)f(x; \theta_1) + \alpha f(x; \theta_2)$.*

*Let the per-sample loss be the Mean Squared Error (MSE), $l(f, y) = \frac{1}{2}\|f - y\|_2^2$, where $y$ is the expected ground-truth trajectory. The expected losses over the data distribution are denoted by $L^{weight}(\alpha) = \mathbb{E}_{x,y}[l(f^{weight}, y)]$ and $L^{ens}(\alpha) = \mathbb{E}_{x,y}[l(f^{ens}, y)]$. Then, the difference in expected loss is approximated by:*

$$L^{weight}(\alpha) - L^{ens}(\alpha) \approx -\frac{\alpha(1-\alpha)}{2}\left(\frac{d^2}{d\alpha^2}L^{weight}(\alpha) - \mathbb{E}_{x,y}\left[\|\Delta f(x)\|_2^2\right]\right), \tag{45}$$

*where $\Delta f(x) = f(x; \theta_2) - f(x; \theta_1)$ is the difference in predictions between the two endpoint models.*

*Proof.* The proof follows Wortsman et al. (2022) and is adapted under the regression framework, which is standard for motion generation (forecasting) tasks. We first establish an exact expression for the loss difference, then approximate the key term related to output difference, connect it to the curvature of the loss landscape, and finally synthesize the result. For clarity, we treat the model output $f$ as a scalar; the extension to the vector case (where $\| \cdot \|_2^2$ is used) is straightforward.

**Step 1.** We begin by deriving an exact, non-approximated expression for the per-sample loss difference. Using the difference of squares formula, $a^2 - b^2 = (a-b)(a+b)$, we have:

$$
\begin{aligned}
l^{\text{weight}} - l^{\text{ens}} &= \frac{1}{2}(f^{\text{weight}} - y)^2 - \frac{1}{2}(f^{\text{ens}} - y)^2, \\
&= \frac{1}{2}\left((f^{\text{weight}} - y) - (f^{\text{ens}} - y)\right)\left((f^{\text{weight}} - y) + (f^{\text{ens}} - y)\right), \\
&= \frac{1}{2}(f^{\text{weight}} - f^{\text{ens}})(f^{\text{weight}} + f^{\text{ens}} - 2y), \\
&= (f^{\text{weight}} - f^{\text{ens}})(f^{\text{ens}} - y) + \frac{1}{2}(f^{\text{weight}} - f^{\text{ens}})^2.
\end{aligned}
\tag{46}
$$

This expression is exact and serves as our starting point. To obtain a simpler, more interpretable approximation, we will focus on the first-order term, $(f^{\text{weight}} - f^{\text{ens}})(f^{\text{ens}} - y)$, and approximate $f^{\text{ens}} - y \approx f^{\text{weight}} - y$.

**Step 2.** The core of the analysis lies in understanding the difference between weight-mixing and ensemble, which relates to the function's convexity. Let $\delta = \theta_2 - \theta_1$ and $\theta_\tau = \theta_1 + \tau\delta$. By applying the fundamental theorem of calculus twice, we can write an exact integral expression for the output difference (Wortsman et al., 2022):

$$
f^{\text{ens}}(x) - f^{\text{weight}}(x) = \int_0^1 \delta^T \nabla_\theta^2 f(x; \theta_\tau)\delta \cdot w_\alpha(\tau)d\tau,
\tag{47}
$$

where $w_\alpha(\tau) = \min((1-\alpha)\tau, \alpha(1-\tau))$ is a triangular weight function and $\nabla_\theta^2 f$ is the Hessian of the model output with respect to its parameters $\theta$.

Following Wortsman et al. (2022), we assume that the Hessian $\nabla_\theta^2 f(x; \theta_\tau)$ is approximately constant along the linear path between $\theta_0$ and $\theta_1$, i.e., $\nabla_\theta^2 f(x; \theta_\tau) \approx \nabla_\theta^2 f(x; \theta_\alpha)$. This leads to:

$$
\begin{aligned}
f^{\text{ens}}(x) - f^{\text{weight}}(x) &\approx \left(\delta^T \nabla_\theta^2 f(x; \theta_\alpha)\delta\right) \int_0^1 w_\alpha(\tau)d\tau, \\
&= \frac{\alpha(1-\alpha)}{2}\left(\delta^T \nabla_\theta^2 f(x; \theta_\alpha)\delta\right).
\end{aligned}
\tag{48}
$$

Therefore, the output difference we need is $f^{\text{weight}}(x) - f^{\text{ens}}(x) \approx -\frac{\alpha(1-\alpha)}{2}\left(\delta^T \nabla_\theta^2 f(x; \theta_\alpha)\delta\right)$.

**Step 3.** Our goal is to express the term involving the output Hessian, $\delta^T \nabla_\theta^2 f\delta$, in terms of the curvature of the loss function itself. The loss curvature along the parameter path is defined by the second derivative of the loss with respect to the interpolation parameter $\alpha$, i.e., $\frac{d^2}{d\alpha^2}l(f(\theta_\alpha), y)$.

First, we compute the first derivative using the chain rule:

$$
\frac{d}{d\alpha}l(f(\theta_\alpha), y) = \frac{\partial l}{\partial f}\frac{df(\theta_\alpha)}{d\alpha}.
\tag{49}
$$

The derivative of the MSE loss with respect to its input $f$ is $\frac{\partial l}{\partial f} = (f - y)$. The derivative of the model output with respect to $\alpha$ is found using the multivariate chain rule:

$$
\frac{df(\theta_\alpha)}{d\alpha} = (\nabla_\theta f(\theta_\alpha))^T \frac{d\theta_\alpha}{d\alpha} = (\nabla_\theta f(\theta_\alpha))^T \delta.
\tag{50}
$$

since $\theta_\alpha = \theta_1 + \alpha\delta$, so $\frac{d\theta_\alpha}{d\alpha} = \delta$. Combining these gives the first derivative of the loss:

$$
\frac{d}{d\alpha}l(f(\theta_\alpha), y) = (f(\theta_\alpha) - y) \cdot ((\nabla_\theta f(\theta_\alpha))^T \delta).
\tag{51}
$$

Next, we compute the second derivative. Let $u = (f(\theta_\alpha) - y)$ and $v = ((\nabla_\theta f(\theta_\alpha))^T \delta)$. Then the derivatives of $u$ and $v$ are:

$$\frac{du}{d\alpha} = \frac{d}{d\alpha}(f(\theta_\alpha) - y) = \frac{df(\theta_\alpha)}{d\alpha} = (\nabla_\theta f(\theta_\alpha))^T \delta,$$
$$\frac{dv}{d\alpha} = \frac{d}{d\alpha}\left((\nabla_\theta f(\theta_\alpha))^T \delta\right) = \delta^T \nabla_\theta^2 f(\theta_\alpha) \delta. \tag{52}$$

Substituting these back into the product rule:

$$\frac{d^2}{d\alpha^2} l(f(\theta_\alpha), y) = \left((\nabla_\theta f(\theta_\alpha))^T \delta\right)\left((\nabla_\theta f(\theta_\alpha))^T \delta\right) + (f(\theta_\alpha) - y)\left(\delta^T \nabla_\theta^2 f(\theta_\alpha) \delta\right),$$
$$= \left((\nabla_\theta f(\theta_\alpha))^T \delta\right)^2 + (f(\theta_\alpha) - y) \cdot (\delta^T \nabla_\theta^2 f(\theta_\alpha) \delta). \tag{53}$$

By rearranging, we can isolate the term containing the output Hessian that appeared in Step 2:

$$(f(\theta_\alpha) - y) \cdot (\delta^T \nabla_\theta^2 f(\theta_\alpha) \delta) = \frac{d^2}{d\alpha^2} l(f(\theta_\alpha), y) - \left((\nabla_\theta f(\theta_\alpha))^T \delta\right)^2. \tag{54}$$

This result forms the link between the model's output geometry and the loss landscape's geometry.

**Step 4.** We combine the results from the previous steps. Starting from the dominant first-order term of the loss difference from Step 1, and substituting the approximations from Steps 2 and 3:

$$l^{\text{weight}} - l^{\text{ens}} \approx (f^{\text{weight}} - f^{\text{ens}})(f^{\text{weight}} - y),$$
$$\approx \left(-\frac{\alpha(1-\alpha)}{2}\left(\delta^T \nabla_\theta^2 f(x; \theta_\alpha)\delta\right)\right)(f(x; \theta_\alpha) - y),$$
$$= -\frac{\alpha(1-\alpha)}{2}\left[(f(\theta_\alpha) - y) \cdot (\delta^T \nabla_\theta^2 f(\theta_\alpha)\delta)\right], \tag{55}$$
$$= -\frac{\alpha(1-\alpha)}{2}\left[\frac{d^2}{d\alpha^2} l(f(\theta_\alpha), y) - \left((\nabla_\theta f(\theta_\alpha))^T \delta\right)^2\right].$$

Taking the expectation over the data distribution $(x, y)$ gives the difference in expected loss:

$$L^{\text{weight}}(\alpha) - L^{\text{ens}}(\alpha) \approx -\frac{\alpha(1-\alpha)}{2}\left[\frac{d^2}{d\alpha^2} L^{\text{weight}}(\alpha) - \mathbb{E}_{x,y}\left[\left((\nabla_\theta f(x; \theta_\alpha))^T \delta\right)^2\right]\right]. \tag{56}$$

Finally, we make the first-order approximation $(\nabla_\theta f(x; \theta_\alpha))^T \delta \approx f(x; \theta_1) - f(x; \theta_0) = \Delta f(x)$. This holds if the gradient $\nabla_\theta f$ is roughly constant between $\theta_0$ and $\theta_1$. This yields the final expression as stated in the lemma. $\square$

**Interpretation.** The approximation in Eq. 45 reveals a trade-off between two key factors:

$$L^{\text{weight}} - L^{\text{ens}} \approx \underbrace{-\frac{\alpha(1-\alpha)}{2}\frac{d^2 L^{\text{weight}}}{d\alpha^2}}_{\text{Term 1: Loss Curvature}} + \underbrace{\frac{\alpha(1-\alpha)}{2}\mathbb{E}\left[\|\Delta f(x)\|_2^2\right]}_{\text{Term 2: Prediction Difference}}. \tag{57}$$

(1) **Curvature:** This term is proportional to the negative second derivative of the loss of the mixed model along the linear path connecting the parameters. For weight mixing to be advantageous, this term needs to be negative, which requires the loss function to be convex along this path ($\frac{d^2 L^{\text{weight}}}{d\alpha^2} > 0$). This directly corresponds to the underlying reward landscape being concave. (2) **Prediction Difference:** This term is proportional to the mean squared difference between the predictions of the two endpoint models. Since it is always non-negative, this term favors the ensemble. It captures the benefit of mixing diverse outputs, a basic principle of ensembling. When two models produce highly dissimilar predictions (high $\|\Delta f\|_2^2$), output mixing is more likely to outperform weight mixing.

LMC suggests that solutions fine-tuned from a common pretrained model lie within a wide and low-loss basin. However, the path connecting two distinct experts is not perfectly zero-curvature. Instead, it can be conceptualized as a high-reward ridge. As the model traverses this path from one expert to another, the combined reward often has an arc-like shape, which signifies reward

concavity. The reward landscape in Fig. 6(c) illustrates this high-reward plateau, and the measured performance in Fig. 6(d) provides direct empirical evidence of this concavity, showing that the rewards of interpolated models are consistently higher than a linear combination of the endpoint rewards. This inherent concavity of the reward path ensures the loss is convex, making Term 1 negative and significant, thereby providing a principled reason for the superiority of weight-space mixing in the fine-tuning regime.

This analysis provides a theoretical basis for why weight mixing is effective, particularly in the fine-tuning regime, where the loss landscape's curvature can make the first term dominant. By training on combined rewards, we ensure our expert models $\theta_{\text{adv}}$ and $\theta_{\text{real}}$ exist in a space where linear interpolation is meaningful for generating intermediate behaviors and desired trade-offs. To help better understand Eq. 57, in the following, we provide an intuitive illustration to formalize the exact condition where Term 1 can dominate.

**Remark 1** (Remark on the Dominance of Loss Curvature Benefit). *Based on Eq. 57, weight mixing is superior if $\mathbb{E}\big[\|\Delta f(x)\|^2\big] < \frac{d^2 L_{weight}}{d\alpha^2}$. To make this condition more intuitive, we formalize it under a local quadratic approximation of the loss landscape. Similar to section C.1, we assume that in the local vicinity of the expert solutions $\theta_1$ and $\theta_2$, the loss function $L(\theta)$ can be approximated by a quadratic form with $L(\theta) \approx \frac{\eta}{2}\|\theta - \theta^*\|^2 + C$, where $\theta^*$ is the minimum of the loss basin, $\eta > 0$ is a scalar representing the curvature of the basin, and $C$ is a constant.*

*The right-hand side represents the curvature of the loss along the linear path $\theta_\alpha = (1-\alpha)\theta_1 + \alpha\theta_2$. We can compute its second derivative with respect to $\alpha$:*

$$\frac{dL_{weight}}{d\alpha} = \nabla L(\theta_\alpha)^\top \frac{d\theta_\alpha}{d\alpha} = \eta(\theta_\alpha - \theta^*)^\top(\theta_2 - \theta_1),$$

$$\frac{d^2 L_{weight}}{d\alpha^2} = \eta(\theta_2 - \theta_1)^\top(\theta_2 - \theta_1) = \eta\|\theta_2 - \theta_1\|^2.$$

*The left-hand side is the expected squared difference in outputs. We can approximate this difference using a first-order Taylor expansion of the model function $f(x; \theta)$ around $\theta_1$:*

$$\Delta f(x) = f(x; \theta_2) - f(x; \theta_1) \approx J(x; \theta_1)(\theta_2 - \theta_1),$$

*where $J(x; \theta) = \frac{\partial f}{\partial \theta}$ is the Jacobian of the model's output with respect to its parameters. The expected squared difference is then:*

$$\mathbb{E}\big[\|\Delta f(x)\|^2\big] \approx \mathbb{E}\big[\|J(x; \theta_1)(\theta_2 - \theta_1)\|^2\big]$$
$$= (\theta_2 - \theta_1)^\top \mathbb{E}\big[J(x; \theta_1)^\top J(x; \theta_1)\big](\theta_2 - \theta_1)$$
$$= (\theta_2 - \theta_1)^\top F(\theta_2 - \theta_1),$$

*where $F$ is the Fisher information matrix, which measures the sensitivity of the model's output to changes in its parameters. By substituting these results back, we obtain the final condition:*

$$(\theta_2 - \theta_1)^\top F(\theta_2 - \theta_1) < \eta\|\theta_2 - \theta_1\|^2,$$
$$\Rightarrow \eta > \frac{(\theta_2 - \theta_1)^\top F(\theta_2 - \theta_1)}{\|\theta_2 - \theta_1\|^2}. \tag{58}$$

*This derivation yields an intuitive condition: weight mixing is superior whenever the intrinsic curvature of the loss landscape exceeds the sensitivity of the model's output in the direction connecting the experts. This condition is readily satisfied in our fine-tuning setting. On one hand, LMC implies that fine-tuned models develop robust representations, making their outputs relatively insensitive to parameter changes along the solution path. On the other hand, because the experts are trained on conflicting objectives, the path connecting them has a non-zero curvature ($\eta > 0$) to reflect the trade-off in rewards, which provides the necessary signal for alignment.*

*This theoretical insight is directly justified by our empirical findings in Fig. 6. The superiority of weight mixing requires the reward function to be concave along the interpolation path. Fig. 6(d) provides direct visual proof, showing the measured reward curves of interpolated models lying strictly above the linear interpolation line. This confirms that the reward landscape possesses the required curvature. This view also reconciles with LMC: the path connecting experts is not a completely flat plateau but a high-reward ridge (Fig. 6(c)). While LMC ensures the solution remains on this low-loss ridge, our analysis shows that it is the ridge's concave curvature that allows weight-space interpolation to discover superior trade-offs.*

# D SUPPLEMENTARY RESULTS

## D.1 ADVERSARIAL GENERATION EVALUATION AGAINST IDM AND RULE-BASED POLICIES

In addition to the Replay and RL policies presented in the main paper, we provide comprehensive benchmark results against two other common types of policies: a rule-based autopilot and the Intelligent Driver Model (IDM). Tab. 8 and Tab. 9 show that the trends observed previously hold. SAGE consistently achieves the best trade-off, delivering competitive adversarial performance while maintaining unparalleled realism and map compliance. Specifically, the realism and map compliance penalties of our generated scenarios are an order of magnitude lower than most baselines, underscoring the general applicability and effectiveness of our HGPO framework across a variety of ego agent behaviors. Without the elaborated design of reward functions, some baselines can generate unexpected and unstable behaviors, which can violate basic traffic rules (see Fig. 8).

Table 8: Evaluation of adversarial generation methods against the **Rule-based/Autopilot** policy.

| Methods | Attack Succ. Rate ↑ | Adv. Reward ↑ | Real. Pen. ↓ Behav. | Kine. | Map Comp. ↓ Crash Obj. | Cross Line | Dist. Diff. (WD) Accel. | Vel. | Yaw |
|---|---|---|---|---|---|---|---|---|---|
| Rule | 24.18% | 0.724 | 2.623 | 111.184 | 1.658 | 6.386 | 7.631 | 10.572 | 0.248 |
| CAT (Zhang et al., 2023) | 18.21% | 0.659 | 8.346 | 3.071 | 2.799 | 9.647 | 1.565 | 7.781 | 0.201 |
| KING (Hanselmann et al., 2022) | 16.43% | 0.558 | 2.445 | 2.565 | 3.116 | 5.807 | 0.956 | 256.921 | 0.097 |
| AdvTrajOpt (Zhang et al., 2022) | 19.00% | 0.543 | 4.445 | 2.818 | 2.567 | 10.833 | 1.753 | 6.207 | 0.271 |
| SEAL (Stoler et al., 2025) | 19.45% | 1.584 | 4.902 | 2.584 | 3.260 | 9.178 | 1.494 | 6.756 | 0.248 |
| GOOSE (Ransiek et al., 2024a) | 5.10% | 0.325 | 2.387 | 14.56 | 4.363 | 14.31 | 5.322 | 8.107 | 0.167 |
| SAGE ($w_{adv} = 0.0$) | 4.89% | 0.397 | 0.332 | 2.000 | 0.707 | 0.951 | 1.460 | 9.332 | 0.055 |
| SAGE ($w_{adv} = 0.5$) | 11.14% | 0.466 | 0.497 | 2.068 | 0.815 | 0.951 | 1.523 | 8.482 | 0.080 |
| SAGE ($w_{adv} = 1.0$) | 19.84% | 0.749 | 1.430 | 2.480 | 0.788 | 1.087 | 2.100 | 8.100 | 0.185 |

Table 9: Evaluation of adversarial generation methods against the **IDM** policy.

| Methods | Attack Succ. Rate ↑ | Adv. Reward ↑ | Real. Pen. ↓ Behav. | Kine. | Map Comp. ↓ Crash Obj. | Cross Line | Dist. Diff. (WD) Accel. | Vel. | Yaw |
|---|---|---|---|---|---|---|---|---|---|
| Rule | 60.54% | 2.763 | 4.833 | 36.947 | 1.973 | 6.856 | 4.466 | 8.447 | 0.309 |
| CAT (Zhang et al., 2023) | 43.48% | 1.746 | 10.275 | 3.207 | 3.345 | 8.696 | 1.595 | 7.550 | 0.231 |
| KING (Hanselmann et al., 2022) | 16.48% | 1.146 | 2.451 | 2.566 | 3.097 | 5.824 | 0.955 | 257.619 | 0.097 |
| AdvTrajOpt (Zhang et al., 2022) | 20.44% | 0.963 | 4.528 | 2.797 | 2.561 | 10.354 | 1.750 | 6.173 | 0.270 |
| SEAL (Stoler et al., 2025) | 33.70% | 0.942 | 5.785 | 2.719 | 3.562 | 11.233 | 1.695 | 7.368 | 0.276 |
| GOOSE (Ransiek et al., 2024a) | 16.39% | 0.439 | 5.092 | 11.25 | 3.722 | 12.64 | 3.716 | 8.223 | 0.157 |
| SAGE ($w_{adv} = 0.0$) | 13.90% | 0.864 | 0.332 | 2.000 | 0.708 | 0.954 | 1.461 | 9.318 | 0.055 |
| SAGE ($w_{adv} = 0.5$) | 20.98% | 1.183 | 0.498 | 2.068 | 0.817 | 0.954 | 1.522 | 8.458 | 0.080 |
| SAGE ($w_{adv} = 1.0$) | 33.51% | 1.779 | 1.433 | 2.480 | 0.790 | 1.090 | 2.098 | 8.070 | 0.185 |

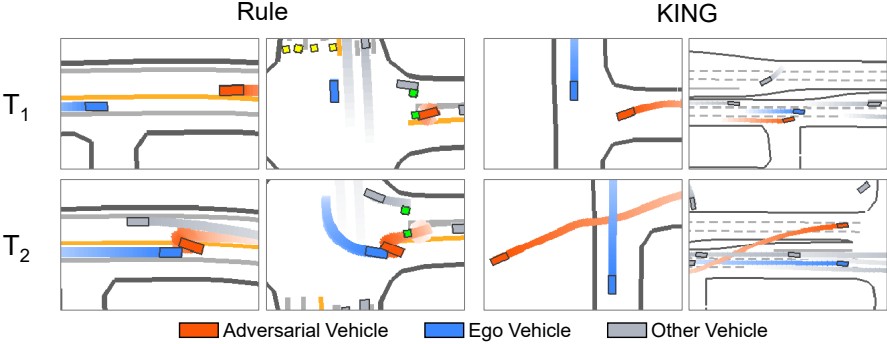

Figure 8: Unexpected case of KING and Rule baselines.

## D.2 FULL RESULTS OF CLOSED-LOOP RL TRAINING

Tab. 10 provides the full evaluation results for the trained RL policies in normal driving environments, sourced from the WOMD log-replay data. We also display the training trajectories of different methods on all metrics in Fig. 9. As can be seen, the agent trained with our adversarial generator not only demonstrates superior robustness in adversarial settings (as shown in Tab. 3 in the main paper and Fig. 9) but also achieves the highest reward, route completion, and average speed in these standard scenarios. This result is critical as it shows our curriculum-based adversarial training enhances the agent's general driving capabilities, rather than causing it to overfit to rare edge cases at the expense of normal performance. This alleviates catastrophic forgetting, which is a key advantage of our closed-loop training framework.

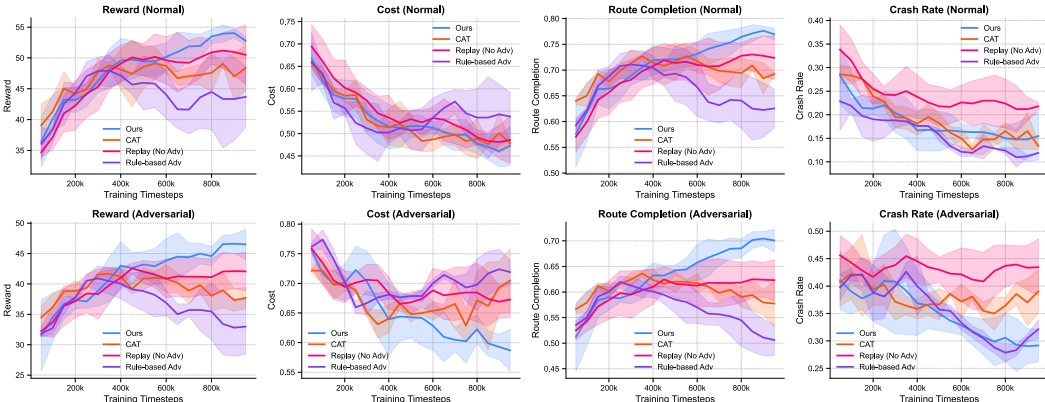

Figure 9: Training performances of the agent under different scenario generation methods.

Table 10: Evaluation of Trained RL Policies in the Log-replay (Normal, WOMD) Environments.

| Methods | Reward ↑ | Cost ↓ | Compl. ↑ | Coll. ↓ | Ave. Speed ↑ | Ave. Jerk ↓ |
|---|---|---|---|---|---|---|
| SAGE | **51.99 ± 1.22** | **0.48 ± 0.05** | **0.77 ± 0.02** | 0.16 ± 0.05 | **9.27 ± 0.03** | **24.97 ± 0.53** |
| CAT | 46.81 ± 4.33 | 0.50 ± 0.05 | 0.67 ± 0.02 | 0.18 ± 0.05 | 7.21 ± 0.05 | 28.15 ± 1.06 |
| Replay (No Adv) | 50.16 ± 5.32 | 0.50 ± 0.07 | 0.72 ± 0.04 | 0.23 ± 0.02 | 9.03 ± 0.03 | 27.53 ± 0.98 |
| Rule-based Adv | 44.61 ± 3.88 | 0.52 ± 0.05 | 0.63 ± 0.04 | **0.13 ± 0.00** | 6.00 ± 0.10 | 28.22 ± 1.44 |

## D.3 FULL RESULTS OF ABLATION STUDIES

To evaluate the effectiveness of our design, we evaluate the performance of the following variations:

1. w/o map penalty: The hierarchical map feasibility preconditioning procedure in section 3.2 is removed;

2. w/o realism penalty: The realism penalty function is removed;

3. w/ weighted map penalty: The hierarchical map feasibility preconditioning procedure is replaced by a direct weighted map penalty, i.e., Eq. 2;

4. w/ DPO: We replace the proposed HGPO with the standard DPO for fine-tuning;

5. The impact of reward margin $\delta_m$ in HGPO: we report the training performances under different $\delta_m$ values;

6. We also report the performance of separately trained models with different expert weights.

Tab. 11 presents quantitative results for ablation studies. The "w/o Map Pen." variant shows a dramatic increase in map compliance penalties (Crash Obj. and Cross Line), confirming the necessity of explicitly handling map constraints. The "w/ Weighted Map Pen." variant improves over having no penalty but is still inferior to our hierarchical conditioning approach, yielding lower adversarial rewards and higher map violations. This reinforces the core idea of HGPO that separating hard

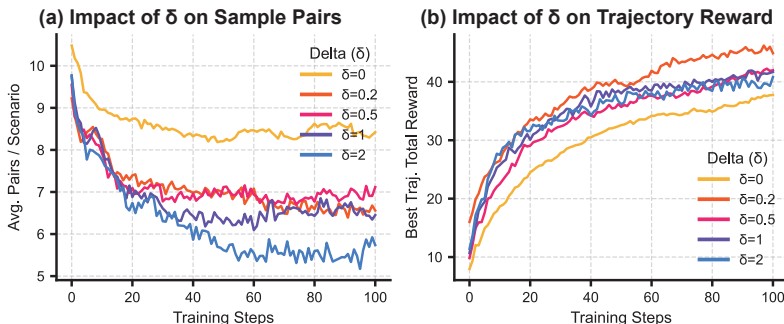

Figure 10: Ablation studies on reward margin $\delta_m$.

feasibility constraints from soft preferences simplifies the learning problem and leads to better outcomes. Comparing our steerable model to separately trained models with fixed scalarized weights ("Scalar. Weight" rows) shows that our interpolation approach can effectively span the performance range of multiple individually trained models, highlighting its great flexibility.

Furthermore, Fig. 10 investigates the impact of the reward margin $\delta_m$. Fig. 10(a) shows that as $\delta_m$ increases, the number of generated preference pairs per scenario decreases, as a larger reward difference is required to form a valid pair. More importantly, Fig. 10(b) illustrates the impact on learning performance. The setting with $\delta_m = 0$ results in the poorest performance, confirming that learning from pairs with insignificant reward differences introduces noise and impedes the training process. Conversely, setting a moderate margin ($\delta_m$ in [0.2, 1.0]) significantly improves performance by focusing the model on more distinct and informative preferences. The model shows robust performance within this range, indicating that it is not overly sensitive to this hyperparameter. We selected $\delta_m = 0.2$ for our main experiments as it empirically provided the best balance between sample efficiency and final performance. An overly large margin, such as $\delta_m = 2.0$, can slightly degrade performance by filtering out too many useful training samples.

Table 11: Ablation study ($w_{\text{adv}} = 1.0, w_{\text{real}} = 0.0$).

| Methods | Attack Succ. Rate ↑ | Adv. Reward ↑ | Real. Pen. ↓ Behav. | Kine. | Map Comp. ↓ Crash Obj. | Cross Line |
|---|---|---|---|---|---|---|
| w/o Map Pen. | 66.40% | 3.841 | 1.580 | 2.665 | 3.848 | 11.65 |
| w/o Real. Pen. | 77.78% | 4.308 | 1.966 | 3.077 | 0.705 | 1.762 |
| w/ Weighted Map. Pen. | 78.05% | 4.115 | 1.438 | 2.411 | 1.165 | 1.491 |
| Scalar. Weight ($w_{\text{adv}} = 3, w_{\text{real}} = 7$) | 50.68% | 2.592 | 0.444 | 2.071 | 0.785 | 0.952 |
| Scalar. Weight ($w_{\text{adv}} = 5, w_{\text{real}} = 5$) | 57.99% | 3.047 | 0.553 | 2.167 | 0.650 | 1.084 |
| Scalar. Weight ($w_{\text{adv}} = 7, w_{\text{real}} = 3$) | 66.67% | 3.508 | 0.786 | 2.396 | 0.678 | 1.491 |
| SAGE ($w_{\text{adv}} = 1.0$) | 76.15% | 4.121 | 1.429 | 2.479 | 0.731 | 1.084 |

### D.4 ADDITIONAL RESULTS OF WEIGHT EXTRAPOLATION

We compare the task vector-based extrapolation scheme (sections B.3 and 3.3) with a more naive approach of directly extrapolating the model weights, i.e., $\theta_{\text{ext}} = (1-\lambda)\theta_{\text{real}} + \lambda\theta_{\text{adv}}$ for $\lambda \notin [0, 1]$. As shown in Fig. 11(b), this direct weight extrapolation quickly leads to unstable models and degraded performance, as evidenced by the sharp drop in preference rewards outside the $[0, 1]$ interpolation range. In contrast, Fig. 11(a) shows that the task vector-based extrapolation provides a much more stable and effective means of pushing the model's behavior beyond its trained distribution. By adding multiples of the learned preference vectors, we can controllably increase both adversarial reward and realism penalty in their respective directions, demonstrating that this method correctly identifies and amplifies the underlying skill vectors within the model's parameter space.

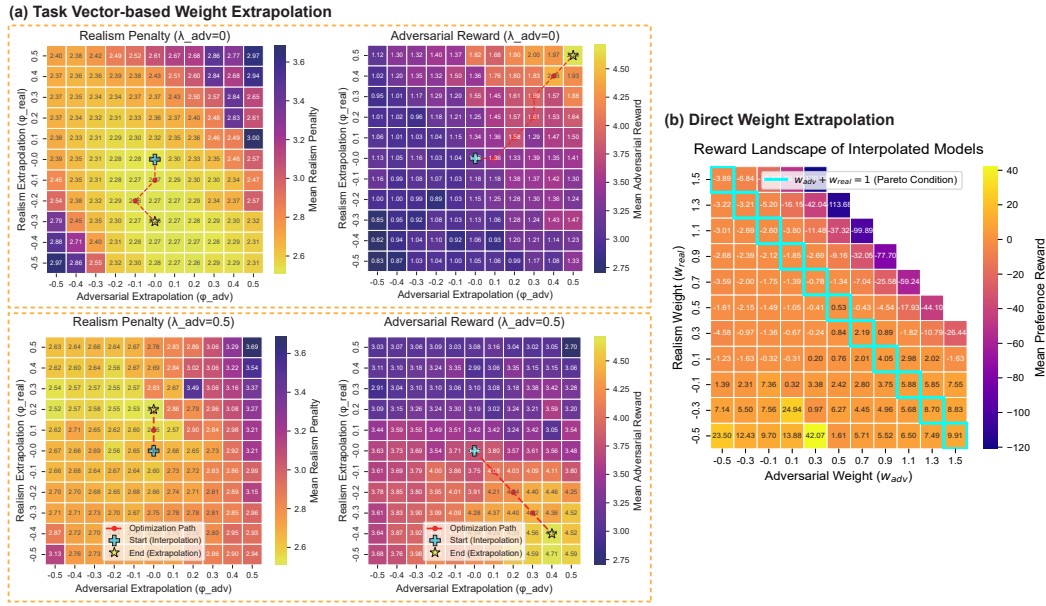

Figure 11: Comparison of vector-based extrapolation and direct weight extrapolation.

## D.5 ADDITIONAL CASE STUDIES

### D.5.1 EXAMPLES OF ADVERSARIAL GENERATION AGAINST DIFFERENT EGO POLICIES

Fig. 12 demonstrates the ability of SAGE to generate effective adversarial behaviors against reactive policies. In addition to the fixed Replay policy, SAGE can also be adopted for discovering long-tailed events of reactive and well-developed driving policies. This indicates its practical value in developing learning- or rule-based ADS.

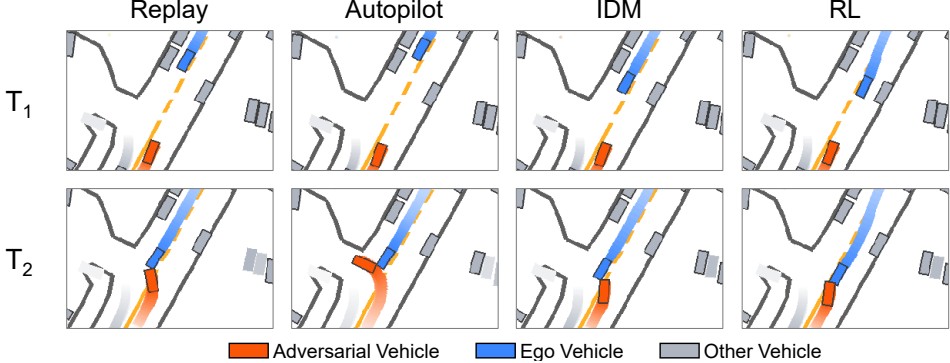

Figure 12: Examples of SAGE-generated adversarial behaviors against different ego policies.

### D.5.2 MORE EXAMPLES OF GENERATED SCENARIOS

Figs. 13 and 14 provide additional qualitative comparisons between SAGE and SOTA baselines across a variety of scenarios, including intersections and lane changes. These examples further illustrate the superior quality of scenarios generated by SAGE. While baselines frequently produce unrealistic behaviors, such as unnatural swerves (CAT), physically impossible braking (Rule), or trajectories that disregard lane boundaries, SAGE consistently generates adversarial behaviors that are challenging, contextually appropriate, and physically plausible. For instance, in Scenario 5, SAGE generates a subtle but dangerous squeeze maneuver, whereas baselines resort to more chaotic actions.

### D.5.3 MORE CASES ON PREFERENCE CONTROLLABLE GENERATION

Fig. 15 shows more examples of the test-time controllability of SAGE. By simply increasing the adversarial weight from 0 to 1, the generated scenarios transition from naturalistic to highly aggressive, enabling efficient target-conditioned testing of AD systems.

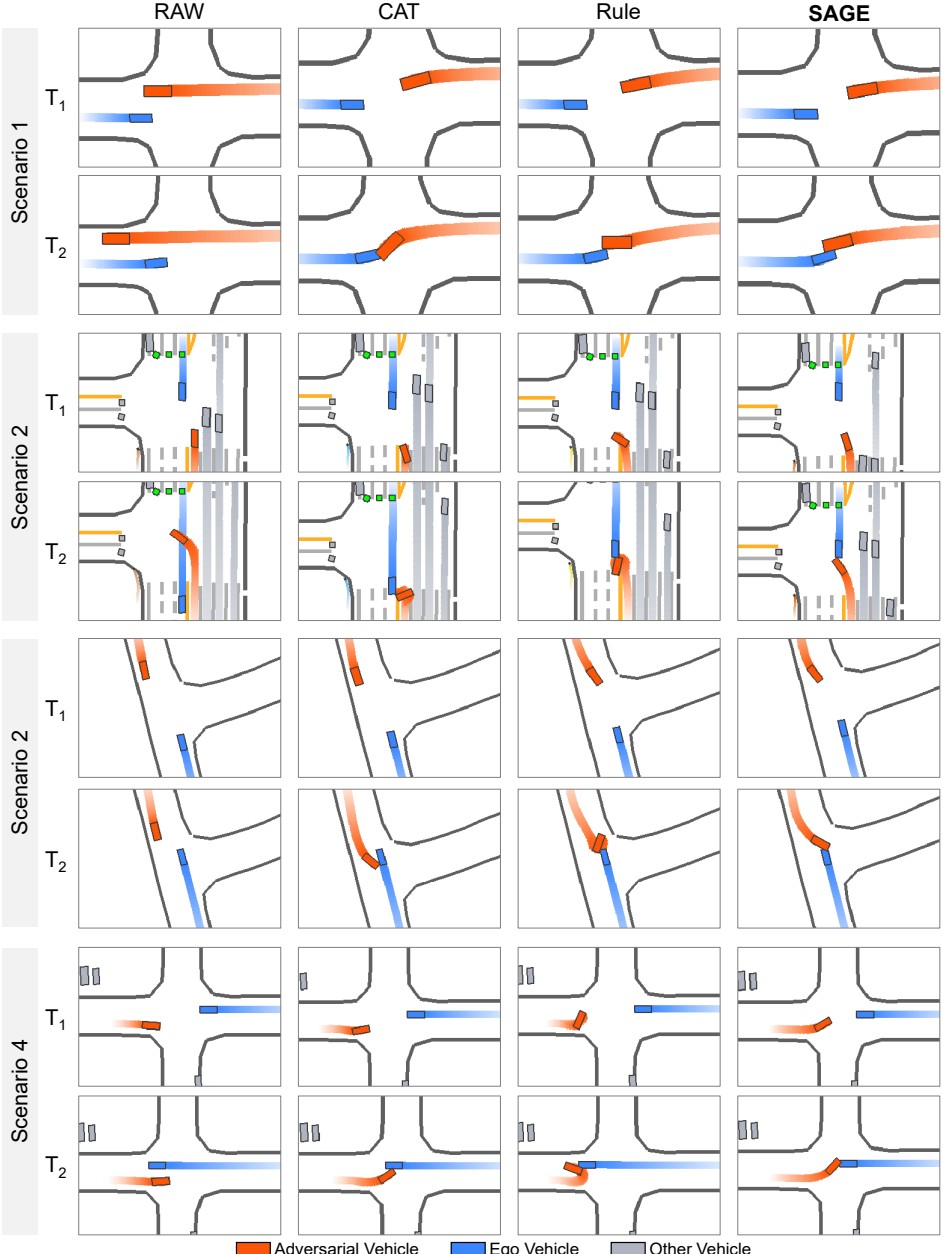

Figure 13: Examples of generated scenarios by different methods.

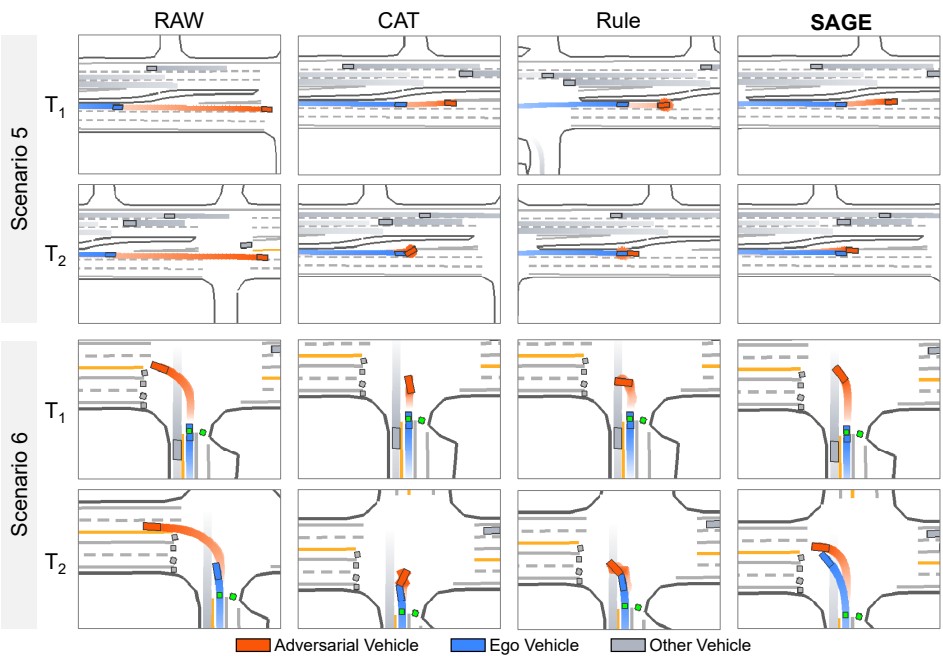

Figure 14: Examples of generated scenarios by different methods.

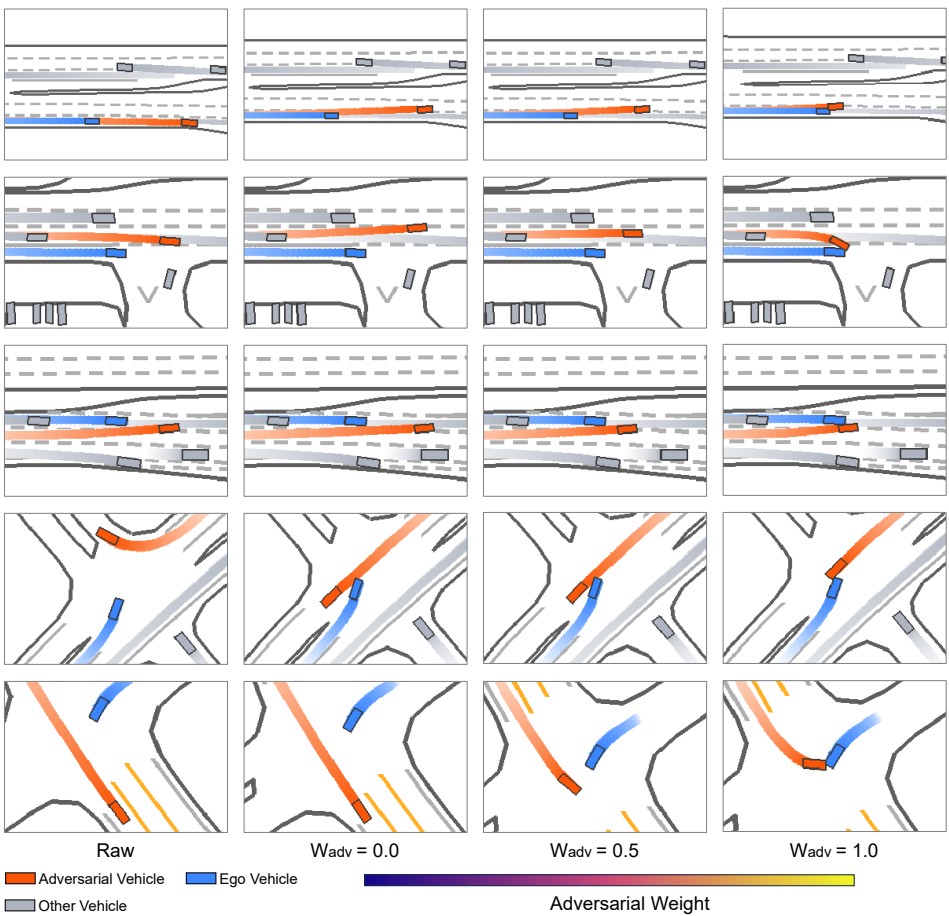

Figure 15: Examples of generated scenarios (SAGE) under different adversarial weights.

