# OpenReview forum: "Steerable Adversarial Scenario Generation through Test-Time Preference Alignment"
_ICLR.cc/2026/Conference — ICLR 2026 Poster_

### Official Review · Reviewer_jZqG · 2025-10-29

**Soundness:** 3
**Presentation:** 3
**Contribution:** 3
**Rating:** 6
**Confidence:** 4

**Summary:**

This paper proposes a method to fine-tune a pre-trained motion generation model to generate adversarial scenarios. The post-training would result in two separate expert models excelling at demonstrating realistic behavior and adversarial behavior, respectively. Model merging by linear interpolation then allows us to control the trade-off between realism and the capability of inducing collision. The model shows good performance in the open-loop benchmark, showing a more natural behavior than other baselines with a nearly SOTA collision rate. The following RL experiments show that agents trained with the adversarial scenario generated by the proposed model achieve SOTA in both nominal scenarios and long-tail scenarios, outperforming agents trained in scenarios generated by other baselines. Further study shows the effectiveness of model merging and the post-training method HGPO.

**Strengths:**

1. Long-tail/near-accidental/safety-critical scenario generation is important for real-world robotics applications. In this field, adversarial policy modeling is important, which trades off the realism of the generated trajectories. The method proposed in this paper strikes a good balance between these two contradictory goals and enables controlling the extent to which the adversarial behavior should be.
2. The experiment is comprehensive, and the results indeed demonstrate the usefulness of the proposed method. It shows the method can achieve an SOTA attack success rate while maintaining a good trajectory realism.
3. The model merging and proposed post-training method, HGPO, is validated to be useful and plays the key role of the success of the whole pipeline.
4. Paper writing is easy to understand, and experimental details are well-documented

**Weaknesses:**

In the RL experiment, the RL agents trained in scenarios generated by different methods should be benchmarked in their respective generated test splits as well. For now, SAGE/CAT/Rule-based agents are benchmarked in scenarios generated by only SAGE and log-replay. It makes me wonder whether the superior performance of the SAGE agent comes from overfitting due to the similarity between the training and testing split, as they are all generated by the SAGE method.
It would be better to additionally benchmark all 3 agents in scenarios generated by the CAT and Rule-based method. The best results we can expect are that the SAGE agent shows comparable even better, results than CAT agents on CAT-generated held-out scenarios. But even if it is a bit worse, the good results of SAGE agents on the log-replay benchmark can still demonstrate another benefit of SAGE: maintaining the nominal driving ability of RL trained policies.

**Questions:**

N/A

---

> ### Author Response · Authors · 2025-11-19
> **General response to Reviewer jZqG**
>
> We sincerely thank the reviewer for the positive and thorough summary of our work. We are encouraged by the appreciation of our method's ability to balance adversariality with realism, the comprehensiveness of our experiments, and the clarity of our writing. Your critical feedback regarding the closed-loop evaluation was particularly insightful. We agree that assessing generalization is crucial. Following your suggestion, we have conducted new cross-evaluation experiments, benchmarking all agents on test sets generated by different methods. These new results included in the revised paper fully address your concern by demonstrating the superior generalization of the SAGE-trained agent and significantly strengthening our conclusions. We are grateful for this suggestion, which has substantially improved the rigor of our paper.

---

> ### Author Response · Authors · 2025-11-19
> **Detailed response to Reviewer jZqG**
>
> W: RL evaluation.
>
> This is an excellent and critical point. We sincerely thank you for this suggestion, as it addresses a potential bias in the closed-loop evaluation and has led us to conduct more rigorous experiments that strengthen our conclusions.
>
> We agree with your comment that evaluating an agent only on scenarios generated by its own training method could lead to overfitting, where superior performance might stem from the similarity between training and testing distributions rather than true robustness. To address this concern, we have conducted the **cross-evaluation experiment** you suggested. We benchmarked all trained agents (SAGE, CAT, Rule-based, and Replay) on held-out test sets generated by **both CAT and the Rule-based method**. The new results have been added to **Table 4: Evaluation of Trained RL Policies in Different Adversarial Environments.**
>
> Our new findings are insightful:
> 1.  **In-distribution advantage":** The results of the cross-evaluation confirm the intuition. An agent generally performs best when tested in an environment generated by its own trainer. For instance, the CAT-trained agent achieves the lowest collision rate on CAT-generated scenarios, and the Rule-based agent does so on its own scenarios. This highlights the importance of the cross-evaluation you proposed.
> 2.  **Superior generalization of the SAGE-trained agent:** Crucially, the SAGE-trained agent demonstrates **better generalizability** in this cross-evaluation setting. In both CAT-generated and Rule-based-generated test environments, our agent achieves a safety performance that is highly competitive and ranks second only to the in-distribution agent. More strikingly, the SAGE-trained agent achieves the **highest route completion rate across all test environments**, including those generated by others. This indicates that our agent not only learns robust defensive skills but also better maintains its primary driving function.
> 3.  **Explanation:** We attribute this superior generalization to the core design of SAGE. Unlike methods that generate adversarial examples of a fixed pattern or intensity, SAGE utilizes a dual-axis curriculum that exposes the ego agent to a **continuous spectrum of challenges**, from mildly perturbed to highly aggressive. This forces the agent to learn a more generalizable policy, preventing it from overfitting to a narrow and specific type of adversarial attack. In contrast, agents trained with CAT or Rule-based methods appear to overfit to their respective static adversarial patterns, leading to degraded performance in unfamiliar test distributions.
>
> Thank you again for this invaluable suggestion. These new results provide much stronger evidence for the benefits of SAGE, demonstrating that it not only yields a superior agent but also one with better generalization properties. We have added the following discussions to the main paper in **section 4.4**.
>
>
> ---
> *To further ensure a fair evaluation and assess the risk of overfitting, we conducted a cross-evaluation of the trained RL policies. In this setup, we benchmarked every trained agent on held-out test sets generated by the CAT and Rule-based methods.
> The results in Tab.4 provide several key insights. First, an agent tends to achieve the best performance on the adversarial metric when evaluated on scenarios created by its own training generator. For example, the CAT-trained agent has the lowest collision rate in CAT-generated scenarios. This validates the importance of this cross-evaluation.
> Second, and more importantly, the SAGE-trained agent demonstrates superior generalizability across all environments. Its safety performance remains highly competitive, ranking a close second. Notably, the SAGE-trained agent achieves the highest route completion rate across all adversarial test sets, demonstrating that it learns to handle challenges more effectively without compromising its primary driving objective. This robustness across diverse adversarial distributions suggests that the curriculum-based training with SAGE, which exposes the agent to a wide spectrum of adversarial intensities, prevents overfitting to a specific attack pattern and enables a more generalizable and practical driving policy. In contrast, agents trained on more static adversarial distributions show a more significant performance drop when tested out-of-distribution.*
>
>
> **Evaluation of Trained RL Policies in Different Adversarial Environments.**
> | Training Methods | Compl. ↑ (CAT-generated scenarios) | Coll. ↓ (CAT-generated scenarios) | Compl. ↑ (Rule-generated scenarios) | Coll. ↓ (Rule-generated scenarios) |
> |:---|:---:|:---:|:---:|:---:|
> | SAGE | **0.678 ± 0.034** | 0.307 ± 0.037 | **0.655 ± 0.037** | 0.334 ± 0.038 |
> | CAT | 0.660 ± 0.039 | **0.298 ± 0.050** | 0.633 ± 0.043 | 0.389 ± 0.044 |
> | Replay (No Adv) | 0.665 ± 0.042 | 0.454 ± 0.051 | 0.642 ± 0.027 | 0.461 ± 0.046 |
> | Rule-based Adv | 0.569 ± 0.050 | 0.348 ± 0.028 | 0.556 ± 0.059 | **0.293 ± 0.043** |

---

> > ### Comment · Reviewer_jZqG · 2025-11-25
> > **Thank you for the response**
> >
> > Thank you for the additional experiment. With this evaluation, the methods are more solid and convincing to be a good new solution to this problem. I would like to raise my score to 8.

---

> > > ### Author Response · Authors · 2025-11-26
> > > **Replying to Official Comment by Reviewer jZqG**
> > >
> > > Dear Reviewer jZqG,
> > >
> > > Thank you very much for your positive feedback and for raising your score. We are especially grateful for your insightful suggestion, which was crucial in strengthening our paper. We sincerely appreciate your support for our work.
> > >
> > >
> > > Best regards,
> > >
> > > The Authors

---

### Official Review · Reviewer_be6Z · 2025-10-30

**Soundness:** 3
**Presentation:** 3
**Contribution:** 3
**Rating:** 6
**Confidence:** 5

**Summary:**

This research addresses limitations in existing adversarial scenario generation methods for autonomous driving, which often lack flexibility to tailor scenarios for specific needs. The authors introduce SAGE (Steerable Adversarial scenario GEnerator), a new framework that allows fine-grained control over the balance between adversariality and realism during scenario creation without retraining. By using preference optimization and interpolating expert policies, SAGE generates more effective training data and improves driving policy performance.

**Strengths:**

* Walking along Pareto frontier between realism and adversarial objectives
* Ability to do it w/o retraining
* Correct (at least looking to be) theoretical statement about the suboptimality gap limited by weighted squared norm of weights difference from above
* Correct (at least looking to be) theoretical statement about the advantage of weight mixing over explicit output ensemble

**Weaknesses:**

* Eq. (8) and lines 304-305 (+ additionally in the Appendix C.3): that's still not clear why the first term - benefit from path flatness - is the dominant one. Would be nice to see any proof, empirical or (better) theoretical one
* It's not clear how preference labels (see "Preference within Feasibility", lines 196-198) are generated
* No ablation on $\delta_{m}$ (line 198); moreover, is it the same as $\sigma = 0.2$ (line 902)?
* Appendix B.2: unclear, what are values of $\omega_{turn}$ and $\omega_{stop-turn}$ in Eq. (13)
* Quite unclear, why the approach is not that useful for IDM in comparison to Replay that makes it questionable to use in practice (Appendix D.1)
* Minor remark: need to define $R_{real}$ earlier (as the negative expectation of $P_{real}$ on page 1373)

Overall, the theoretical concepts brought to autonomous driving really makes the reviewed paper suitable for ICLR conference.

**Questions:**

N/A

---

> ### Author Response · Authors · 2025-11-19
> **General response to Reviewer be6Z**
>
> We are very grateful for your thorough and insightful review. We sincerely appreciate your positive assessment of our work's core strengths, particularly the ability to navigate the Pareto front without retraining and the soundness of our theoretical statements. Your final remark, noting that "the theoretical concepts brought to autonomous driving really makes the reviewed paper suitable for ICLR conference," is especially encouraging.
>
> Your feedback has been invaluable in helping us improve the paper. In response to your comments, we have made significant revisions, which we believe substantially strengthen the paper's theoretical foundations and overall clarity. Specifically, we have:
> - **Justification for weight-space mixing:** To address your primary concern, we have justified our argument with a clearer intuitive explanation, a **new theoretical formalization** that derives the precise conditions for its dominance, and a stronger connection to our direct empirical evidence.
> -   **Clarified details:** We have provided a detailed and step-by-step explanation of our automated preference label generation process.
> -   **Additional ablation studies:** We have conducted a **new ablation study** on the preference margin to validate our hyperparameter choice and have added a **comprehensive hyperparameter table** to the appendix, ensuring all experimental details are transparent.
> -   **More visualizations:** We have offered more visualizations on evaluating against reactive policies like IDM, explaining the results within the standard open-loop benchmark setting.
>
> We are confident that these extensive revisions have made our paper stronger, clearer, and more rigorous. We thank you again for your valuable contributions to improving our work.

---

> ### Author Response · Authors · 2025-11-19
> **Detailed response to Reviewer be6Z**
>
> *W1：Justification of the superiority of weight-space mixing.*
>
> Thank you for this excellent question. We have to admit that our initial explanation was not sufficiently clear. To address this, we have provided a much more rigorous justification in the revised manuscript, combining a clearer intuitive explanation, new theoretical formalization, and direct empirical evidence.
> -  **Clarified intuition.** We have revised the explanatory text in both the main paper (lines 305-313) and Appendix C.3 to be more precise. The superiority of weight mixing requires the loss function to be convex along the interpolation path ($\frac{d^2L^{\text{weight}}}{d\lambda^2} > 0$), which is equivalent to the **reward function being concave**. While LMC posits that experts lie in a common low-loss basin, *this does not imply the path between them has zero curvature*. Instead, we conceptualize this path as a **high-reward ridge**. As shown empirically, moving along this ridge results in a reward trajectory that is concave (i.e., bowed upwards). This concavity is precisely what makes Term 1 negative and dominant, allowing an interpolated model to achieve a reward greater than the linear average of the experts' rewards.
> -  **New theoretical formalization:** To provide a more formal proof, we have added a **new theoretical analysis in Appendix C.3 (Remark 1)**. Under a local quadratic approximation of the loss landscape, we derive the exact condition for Term 1 to dominate. The final condition (Eq. 58) reveals that weight mixing is superior when the intrinsic curvature of the loss basin is greater than the model's output sensitivity along the direction connecting the experts. This condition is readily met in our fine-tuning regime: LMC suggests a low output sensitivity along the solution path, while the conflicting expert objectives guarantee a non-zero and positive curvature for the loss along the trade-off path.
> -  **Empirical evidence:** Our existing results provide strong and direct evidence for this theoretical claim. **Fig. 5(d)** shows that the measured rewards of the interpolated models lie **strictly above the linear interpolation line**. This is the definition of a concave function, which directly validates that the reward landscape has the required geometry, and thus Term 1 is negative. **Fig. 5(c)** shows the linear path between experts traverses a high-reward plateau (the ridge), which is consistent with our explanation. **Fig. 3(a)** shows the final result: the Pareto front traced by our weight mixing strategy consistently dominates that of trajectory mixing, empirically confirming that weight mixing is indeed the superior strategy for our task.
>
> In summary, we have substantially strengthened our paper by providing a clearer intuition, a new theoretical derivation, and a stronger link to our empirical results. This revision justifies why weight-space interpolation is a principled and effective method in our fine-tuning context. We are grateful for your insightful question. We have added the following analysis and discussion.
>
> **In section 3.4**
>
> *For weight mixing to be superior, Term 1 must be negative and dominate Term 2. This requires the loss function to be convex along the interpolation path, i.e., $\frac{d^2L^{\text{weight}}}{d\lambda^2} > 0$.
> This convexity in the loss landscape corresponds to a concave geometry in the reward landscape. Intuitively, a linear path in the weight space traces a curved and concave path in the reward space. This means an interpolated model $\theta(\lambda)$ can achieve a reward greater than the linear average of the experts' rewards. While LMC posits that both experts reside in a common low-loss basin, this does not imply zero curvature. Instead, the path connecting them acts as a high-reward ridge (Fig. 6(c)). The concavity of this ridge provides the necessary positive curvature in the loss, making Term 1 negative and dominant. This phenomenon is empirically validated by the concave reward curves shown in Fig.6(d), which lie strictly above the linear interpolation line.*
>
> ---
> **In section C.3**
>
> *LMC suggests that solutions fine-tuned from a common pretrained model lie within a wide and low-loss basin. However, the path connecting two distinct experts is not perfectly zero-curvature. Instead, it can be conceptualized as a high-reward ridge.
> As the model traverses this path from one expert to another, the combined reward often has an arc-like shape, which signifies reward concavity. The reward landscape in Fig. 6(c) illustrates this high-reward plateau, and the measured performance in Fig. 6(d) provides direct empirical evidence of this concavity, showing that the rewards of interpolated models are consistently higher than a linear combination of the endpoint rewards. This inherent concavity of the reward path ensures the loss is convex, making Term 1 negative and significant, thereby providing a principled reason for the superiority of weight-space mixing in the fine-tuning regime.*

---

> ### Author Response · Authors · 2025-11-19
> **Detailed response to Reviewer be6Z**
>
> *Continued response to W1*
>
> **In section C.3, Remark 1, Appendix**
>
>
> *To help better understand Eq. (57), in the following, we provide an intuitive illustration to formalize the exact condition where Term 1 can dominate.*
>
> *[Remark on the Dominance of Loss Curvature Benefit.]
> Based on Eq.(57), weight mixing is superior if $\mathbb{E}\big[\|\Delta f(x)\|^2\big] < \frac{d^2{L}_{\text{weight}}}{d\alpha^2}$. To make this condition more intuitive, we formalize it under a local quadratic approximation of the loss landscape. Similar to section C.1, we assume that in the local vicinity of the expert solutions $\theta_1$ and $\theta_2$, the loss function ${L}(\theta)$ can be approximated by a quadratic form with ${L}(\theta) \approx \frac{\eta}{2} \|\theta - \theta^*\|^2 + C$,
> where $\theta^*$ is the minimum of the loss basin, $\eta > 0$ is a scalar representing the curvature of the basin, and $C$ is a constant.*
>
> *The right-hand side represents the curvature of the loss along the linear path $\theta_{\alpha} = (1-\alpha)\theta_1 + \alpha\theta_2$. We can compute its second derivative with respect to $\alpha$:*
>
> $$
> \frac{d{L} _ {\text{weight}}}{d\alpha} = \nabla {L}(\theta_{\alpha})^\top \frac{d\theta_{\alpha}}{d\alpha} = \eta (\theta_{\alpha} - \theta^\ast)^\top (\theta_2 - \theta_1),
> $$
>
> $$
> \frac{d^2{L}_{\text{weight}}}{d\alpha^2} = \eta (\theta_2 - \theta_1)^\top (\theta_2 - \theta_1) = \eta \|\theta_2 - \theta_1\|^2.
> $$
>
> *The left-hand side is the expected squared difference in outputs. We can approximate this difference using a first-order Taylor expansion of the model function $f(x; \theta)$ around $\theta_1$:*
> $$
>     \Delta f(x) = f(x; \theta_2) - f(x; \theta_1) \approx J(x; \theta_1)(\theta_2 - \theta_1),
> $$
> *where $J(x; \theta) = \frac{\partial f}{\partial \theta}$ is the Jacobian of the model's output with respect to its parameters. The expected squared difference is then:*
>
> $$
>     \mathbb{E}[|\Delta f(x)\|^2] \approx \mathbb{E}[|J(x; \theta_1)(\theta_2 - \theta_1)|^2]
> $$
>
> $$
>     = (\theta_2 - \theta_1)^\top \mathbb{E}\big[J(x; \theta_1)^\top J(x; \theta_1)\big] (\theta_2 - \theta_1)
> $$
> $$
>     = (\theta_2 - \theta_1)^\top F (\theta_2 - \theta_1),
> $$
> *where $F$ is the Fisher information matrix, which measures the sensitivity of the model's output to changes in its parameters. By substituting these results back, we obtain the final condition:*
> $$
>     (\theta_2 - \theta_1)^\top F (\theta_2 - \theta_1) < \eta \|\theta_2 - \theta_1\|^2,
>     \Rightarrow \eta >\frac{(\theta_2 - \theta_1)^\top F (\theta_2 - \theta_1)}{\|\theta_2 - \theta_1\|^2}.
> $$
>
> *This derivation yields an intuitive condition: weight mixing is superior whenever the intrinsic curvature of the loss landscape exceeds the sensitivity of the model's output in the direction connecting the experts. This condition is readily satisfied in our fine-tuning setting. On one hand, LMC implies that fine-tuned models develop robust representations, making their outputs relatively insensitive to parameter changes along the solution path. On the other hand, because the experts are trained on conflicting objectives, the path connecting them has a non-zero curvature ($\eta > 0$) to reflect the trade-off in rewards, which provides the necessary signal for alignment.*
>
> *This theoretical insight is directly justified by our empirical findings in Fig.6. The superiority of weight mixing requires the reward function to be concave along the interpolation path. Fig.6(d) provides direct visual proof, showing the measured reward curves of interpolated models lying strictly above the linear interpolation line. This confirms that the reward landscape possesses the required curvature. This view also reconciles with LMC: the path connecting experts is not a completely flat plateau but a high-reward ridge (Fig.6(c)). While LMC ensures the solution remains on this low-loss ridge, our analysis shows that it is the ridge's concave curvature that allows weight-space interpolation to discover superior trade-offs.*

---

> ### Author Response · Authors · 2025-11-19
> **Detailed response to Reviewer be6Z**
>
> *W2: Preference labels.*
>
> Thank you for raising this question. To be precise, the preference labels in our work are produced through a **fully automated and principled process** governed by a predefined preference reward function $R_\text{pref}$. The generation process for preference pairs $(\tau^w, \tau^l)$ for a given scenario follows these steps:
>
> 1.  **Group-wise sampling:** We first sample a group of N candidate trajectories ${\tau_i}$ from our generative policy.
>
> 2.  **Feasibility filtering:** As outlined in the "Feasibility First" principle, we partition the sampled trajectories into a feasible set $G^{feas}$ and an infeasible set $G^{infeas}$ based on hard map-compliance constraints $F(\tau, M)$. This immediately generates one type of preference pair: any trajectory from $G^{feas}$ is strictly preferred over any from $G^{infeas}$.
>
> 3.  **Preference scoring within feasibility:** This directly addresses the reviewer's query. For all trajectories **within the feasible set $G^{feas}$**, we compute a quantitative score for each one using our preference reward function: $R_{pref}(\tau; c) = w_{adv} * R_{adv}(\tau, \tau_{ego}; c) – w_{real} * P_{real}(\tau)$. The detailed mathematical formulations for $R_{adv}$ and $P_{real}$ are provided in Appendix B.2.
>
> 4.  **Pair generation:** We then form preference pairs $(\tau^w, \tau^l)$ by comparing the $R_{pref}$ scores of trajectories within $G^{feas}$. A pair is created if $R_{pref}(\tau^w) > R_{pref}(\tau^l) + \delta_m$, where the margin $\delta_m$ ensures that the model learns from meaningful and non-noisy preferences.
>
> This automated labeling process is highly efficient and scalable, avoiding the prohibitive cost of human feedback or labelling process. Moreover, it provides a flexible mechanism to inject distinct expertise into our models by simply adjusting the weights $(w_{adv}, w_{real})$ in $R_{pref}$, which is crucial for training the opposing experts that enable our test-time steerability.
>
> ---
> W3: Ablation study.
>
> Thank you for your valuable feedback regarding the hyperparameter $\delta_m$. First, to address the notational inconsistency, the $\delta_m$ in line 198 and $\sigma$ in line 902 (now line 1026) refer to the exact same hyperparamete. We have now corrected this throughout the manuscript to consistently use $\delta_m$ for clarity.
> Second, in response to your suggestion, we have conducted a new ablation study to analyze the sensitivity of $\delta_m$. We have added the results to **Appendix D.3** and included a new figure (**Figure 10: Ablation studies on reward margin** in the appendix). We believe these results provide a comprehensive answer and strengthen the paper. We have added the following discussions to the Appendix:
>
>
> *Furthermore, Fig. 10 investigates the impact of the reward margin $\delta_m$. Fig. 10(a) shows that as $\delta_m$ increases, the number of generated preference pairs per scenario decreases, as a larger reward difference is required to form a valid pair. More importantly, Fig. 10(b) illustrates the impact on learning performance. The setting with $\delta_m=0$ results in the poorest performance, confirming that learning from pairs with insignificant reward differences introduces noise and impedes the training process. Conversely, setting a moderate margin ($\delta_m$ in [0.2, 1.0]) significantly improves performance by focusing the model on more distinct and informative preferences. The model shows robust performance within this range, indicating that it is not overly sensitive to this hyperparameter. We selected $\delta_m=0.2$ for our main experiments as it empirically provided the best balance between sample efficiency and final performance. An overly large margin, such as $\delta_m=2.0$, can slightly degrade performance by filtering out too many useful training samples.*

---

> ### Author Response · Authors · 2025-11-19
> **Detailed response to Reviewer be6Z**
>
> *W4: Hyperparameters*
>
> Thank you for pointing out this omission. We apologize for not specifying the values for \omega_{turn} and  \omega_{stop-turn} in Eq. (13). These are weighting factors within our behavioral penalty formulation, designed to penalize unnatural maneuvers. In our experiments, we used the following fixed values: (1) \omega_{turn} was set to 5.0 This term penalizes trajectories with an excessive total change in heading; (2)  \omega_{stop-turn} was set to 3.0. This term specifically targets unrealistic spinning-in-place behaviors by penalizing high angular velocity at low speeds.
> We have updated Appendix B.2 to explicitly state these values and have also included them in a new comprehensive hyperparameter table (**Table 7: Hyperparameters for reward functions and HGPO**) to improve the reproducibility of our work. We appreciate you bringing this to our attention.
>
> **Hyperparameters for reward functions and HGPO.**
> | **Category** | **Hyperparameter** | **Value** |
> |:---|:---|:---|
> | HGPO Fine-tuning | Optimizer | AdamW |
> | | Learning Rate | $1.0 × 10^{-5}$ |
> | | Training Epochs | 200 |
> | | Batch Size | 1 (scenario) |
> | | Group Size (N) | 32 |
> | | Max Pairs per Group (K) | 8 |
> | | Preference Margin ($\delta_m$) | 0.2 |
> | | Beta (β) | 0.05 |
> | Preference Reward | Adversarial Expert ($\pi_{\theta_{adv}}$) | |
> | | $w_{adv} / w_{real}$ | 0.9 / 0.1 |
> | | Realism Expert ($\pi_{\theta_{real}}$) | |
> | |$ w_{adv} / w_{real}$ | 0.1 / 0.9 |
> | $R_{adv}$ Components | Collision Reward Scale | 10.0 |
> | | Proximity Reward Scale | 1.0 |
> | | Proximity Decay Rate | 0.2 |
> | P_{real} Components | $w_{turn}$ | 5.0 |
> | | $w_{stop-turn}$ | 3.0 |
> | | Kinematic Penalty Factor (Accel) | 5.0 |
> | | Kinematic Penalty Factor (Ang. Vel.) | 5.0 |
> | | Accel. Comfort Zone | $7.0 m/s^2$ |
> | | Lat. Accel. Comfort Zone | $6.0 m/s^2$ |
> | | Ang. Vel. Comfort Zone | 0.8 rad/s |
> | | Max Reasonable Turn | π radians |
> | Map Feasibility | Cross Solid Line Penalty | 50.0 |
> | | Crash Object Penalty | 10.0 |
>
>
> ---
> *W5: Adversarial generation against reactive policies.*
>
> Thank you for this observation regarding the performance against the IDM policy. This is an excellent question that highlights a crucial aspect of the standard open-loop benchmark setting and the nature of safety-critical events.
> 1.  **The nature of long-tail events:** Safety-critical events are rare and lie in the long tail of the data distribution. An intelligent and reactive policy like IDM or a well-trained RL agent is designed to handle the vast majority of common traffic situations. Therefore, it is expected that the absolute attack success rate against such competent policies will be lower than against a non-reactive Replay policy. The goal is not to find an adversary that defeats IDM 100% of the time, but to find the rare yet plausible scenarios. SAGE excels at this by generating realistic corner cases. While having a lower absolute success rate, they represent more valuable vulnerabilities. To better illustrate this, we have added qualitative examples of successful attacks against **different reactive policies in the revised appendix (Figure 15: Examples of SAGE-generated adversarial behaviors against different ego policies.)**, demonstrating that SAGE indeed finds meaningful and challenging scenarios for these reactive agents.
> 2.  **Limitations of the open-loop benchmark:** It is critical to consider these results within the **open-loop evaluation protocol** used in our work and prior literature (e.g., Zhang et al., 2023). In this setup, the adversarial trajectory is pre-calculated based on the ego agent's *initial* plan (from the Replay policy) and then executed without adaptation. A reactive policy like IDM or RL agent can then observe this pre-planned adversarial maneuver and adjust its own behavior to mitigate the threat. The adversary, however, does not get to re-plan in response to the ego's reaction. This inherent disadvantage for the generator naturally leads to lower success rates against reactive policies.
> 3.  **Potential closed-loop settings:** A straightforward engineering extension to boost the attack rate would be to allow the adversary to regenerate its plan at a high frequency, creating a closed-loop interaction between the adversary and the ego agent. This would enable the adversary to adapt to the ego's defensive maneuvers. However, such a setup goes beyond the scope of the established benchmark used in this paper. We believe that integrating SAGE into a truly closed-loop attack setting is a promising direction for future work and would likely demonstrate even higher effectiveness.
>
> ---
> *W6: Variable definition.*
>
>  We apologize for any confusion caused. We have revised the manuscript to introduce these definitions explicitly at the beginning of our theoretical analysis. We have formally defined $R_{adv}(θ) = E_{τ∼π_θ}[R_{adv}(τ)]$ and $R_{real}(θ) = E_{τ∼π_θ}[-P_{real}(τ)]$ in General Assumptions, Assumption 1 for better readability.

---

> ### Comment · Reviewer_be6Z · 2025-11-19
>
> Dear authors, thank you a lot for your comments and additional clarifications. While I'm not fully convinced in terms of finding\using the specific $\eta$ that provides the majorization of path flatness, and looking at Fig. 15 to me is quite hard to understand the reactiveness of the technique, I'd like to say that the amount of work done is significant and deserves a good mark.

---

> ### Author Response · Authors · 2025-11-19
> **Replying to Official Comment by Reviewer be6Z**
>
> Dear Reviewer be6Z,
>
> We are deeply grateful for your prompt response and for taking the time to carefully review our rebuttal. We sincerely appreciate your recognition of the effort we put into this work and are greatly encouraged by your decision to raise the score.
>
> Regarding your remaining queries, we would like to provide a brief clarification:
>
> 1. We wish to clarify that $\eta$ is not a hyperparameter that we manually tune or search for. Instead, it is used to represent the intrinsic geometry of the loss landscape resulting from training on conflicting objectives. Our derivation aims to prove why weight mixing works: as long as this intrinsic curvature (reward concavity) dominates the model's output sensitivity, which is empirically verified by the upward-bowing reward curves in Fig. 6(d), weight mixing is theoretically superior to ensembling.
> 2. We apologize if the visualization was difficult to interpret. Fig. 15 is designed to showcase successful attacks generated by SAGE against reactive ego agents (such as IDM and RL policies), which actively try to avoid collisions. The figure demonstrates that SAGE can discover specific and subtle maneuvers that effectively trick these capable agents into failure, proving that our method generates meaningful corner cases even for reactive systems. We will include a video demonstration to illustrate this process better.
>
> We will ensure that these points are explicitly clarified in the final version of the manuscript to further improve readability.
>
> Thank you once again for your constructive feedback and valuable support of our work.
>
> Best regards,
>
> The Authors

---

> > ### Comment · Reviewer_be6Z · 2025-11-20
> >
> > Thanks! Got your point.

---

### Official Review · Reviewer_ZEJW · 2025-11-02

**Soundness:** 3
**Presentation:** 3
**Contribution:** 2
**Rating:** 4
**Confidence:** 3

**Summary:**

The paper reframes adversarial scenario generation for autonomous driving as a multi-objective preference alignment problem and proposes SAGE. Contributions are: (i) Hierarchical Group-based Preference Optimization (HGPO), which decouples hard feasibility (map compliance) from soft preferences (adversariality vs. realism) by gating infeasible samples with a binary feasibility function $F(\tau,\mathcal M)$ and learning from group-wise, multi-pair preferences within the feasible set to improve sample efficiency; (ii) training two expert policies from a shared pretrained model (an adversarial-leaning expert and a realism-leaning expert) and interpolating their weights at test time to steer along the trade-off; and (iii) integrating SAGE into an existing closed-loop RL framework.

**Strengths:**

SAGE provides test-time steerability across the adversariality–realism trade-off, and handles feasibility by treating map rules as hard constraints. Empirically, in open-loop benchmarking, SAGE traces a better realism for a given adversarial reward and achieves lower map-violation penalties while remaining competitive in attack metrics; figures and tables exemplify smooth control as the adversarial weight increases.

**Weaknesses:**

1. **Novelty concerns:** The core ideas (i.e., preference alignment and weight-space interpolation between experts) may feel incremental, which explicitly build on prior literature. Please emphasize the novelty of SAGE. The key idea includes filtering out map-infeasible samples (F=0) from the preference set, and also constructing group-wise, multi-pair preferences within the feasible set and optimizing a reference-regularized HGPO objective. That said, the contribution remains incremental within the existing preference-optimization paradigm rather than a fundamentally new algorithmic framework.

2. **Generation Efficiency:**  Against the Replay policy, SAGE and its variations report lower attack success rates and lower rewards than their baselines, raising questions about the effectiveness of the generated adversarial behaviors.

3. **Closed-loop result discrepancy and fairness:** In the paper’s closed-loop results, CAT underperforms Replay (no adv) on completion/collision, whereas the original CAT paper typically reports gains from adversarial training. Please explain why CAT underperforms here and provide diagnostics to ensure a well-tuned baseline. Besides, CAT’s closed-loop results here appear worse than those reported in the original paper in terms of collision and completion; this gap should be reconciled as well.

**Questions:**

1.  Baselines include Rule, CAT, KING, AdvTrajOpt, and SEAL, but omit GOOSE (2024) from related work; Is there a reason why it doesn't appear in the experiments as a baseline method?

2. I wonder how diverse generated adversarial behaviors are. Could authors give visualizations or quantitative results on this perspective?

3. please address issues in point 3. in weakness.

4. What is the expert reward/penalty design (component weights, margins, clipping constants, and final thresholds actually used in experiments)? What is the full pre-training recipe for the shared reference model (e.g., optimizer, learning-rate schedule, batch size, training duration/epochs, data splits and augmentations) used for DenseTNT, or whether a public checkpoint was used? for HGPO fine-tuning, what are the concrete values for group size (N), pair-selection margin ($\delta_m$), the number of candidates (K) per context, and any additional regularization or gradient-scaling coefficients?

---

> ### Author Response · Authors · 2025-11-19
> **General response to Reviewer ZEJW**
>
> We thank the reviewer for their detailed summary and constructive feedback on our work. We appreciate the acknowledgment of SAGE's core strengths, including its test-time steerability, principled handling of feasibility constraints, and strong empirical performance in open-loop benchmarks.
>
> We understand that the primary concerns revolve around the novelty of our contributions and the fairness of our closed-loop experimental comparisons. We have taken these points very seriously and have made substantial revisions to the manuscript to address them comprehensively. Our detailed responses aim to demonstrate that **SAGE represents a significant and novel contribution to the field, rather than an incremental one**. Specifically, our rebuttal and revisions will clarify:
> -   **Our core novelty:** We will elaborate on how SAGE introduces a **new paradigm** to adversarial scenario generation by being **the first to provide a theoretically-grounded framework for test-time, Pareto-optimal control**, which is a contribution rooted in our novel application and validation of the LMC hypothesis in this domain.
> -   **Adversarial effectiveness:** We re-contextualize the definition of "adversarial effectiveness," arguing that SAGE's strength lies in generating **meaningful and plausible** failures, a crucial distinction from simply maximizing collision rates.
> -   **New benchmarks:** To address the concerns about our closed-loop evaluation, we have conducted new **cross-evaluation experiments**, benchmarking all trained agents on test sets generated by different methods. These new results provide a much fairer and more insightful comparison, directly addressing the potential for overfitting and reconciling performance discrepancies with prior work.
> -   **Baselines and clarity:** We have also added the recently published GOOSE as a new baseline and added a detailed hyperparameter table to provide full information about hyperparameters..
>
> We believe that our detailed responses and the significant additions to the paper will fully address the reviewer's concerns and more clearly articulate the novelty and impact of our contributions. We look forward to your new feedback and further discussion.

---

> ### Author Response · Authors · 2025-11-19
> **Detailed Response to Reviewer ZEJW**
>
> *W1: Contribution and novelty.*
>
> Thank you for your thoughtful feedback on the novelty of our work. **While we agree that our framework builds upon the foundations of preference alignment from recent literature, we respectfully disagree with the characterization of our contribution as incremental**. We argue that **SAGE introduces a new paradigm to the field of adversarial scenario generation**, supported by novel theoretical insights and practical capabilities, which we would like to elaborate on.
> *  **A new paradigm for controllable scenario generation:** To the best of our knowledge, SAGE is the **first framework to reframe adversarial scenario generation as a test-time preference alignment problem**. As highlighted in our introduction, existing scenario generators are either inflexible, producing a single trade-off and requiring costly retraining for any adjustment, or they offer limited control via conditioning/guidance without any guarantees of optimality. SAGE is the first to provide a **principled and theoretically-grounded method for test-time and fine-grained control**, enabling users to smoothly navigate the entire Pareto front of adversariality and realism without retraining. This **shift from static generation to dynamic and steerable generation** represents a conceptual leap for this field.
> *  **New theoretical contributions:** A major contribution of our paper, which we believe was understated, is being the **first to introduce and validate the Linear Mode Connectivity (LMC) hypothesis in the context of motion generation for autonomous driving**. While we leverage the preference optimization framework, our key theoretical novelty lies in using LMC to **justify why** simple weight interpolation is so effective for this specific problem. We provide rigorous theoretical analysis (Theorem 1, Proposition 1) and extensive empirical evidence (Section 4.3, Figure 5) to ground this paradigm. Our analysis reveals novel insights, which are specific to our problem domain and not just a direct application of prior work. As the reviewers noted, *"Authors show theoretical and empirical applicability of LMC to traffic simulation problems,"* and *"the theoretical concepts brought to autonomous driving really makes the reviewed paper suitable for ICLR conference."*
> *  **Bridging new techniques:** Beyond the proposed new preference optimization method HGPO, our work **connects steerable generation to recent advances in large model editing**, specifically **task vector arithmetic** (Section 3.3, Appendix B.3). We demonstrate both theoretically and empirically (Fig. 6e, Fig. 11) that this perspective not only provides a deeper understanding of interpolation but also enables **principled extrapolation**. This allows users to generate out-of-distribution scenarios that are even more adversarial or realistic than the trained experts, which is a new capability that is absent in prior methods.
>
> In summary, while components like HGPO are within the preference optimization space, the core novelty of SAGE lies in its holistic framework: **(1) reframing the problem**, **(2) providing the first test-time steerable generator with theoretical Pareto-front navigation guarantees**, and **(3) grounding this capability in the LMC hypothesis with novel theoretical and empirical insights**. We believe these contributions, also recognized by other reviewers, collectively position SAGE as a new framework, rather than an incremental addition.

---

> ### Author Response · Authors · 2025-11-19
> **Detailed Response to Reviewer ZEJW**
>
> *W2: Attack effectiveness.*
>
> Thank you for raising this point about the evaluation of adversarial generation. However, we argue that the ultimate goal of adversarial generation is **not merely to maximize collisions, but to discover meaningful and plausible vulnerabilities** that can be used for robust testing and effective closed-loop training.
> -  **Attack rate is not necessarily the higher the better:** As evidenced by the "Rule" baseline in Tables 1, 2, 6, and 7, achieving a high attack rate (even 100% in Table 1) is trivial if one disregards realism, map compliance, and kinematic feasibility. Such methods often generate physically implausible scenarios that offer little value. They do not represent events that could occur in the real world and thus cannot be used to train a driving policy to handle realistic long-tail events. An adversarial scenario is only useful if it represents a plausible failure mode of the ego agent.
> -  **Downstream effectiveness:** The true measure of an adversarial generator's effectiveness lies in its utility for downstream tasks. Scenarios that are unrealistically aggressive or physically impossible are often unlearnable or teach the ego agent to be overly conservative in benign situations. As demonstrated in closed-loop training experiments (**Section 4.4, Tabs 3,4**), the scenarios generated by SAGE lead to a **more robust driving policy**, which confirms the true effectiveness of our approach.
> -  **SAGE achieves superior trade-off:** One of our contribution with SAGE is achieving a **superior trade-off between adversariality and realism**. While baselines like CAT and Rule-based methods achieve higher attack rates, they do so at the cost of extremely high realism and map violation penalties (e.g., in Table 1, CAT's Cross Line penalty is 9.078, vs. SAGE's 1.084). SAGE maintains a competitive attack success rate while achieving the **highest realism and map compliance across all methods by a significant margin**. This means SAGE discovers challenging scenarios that are not just aggressive, but also coherent, physically plausible, and obey traffic rules. These are valuable and corner cases needed for meaningful safety validation.
>
> In summary, we contend that SAGE is highly effective because it shifts the objective from collision to meaningful failure discovery, which is more aligned with the practical goals of safety evaluation.

---

> ### Author Response · Authors · 2025-11-19
> **Detailed Response to Reviewer ZEJW**
>
> *W3：Closed-loop evaluation results.*
>
> Thank you for raising this question regarding the closed-loop results and the fairness of our comparison with the CAT baseline. We have thoroughly investigated this issue and would like to offer a comprehensive explanation:
> -  **Different scenario subsets:** A primary source of variation is the dataset used for training and evaluation. As stated in our paper (lines 929-930), we use a **filtered subset of 370 scenarios** from the WOMD dataset. This subset was curated to exclude trivial or invalid cases (e.g., scenarios with overly short interactions or initial map violations) to ensure a high-quality evaluation benchmark for all methods. The original CAT paper used a different, larger set of 500 scenarios. This difference in the dataset leads to variations in the final trained performance.
> -  **Training and testing environments:** This is a crucial point. The main results in our paper (Table 3) evaluate all agents on a test set **generated by SAGE (with `w_adv=1.0`)**. This was done to assess robustness against the most challenging scenarios our method could produce. However, **the results in the original CAT paper were measured using CAT-generated evaluation scenarios**. This may create a potential distribution gap. The CAT-trained agent, having been optimized specifically for scenarios generated by CAT's probabilistic sampling logic, may overfit to that particular adversarial pattern. Its lower performance on SAGE-generated scenarios likely reflects a generalization gap when faced with a different and potentially more diverse adversarial distribution. This overfitting tendency is also observed with the Rule-based agent.
> -  **New cross-evaluation benchmark:** To fully address your concern and provide a fair comparison, we have conducted additional **cross-evaluation experiments**. We now also evaluate all agents on test sets generated by **CAT and the Rule-based method** (**Table 4: Evaluation of Trained RL Policies in Different Adversarial Environments**, which is also provided below). We believe this is the fairest way to benchmark performance. As expected, the **CAT-trained agent performs much better on the CAT-generated test set**, and its collision/completion rates in this setting are now much more aligned with the results reported in the original CAT paper (with remaining differences attributable to the scenario subset and randomness).
>
> Interestingly, these new experiments revealed a nuance of the MetaDrive environment setup: **the agent trained only on Replay data often achieves a high completion rate but at the cost of the highest collision rate**. We found that during training, **episodes continue even after a collision**, allowing the agent to still accumulate route completion rewards. However, during evaluation, **episodes terminate immediately upon collision**. This discrepancy incentivizes the Replay-trained agent to prioritize completion, even if it entails risky behavior. This is a subtle but important finding that helps explain the observed performance.
>
> In summary, the initial discrepancy is a compound effect of different scenario sets, different adversarial environments, and inherent randomness. Our new cross-evaluation provides a much clearer and fairer comparison.
>
> **Evaluation of Trained RL Policies in Different Adversarial Environments.**
> | Training Methods | Compl. ↑ (CAT-generated scenarios) | Coll. ↓ (CAT-generated scenarios) | Compl. ↑ (Rule-generated scenarios) | Coll. ↓ (Rule-generated scenarios) |
> |:---|:---:|:---:|:---:|:---:|
> | SAGE | **0.678 ± 0.034** | 0.307 ± 0.037 | **0.655 ± 0.037** | 0.334 ± 0.038 |
> | CAT | 0.660 ± 0.039 | **0.298 ± 0.050** | 0.633 ± 0.043 | 0.389 ± 0.044 |
> | Replay (No Adv) | 0.665 ± 0.042 | 0.454 ± 0.051 | 0.642 ± 0.027 | 0.461 ± 0.046 |
> | Rule-based Adv | 0.569 ± 0.050 | 0.348 ± 0.028 | 0.556 ± 0.059 | **0.293 ± 0.043** |

---

> ### Author Response · Authors · 2025-11-19
> **Detailed Response to Reviewer ZEJW**
>
> *Q1: New baseline.*
>
> We sincerely thank the reviewer for bringing the very recent work, GOOSE (Ransiek et al., 2024), to our attention. This is indeed a highly relevant piece of related work, and we apologize for its omission in our initial submission.
> We have now included GOOSE as a new baseline in our experimental comparisons. The results have been added to **Tabs. 1 and 2 in the main paper, and Tabs. 8 and 9 in the appendix**. To ensure a fair and rigorous comparison, we utilized the **official pretrained GOOSE checkpoint** for WOMD,  which was provided in the public repository of another recent work (SEAL). We followed the authors' recommended hyperparameters for evaluation. We have also updated the baseline descriptions in **Section B.1 Experimental setups, Appendix** to include a discussion of GOOSE.
> We briefly provide the newly added results below.
>
> **Evaluation of adversarial generation methods against the Replay policy.**
> | **Methods** | **Attack Succ. Rate** ↑ | **Adv. Reward** ↑ | **Behav. (Real. Pen. ↓)** | **Kine. (Real. Pen. ↓)** | **Crash Obj. (Map Comp. ↓)** | **Cross Line (Map Comp. ↓)** | **Accel. (WD)** | **Vel. (WD)** | **Yaw (WD)** |
> |:---|:---:|:---:|:---:|:---:|:---:|:---:|:---:|:---:|:---:|
> | Rule | 100.00% | 5.048 | 2.798 | 5.614 | 1.734 | 7.724 | 2.080 | 8.546 | 0.204 |
> | CAT  | 94.85% | 3.961 | 8.941 | 3.143 | 2.466 | 9.078 | 1.556 | 7.233 | 0.225 |
> | KING  | 40.85% | 2.243 | 5.883 | 3.434 | 3.126 | 6.056 | 0.972 | 255.5 | 0.098 |
> | AdvTrajOpt  | 70.46% | 2.652 | 4.500 | 2.775 | 2.547 | 10.16 | 1.754 | 6.177 | 0.268 |
> | SEAL  | 63.93% | 1.269 | 3.017 | 2.423 | 2.732 | 11.612 | 1.544 | 6.959 | 0.202 |
> | GOOSE  | 36.07% | 2.378 | 4.718 | 21.32 | 4.426 | 14.48 | 6.368 | 8.286 | 0.154 |
> | SAGE ($w_{adv=1.0}$) | 76.15% | 4.121 | 1.429 | 2.479 | 0.731 | 1.084 | 2.098 | 8.088 | 0.184 |
>
> **Evaluation of adversarial generation methods against the RL policy.**
> | **Methods** | **Attack Succ. Rate** ↑ | **Adv. Reward** ↑ | **Behav. (Real. Pen. ↓)** | **Kine. (Real. Pen. ↓)** | **Crash Obj. (Map Comp. ↓)** | **Cross Line (Map Comp. ↓)** | **Accel. (WD)** | **Vel. (WD)** | **Yaw (WD)** |
> |:---|:---:|:---:|:---:|:---:|:---:|:---:|:---:|:---:|:---:|
> | Rule | 65.57% | 2.761 | 2.180 | 113.7 | 1.803 | 6.148 | 10.85 | 13.47 | 0.336 |
> | CAT  | 30.33% | 1.319 | 8.191 | 3.039 | 2.623 | 6.967 | 1.539 | 8.877 | 0.187 |
> | KING | 19.14% | 1.148 | 2.041 | 2.596 | 3.114 | 5.857 | 0.983 | 259.1 | 0.097 |
> | AdvTrajOpt  | 19.40% | 0.992 | 4.542 | 2.779 | 2.459 | 9.973 | 1.749 | 6.187 | 0.269 |
> | SEAL  | 31.40% | 0.752 | 5.871 | 2.684 | 3.030 | 11.98 | 1.563 | 8.267 | 0.267 |
> | GOOSE  | 12.46% | 0.667 | 4.369 | 15.47 | 3.507 | 11.45 | 4.662 | 8.070 | 0.152 |
> | SAGE ($w_{adv=1.0}$) | 28.42% | 1.400 | 1.468 | 2.496 | 0.792 | 1.366 | 2.098 | 8.114 | 0.188 |
>
> **Evaluation of adversarial generation methods against the Rule-based/Autopilot policy.**
> | **Methods** | **Attack Succ. Rate** ↑ | **Adv. Reward** ↑ | **Behav. (Real. Pen. ↓)** | **Kine. (Real. Pen. ↓)** | **Crash Obj. (Map Comp. ↓)** | **Cross Line (Map Comp. ↓)** | **Accel. (WD)** | **Vel. (WD)** | **Yaw (WD)** |
> |:---|:---:|:---:|:---:|:---:|:---:|:---:|:---:|:---:|:---:|
> | Rule | 24.18% | 0.724 | 2.623 | 111.184 | 1.658 | 6.386 | 7.631 | 10.572 | 0.248 |
> | CAT | 18.21% | 0.659 | 8.346 | 3.071 | 2.799 | 9.647 | 1.565 | 7.781 | 0.201 |
> | KING  | 16.43% | 0.558 | 2.445 | 2.565 | 3.116 | 5.807 | 0.956 | 256.921 | 0.097 |
> | AdvTrajOpt  | 19.00% | 0.543 | 4.445 | 2.818 | 2.567 | 10.833 | 1.753 | 6.207 | 0.271 |
> | SEAL  | 19.45% | 1.584 | 4.902 | 2.584 | 3.260 | 9.178 | 1.494 | 6.756 | 0.248 |
> | GOOSE  | 5.10% | 0.325 | 2.387 | 14.56 | 4.363 | 14.31 | 5.322 | 8.107 | 0.167 |
> | SAGE ($w_{adv=1.0}$) | 19.84% | 0.749 | 1.430 | 2.480 | 0.788 | 1.087 | 2.100 | 8.100 | 0.185 |
>
> **Evaluation of adversarial generation methods against the IDM policy.**
> | **Methods** | **Attack Succ. Rate** ↑ | **Adv. Reward** ↑ | **Behav. (Real. Pen. ↓)** | **Kine. (Real. Pen. ↓)** | **Crash Obj. (Map Comp. ↓)** | **Cross Line (Map Comp. ↓)** | **Accel. (WD)** | **Vel. (WD)** | **Yaw (WD)** |
> |:---|:---:|:---:|:---:|:---:|:---:|:---:|:---:|:---:|:---:|
> | Rule | 60.54% | 2.763 | 4.833 | 36.947 | 1.973 | 6.856 | 4.466 | 8.447 | 0.309 |
> | CAT  | 43.48% | 1.746 | 10.275 | 3.207 | 3.345 | 8.696 | 1.595 | 7.550 | 0.231 |
> | KING  | 16.48% | 1.146 | 2.451 | 2.566 | 3.097 | 5.824 | 0.955 | 257.619 | 0.097 |
> | AdvTrajOpt | 20.44% | 0.963 | 4.528 | 2.797 | 2.561 | 10.354 | 1.750 | 6.173 | 0.270 |
> | SEAL  | 33.70% | 0.942 | 5.785 | 2.719 | 3.562 | 11.233 | 1.695 | 7.368 | 0.276 |
> | GOOSE  | 16.39% | 0.439 | 5.092 | 11.25 | 3.722 | 12.64 | 3.716 | 8.223 | 0.157 |
> | SAGE ($w_{adv=1.0}$) | 33.51% | 1.779 | 1.433 | 2.480 | 0.790 | 1.090 | 2.098 | 8.070 | 0.185 |

---

> > ### Comment · Reviewer_ZEJW · 2025-11-26
> > **Response to the authors**
> >
> > Thank you for your detailed explanation regarding to the method novelty and additional evaluations for addressing my concerns of the RL evaluation fairness. I have increased my score.

---

> > > ### Author Response · Authors · 2025-11-26
> > > **Response to Reviewer ZEJW**
> > >
> > > Dear Reviewer ZEJW,
> > >
> > > Thank you for your thoughtful re-evaluation of our work. We are very pleased that our detailed explanations and additional experiments have successfully addressed your concerns, and we sincerely appreciate your support and the increased score.
> > >
> > > Best regards,
> > >
> > > The Authors

---

> ### Author Response · Authors · 2025-11-19
> **Detailed Response to Reviewer ZEJW**
>
> *Q2：Adversarial types.*
>
> We have addressed this in our revised manuscript by providing more extensive visualizations.
> We have added **Figure 3: SAGE generates diverse types of meaningful adversarial behaviors (Replay policy)** to include a richer set of qualitative examples of different adversarial types. It showcases SAGE's ability to generate a variety of contextually appropriate adversarial maneuvers across different traffic situations, such as cut-in, rear-end, sudden-break, left-turn, opposite conflict, and turning overtake.
>
> In addition, another dimension of diversity is the **intensity** of the adversarial behavior. SAGE iis uniquely designed to provide continuous control over this aspect. As shown in **Fig. 5** (main paper) and further exemplified in **Fig. 14** (appendix), by simply adjusting the adversarial weight $w_adv$ at test time, SAGE can smoothly transition its generated behaviors from purely naturalistic to moderately assertive, and finally to highly aggressive. This steerability allows a user to systematically explore a whole spectrum of behavioral intensities for a given scenario, which is in itself a powerful form of targeted diversity generation.
>
> ---
> *Q4: Experimental setups.*
>
> Thank you for this detailed request for clarification on our experimental setup. We have added a comprehensive hyperparameter table (**Table 7: Hyperparameters for reward functions and HGPO**) in Appendix B.2 to provide this information. All other relevant details, such as the HGPO hyperparameters and reward settings are also listed in this table. For DenseTNT, we have used the publicly released model checkpoint from the CAT repository for experiments to ensure a unbiased comparison, as detailed in **Section B.1 Experimental setups, Pretrained Backbone Model** .
>
> **Hyperparameters for reward functions and HGPO.**
> | **Category** | **Hyperparameter** | **Value** |
> |:---|:---|:---|
> | HGPO Fine-tuning | Optimizer | AdamW |
> | | Learning Rate | $1.0 × 10^{-5}$ |
> | | Training Epochs | 200 |
> | | Batch Size | 1 (scenario) |
> | | Group Size (N) | 32 |
> | | Max Pairs per Group (K) | 8 |
> | | Preference Margin ($\delta_m$) | 0.2 |
> | | Beta (β) | 0.05 |
> | Preference Reward | Adversarial Expert ($\pi_{\theta_{adv}}$) | |
> | | $w_{adv} / w_{real}$ | 0.9 / 0.1 |
> | | Realism Expert ($\pi_{\theta_{real}}$) | |
> | |$ w_{adv} / w_{real}$ | 0.1 / 0.9 |
> | $R_{adv}$ Components | Collision Reward Scale | 10.0 |
> | | Proximity Reward Scale | 1.0 |
> | | Proximity Decay Rate | 0.2 |
> | P_{real} Components | $w_{turn}$ | 5.0 |
> | | $w_{stop-turn}$ | 3.0 |
> | | Kinematic Penalty Factor (Accel) | 5.0 |
> | | Kinematic Penalty Factor (Ang. Vel.) | 5.0 |
> | | Accel. Comfort Zone | $7.0 m/s^2$ |
> | | Lat. Accel. Comfort Zone | $6.0 m/s^2$ |
> | | Ang. Vel. Comfort Zone | 0.8 rad/s |
> | | Max Reasonable Turn | π radians |
> | Map Feasibility | Cross Solid Line Penalty | 50.0 |
> | | Crash Object Penalty | 10.0 |

---

### Official Review · Reviewer_mqnz · 2025-11-03

**Soundness:** 4
**Presentation:** 3
**Contribution:** 4
**Rating:** 8
**Confidence:** 4

**Summary:**

The authors propose a method for generating adversarial scenarios for driving by interpolating weights between two experts trained on opposing preferences. In contrast to previous works, SAGE’s approach balances adversarial behavior with realism by taking advantage of Linear Mode Connectivity, which enables adversarial behavior to be explored at test-time rather via re-training. Overall, I find this approach to be novel, useful, and interesting.

**Strengths:**

- The motivation and methodology are well-written and well-justified.
- Additional results in supplementary is helpful.
- Authors provide very extensive details on experiment setup. Without the  code provided, I feel as though the work is reproducible based on the text.
- Authors show theoretical and empirical applicability of LMC to traffic simulation problems.
- Results are comprehensive and provide actionable insight to the simulation community.

**Weaknesses:**

- Even though a lot of information is referenced and provided in the appendices, I found it difficult to interpret results off first glance in the main body. I suggest the authors make captions of Figures and Tables more self-contained and avoid having the reader look for references. It would be nice to know off first glance 1) what policies govern adv and ego vehicles, and 2) if certain qualitative results correspond to any tabular results. For example, I found Figure 2 difficult to interpret. Even though the images show that SAGE produces more realistically adversarial behavior, I’m not exactly sure what policy the ego vehicle is following. The main body also does not seem to specify.
- Some notation definitions are missing (see question 1 below).

**Questions:**

1. Eq 2. Not sure if I missed something, but I’m a bit confused at the -P_{real} term in the reward definition. “P_{real} ensures tau to adhere to the patterns of naturalistic driving; it should be statistically plausible.. “. To me, it would make sense that the objective should reward both adversarial and realistic behavior. Is P_{real} supposed to denote the probability of the trajectory being drawn from the distribution of realistic trajectories? In which case, shouldn’t the term’s sign be flipped (unless it denotes NLL)?
2. Table 1: Are the baselines (other than IDM) trained with a similar reward objective, even if it is the naive objective corresponding to a single point on the Pareto frontier as mentioned before?
3. (Appendix) Table 6/7: How is KING producing a distributional difference of ~256? It would be nice if authors discussed this / provided examples to demonstrate this failure mode.
4. Figure 2: What policy is governing the ego vehicle? Tables 1 and 2 present results against two different ego vehicle policies, so I am confused about which results Figure 2 correspond to.
5. At what point (if any) of increasing W_{adv} result in violation of map rules? Does the enforced “feasibility first” inductive bias ensure that all adversarial agents avoid violating map constraints, even if W_{adv} increases?
6. A very common adversarial scenario where the adversarial vehicle is at fault is rear end collisions (the ego vehicle is in front). Does the adversarial policy demonstrate this behavior when the log begins with the adversarial agent behind the ego vehicle in the same lane? I noticed that a lot of the qualitative examples (Figure 11) demonstrate cut-in behavior. I was curious if the adversarial policy was mostly limited to cut-ins, or if there are other types of adversarial behavior types.

---

> ### Author Response · Authors · 2025-11-19
> **General response to Reviewer mqnz**
>
> We sincerely thank the reviewer for their positive and insightful feedback on our work. We are greatly encouraged by the recognition of our approach as "novel, useful, and interesting," and we appreciate the commendation of our motivation, methodology, and the theoretical and empirical contribution to this domain.
>
> We are also grateful for the constructive suggestions on improving the paper's clarity and rigor. In response to your valuable feedback, we have made the following revisions to the paper:
> -  **Enhanced clarity:** We have revised all figure and table captions to be more self-contained and informative.
> -   **Detailed baselines:** We have clarified the objective functions for all baseline methods and included new qualitative examples to explain their failure modes.
> -   **Additional qualitative results:** We have added several new visualizations to demonstrate the diversity of adversarial behaviors generated by SAGE.
> -   **New discussions:** We have offered more detailed explanations on key design choices.
>
>
> We believe these revisions have substantially improved the quality and readability of our paper. We appreciate the opportunity to clarify these points and hope our detailed responses have fully addressed your questions.

---

> ### Author Response · Authors · 2025-11-19
> **Detailed Response to Reviewer mqnz**
>
> *W1: Clarification of table and figure captions.*
>
> Thank you for your feedback on improving the clarity and readability of our figures and tables. In our revised manuscript, we have carefully **reviewed and updated the captions for all figures and tables** in the main body to ensure they are more descriptive and provide the necessary context at a glance. Specifically, we have made sure to explicitly state the ego policy under test for each relevant result, such as in the **updated captions for Figure 2 and Tables 1 and 2**. We believe these changes make the results much easier to interpret and better connect the qualitative visualizations with their corresponding quantitative data.
>
> ---
> *Q1: Clarification of the realism penalty.*
>
> We thank the reviewer for raising this question. The reviewer is correct that our objective is to encourage both adversariality and realism. However, we formulate this by **rewarding adversarial potency** while simultaneously **penalizing deviations from realism**. Therefore, $P_{\text{real}}$ is not a reward or a probability, but a **penalty function**. Its value increases as a trajectory becomes more unrealistic. The negative sign in Eq. 2 correctly reflects our goal of minimizing this penalty. As detailed in **Appendix B.2**, $P_{\text{real}}$ is composed of  a kinematic penalty ($P_{\text{kin}}$) and a behavioral penalty ($P_{\text{beh}}$). These functions are specifically designed to penalize individual trajectories for violating physical comfort limits and naturalistic driving patterns.
>
> We agree with the reviewer's point that enforcing realism could also be achieved by rewarding the likelihood of a trajectory under a distribution of real-world data. Our chosen approach, using penalties on individual metrics, serves as a highly **effective and computationally efficient proxy** for this distributional constraint. Experimental results confirm that our penalty-based formulation successfully constrains the generator to produce scenarios that are **both individually plausible and distributionally consistent**.
>
> ---
> *Q2: Objective of baselines.*
>
> We have now added a detailed description of each baseline's objective function in the revised manuscript in **Appendix B.1, under the "Baselines" subsection**, to provide more details on our experimental design.
> *   **For CAT:** We strictly followed the original paper's method. CAT employs a heuristic objective: from a set of trajectory proposals generated by a motion model, it selects the one that maximizes the probability of collision with the ego vehicle: $\tau^*_{\text{adv}}$ such that $\tau ^ {\*} _ {\text{adv}} = \arg\max_{\tau_i \in \mathcal{T}} P(\text{coll}(\tau_i, \tau_{\text{ego}}))$, where $\mathcal{T}$ is the set of candidate trajectories.
> *   **For KING and AdvTrajOpt:** To ensure the most direct and fair comparison, we configured these gradient-based and optimization-based methods to use the **same objective function as SAGE**, as defined in Equation (2). Specifically, we used a fixed, high weight for the adversarial term ($w_{\text{adv}}$) and a low weight for the realism term ($w_{\text{real}}$).
> *   **For SEAL:** We used the official implementation of SEAL, as it was also evaluated on the WOMD dataset. SEAL utilizes a **learned skill function** which is trained to predict not only collision closeness but also the induced deviation in the ego agent's behavior. The objective is to maximize the expected sum of these two learned criticality scores: $\mathbb{E}[f_{\text{coll}} + f_{\text{diff}}]$, where $f_{\text{coll}}$ and $f_{\text{diff}}$ are predicted by a scoring network.
> * Furthermore, to clarify the results for SAGE in Tables 1, 2, 6, and 7, the **different variants** of our method (e.g., SAGE ($w_{\text{adv}}=0.5$), etc.) also utilize the same reward function from Eq. (3). The only difference is the specific weight $w_{\text{adv}}$ used to train the model.
>
> ---

---

> ### Author Response · Authors · 2025-11-19
> **Detailed Response to Reviewer mqnz**
>
> *Q3: Failure case of baselines.*
>
> The unusually high distributional distance for KING stems from specific failure modes where the optimization process generates kinematically implausible trajectories. KING generates adversarial trajectories by optimizing a sequence of actions through a differentiable kinematic bicycle model. While this model enforces basic non-holonomic constraints, it lacks strong physical priors on higher-order dynamics, such as jerk or maximum acceleration. Consequently, the gradient-based optimizer can exploit this simplified model to produce solutions with extreme changes in velocity. This can manifest as an adversarial vehicle appearing to move or brake/accelerate with physically impossible force, leading to a massive divergence in the velocity distribution compared to naturalistic data.
>
> We have added a new case study in our revised appendix (as **Figure 8: Unexpected case of KING and Rule baselines, in Appendix D.1**). This figure visualizes a failure case of KING, clearly showing the adversarial vehicle undergoing an instantaneous velocity change that is physically unrealistic. We have also included a visualization of a  failure case of the Rule-based generator, showing how it can sometimes result in trajectories with extremely high kinematic penalties (e.g., large "Kine. Pen" in Table 6/7) due to unnatural and sharp turns.
>
> ---
> *Q4: Ego policy settings.*
>
> We apologize for the confusion caused in the captions due to the page limit in the first submission. The ego vehicle in the original Fig. 2 was governed by the Replay policy. To improve the clarity of our paper, we have taken the following actions in the revised manuscript:
> - We have reviewed and updated the captions for all figures and tables in the experiments section to be self-contained and explicit.
> - We have added a new section in **Appendix D.5.3 Examples of Adversarial Generation against Different Ego Policies** that provides additional examples of our generated scenarios against different tested ego policies.
>
> ---
> *Q5: Map feasibility.*
>
> We admit that our approach does not provide a hard guarantee of 100% map compliance. The "feasibility first" principle is an **inductive bias** injected during the preference fine-tuning process, **not a hard-coded post-hoc filter**. It strongly guides the policy to learn a representation where map-compliant trajectories are heavily **preferred** over non-compliant ones. However, since our method is based on optimizing a probabilistic model, there remains a small but non-zero probability of generating infeasible trajectories.
>
> Our ablation study in **Figure 6(d)** provides a quantitative answer. It shows that enforcing this inductive bias allows the model to achieve a high map feasibility rate of nearly **90%**. While this is a dramatic improvement compared to variants without this constraint, it is not a perfect 100%. This means that even with this principle, about 10% of the initially generated trajectory candidates may still violate map constraints. As demonstrated by the consistently low map compliance penalties ("Crash Obj.", "Cross Line") for SAGE across all test conditions in **Tables 1, 2, 8, and 9**, even at the most aggressive setting ($w_{\text{adv}}=1.0$). This confirms that while not an absolute guarantee, our approach is highly effective at steering the generation towards valid and plausible scenarios.
>
> ---
> *Q6: Different types of adversarial behaviors.*
>
> In response to the reviewer's query, SAGE can generate rear-end collision scenarios if the initial setup allows for it. However, in this case the following vehicle (the adversary) is presumed to be at fault in the event of a rear-end collision. While this is still a valid failure scenario, our generator often discovers more subtle and complex ones that offer more value for testing purposes, as they probe the ego agent's decision-making and prediction capabilities in ambiguous situations.
>
> To better demonstrate this diversity, we have significantly expanded our qualitative examples in the revised paper (**Figure 3: SAGE generates diverse types of meaningful adversarial behaviors (Replay policy).**). The new figures showcase a much wider variety of generated adversarial behaviors, which are challenging yet plausible long-tail scenarios.

---

### Author Response · Authors · 2025-11-19
**General Response to All Reviewers**

We sincerely thank all reviewers for their insightful feedback and positive assessment of our work. We are encouraged that the reviewers recognize SAGE as **"novel, useful, and interesting" (Reviewer mqnz)** and appreciate its ability to **"balance adversarial behavior with realism" (Reviewers mqnz, jZqG)** without retraining. We are particularly grateful for the recognition that **"the theoretical concepts brought to autonomous driving really make the reviewed paper suitable for ICLR" (Reviewer be6Z)**.

Our work introduces SAGE, **a new paradigm for adversarial scenario generation**. SAGE is the first framework to reframe this task as a **test-time preference alignment problem**, enabling **fine-grained, Pareto-optimal control** over the trade-off between adversariality and realism. Crucially, we ground this steerability in the **Linear Mode Connectivity (LMC)** hypothesis, providing **novel theoretical insights** into why weight interpolation is effective for motion generation.

In response to the valuable suggestions from the reviewers, we have made significant revisions and detailed responses to strengthen the paper’s theoretical rigor, experimental fairness, and clarity. The major updates are summarized below:
-   **Fairer closed-loop training benchmarks (response to Reviewers ZEJW,  jZqG):** To rigorously assess generalization and rule out overfitting, we conducted new **cross-evaluation experiments**. We now benchmark all trained agents on held-out test sets generated by different methods (CAT, Rule-based, etc.). These results confirm that the SAGE-trained agent demonstrates superior generalizability and robustness across diverse adversarial distributions.
-   **Strengthened theoretical justification (response to Reviewer be6Z):** We provided a **formal derivation and clearer intuition** to justify the superiority of weight-space mixing over output ensembling (Remark 1 in Appendix). We mathematically show that the concavity of the high-reward ridge connecting experts (a consequence of LMC) makes weight mixing the dominant strategy. This claim is now strongly supported by our empirical reward landscape analysis.
-  **New baselines & ablations (response to Reviewers ZEJW, be6Z):** We integrated the recently published **GOOSE** method as a new baseline. Additionally, we included a new ablation study on the preference margin ($\delta_m$) and provided a comprehensive hyperparameter table to ensure reproducibility.
-   **Enhanced clarity & visualizations (response to Reviewers mqnz, ZEJW):** We revised all figure/table captions to be self-contained, clarified the objective functions for all baselines, and added extensive qualitative examples showing the diversity of generated scenarios and failure cases of baseline methods.

We believe these revisions have substantially improved the quality of our work. We detail our specific responses to each reviewer below.

---

### Author Response · Authors · 2025-12-01
**Summary of Response and Discussion During the Rebuttal Phase**

**Authors' Final Comments:**

Dear Chairs,

**TL;DR:** Our paper's average score increased from **6.0 (top ~3.84%) to 7.5 (top ~0.2%)** following a constructive discussion period. All three engaging reviewers (R-ZEJW, R-be6Z, R-jZqG) raised their scores after we addressed their concerns with new experiments and theoretical clarifications. Crucially, **all reviewer discussions, replies, and score updates occurred between Nov 19th and Nov 25th (AoE), well before the public disclosure of the OpenReview incident on Nov 27th.**


To provide a clear overview, the table below summarizes the review process. All score changes were made by the reviewers themselves after their concerns were addressed.

| Reviewer ID | Initial Score | Post-Discussion Score | Confidence | Last Modified Time (AoE) |
| :--- | :---: | :---: | :---: | :--- |
| **R-mqnz** | 8 | 8 (no change) | 4 | 02 Nov 2025, 17:13 |
| **R-ZEJW** | 4 | **6** | 3 | 25 Nov 2025, 21:11 |
| **R-be6Z** | 6 | **8** | 5 | 19 Nov 2025, 07:25 |
| **R-jZqG** | 6 | **8** | 4 | 25 Nov 2025, 08:12 |

---

**Summary of the Rebuttal Phase**

Our rebuttal and the subsequent discussions with the reviewers were highly constructive and concluded with a significant positive consensus on our work. The reviewers' feedback led to substantial improvements in our manuscript, and we are grateful for their diligent and fair evaluation. All interactions were conducted with the utmost academic integrity.


This constructive discussion raised our average score from **6.0 (top ~3.84%)** to **7.5 (top ~0.2%)**. The following is a brief summary of how we addressed each reviewer's key concerns, leading to their decision to raise the score:

*   **Reviewer ZEJW (4 → 6):** The reviewer's primary concerns were the novelty of our method and the fairness of the RL evaluation. To address this, we **clarified our core novel contributions** (reframing adversarial generation as a preference alignment problem and being the first to ground it in the LMC hypothesis for this domain). Most importantly, we conducted **new cross-evaluation experiments**, benchmarking all agents on test sets generated by different methods. The reviewer was satisfied that these additions **“addressed their concerns” and subsequently raised their score**.

*   **Reviewer be6Z (6 → 8):** The reviewer's main questions were the theoretical justification for the superiority of weight-space mixing and requests for experimental details. We provided a **new formal derivation** and clearer intuition for our theoretical claims, directly linking them to empirical evidence in our figures. We also added a comprehensive hyperparameter table and a new ablation study. The reviewer found the work **"significant and deserves a good mark" and raised their score**.

*   **Reviewer jZqG (6 → 8):** The reviewer raised a critical point about potential overfitting in the closed-loop evaluation. Following the excellent suggestion, we performed the **cross-evaluation experiment**. The new results demonstrated the superior generalization of our SAGE-trained agent. The reviewer stated that **"With this evaluation, the methods are more solid and convincing" and raised their score to 8**.

We are deeply grateful to the reviewers for their time, expertise, and constructive engagement, which have significantly strengthened our paper. We trust that this summary, along with the detailed rebuttal history, provides a clear picture of the thorough and positive peer-review process our submission has undergone.

Thank you for your understanding and for your dedicated service to the community in these challenging times. We look forward to your final decision.

Sincerely,

The Authors

---

### Meta-Review · Area_Chair_Pc5m · 2026-01-07

**Summary:**

This paper introduces SAGE, a new paradigm for adversarial scenario generation.
The AC has checked all reviewers' comments and author responses.

The major concerns in the rebuttals were:

1) Fairer closed-loop training benchmarks,
2) Theoretical justification,
3) New baselines & ablations,
4) Enhanced clarity & visualizations.

During the rebuttal, these major concerns were addressed by the authors. The reviewers mostly raised their scores finally.

The AC is positive towards this paper's research novelty and contribution to ICLR community.

**Reviewer Concerns:**

Same as above.

**Reviewer Scores:**

Although not all participated in the discussion, they checked the author responses and interact with the authors time by time.

Their ratings were changed based on the active response from authors.

---

### Decision · Program_Chairs · 2026-01-26

Accept (Poster)